# Bifurcate then Alienate: Incomplete Multi-view Clustering via Coupled Distribution Learning with Linear Overhead

**Shengju Yu** [1]  **Yiu-ming Cheung** [1]  **Siwei Wang** [2]  **Xinwang Liu** [3]  **En Zhu** [3]

## Abstract

Despite remarkable advances, existing incomplete multi-view clustering (IMC) methods typically leverage either perspective-shared or perspective-specific determinants to encode cluster representations. To address this limitation, we introduce a BACDL algorithm designed to explicitly capture both concurrently, thereby exploiting heterogeneous data more effectively. It chooses to bifurcate feature clusters and further alienate them to enlarge the discrimination. With distribution learning, it successfully couples view guidance into feature clusters to alleviate dimension inconsistency. Then, building on the principle that samples in one common cluster own similar marginal distribution and conditional distribution, it unifies the association between feature clusters and sample clusters to bridge all views. Thereafter, all incomplete sample clusters are reordered and mapped to a common one to formulate clustering embedding. Last, the overall linear overhead endows it with a resource-efficient characteristic.

## 1. Introduction

In the era of information, heterogeneous data that are commonly gathered from various channels and modalities of the same one object are growing more prevalent (Ma et al., 2024b; Wang et al., 2023; Yu et al., 2025; Zhang et al., 2025). Accordingly, how to effectively excavate out valuable potential patterns from this type of data is grasping increasing attention (Li et al., 2025; Yu et al., 2024b; Liang et al., 2024). Multi-view clustering (MVC) technology, in virtue of the ability to seamlessly integrate multi-source information and powerfully partition samples into distinct sets without the need of any labels known in advance, is generally perceived as an encouraging method to tackle these data, and has been widely deployed in fraud detection, personalized medicine, social network analysis, etc, (Zhang et al., 2021; Yu et al., 2023; Zhang et al.; Liu et al., 2024). The prerequisite for MVC algorithms' proper execution is that all views are required to be complete (Yu et al., 2024d; Ma et al., 2024a; Gu et al., 2024a; Wan et al., 2024). Due to equipment defects or collector faults, however, in real-life it inevitably causes certain views having missing samples, inducing the incomplete multi-view clustering (IMC) issue (Yu et al.; Wang et al., 2021a; Liu et al., 2023a; Yu et al., 2024a; Gu & Feng).

To cope with IMC tasks, recently, a great deal of promising methods have been carefully presented (Xu et al., 2022; Wang et al., 2021b; Yu et al., 2024c; Tang & Liu, 2022). For instance, Wang et al. (2022) adopt consensus bipartite affinity to characterize arbitrary views and jointly construct anchors using an unified learning mechanism. Li et al. (2023b) decrease the disturbance of superfluous properties through projecting operations and employ low-rank tensor regularizers to leverage high-order representation inside samples. Gu et al. (2024b) utilize dictionary learning strategy to recover missing parts and integrate Gaussian error rank into Laplacian manifold optimization to explore local correlations. Motivated by prototype advances, Li et al. (2024) build up the conjunction between prototypes and observed instances to avoid the generation of full-sized similarity and directly formulate the overall graph structure without additional hyper-parameter searching. These methods enhance the clustering quality from various aspects, nevertheless, they investigate either perspective-shared or perspective-specific determinants to encode cluster representation. This single paradigm could not sufficiently exploit the interrelations among data features, restricting the model's performance.

To get rid of this limitation, in the manuscript we propose an IMC algorithm named BACDL, and its overall pipeline is described in Fig. 1. Concretely, inspired by non-negative matrix factorization, we choose to bifurcate feature clusters and each bifurcation is explicitly interconnected to a type of determinants. Through mutual exclusion learning, we alienate them to strengthen their discrimination. Subsequently, in virtue of distribution learning, we couple view

[1]Department of Computer Science, Hong Kong Baptist University. [2]Intelligent Game and Decision Lab. [3]National University of Defense Technology. yu-shengju@foxmail.com. Correspondence to: Yiu-ming Cheung <ymc@comp.hkbu.edu.hk>.

*Proceedings of the $42^{nd}$ International Conference on Machine Learning*, Vancouver, Canada. PMLR 267, 2025. Copyright 2025 by the author(s).

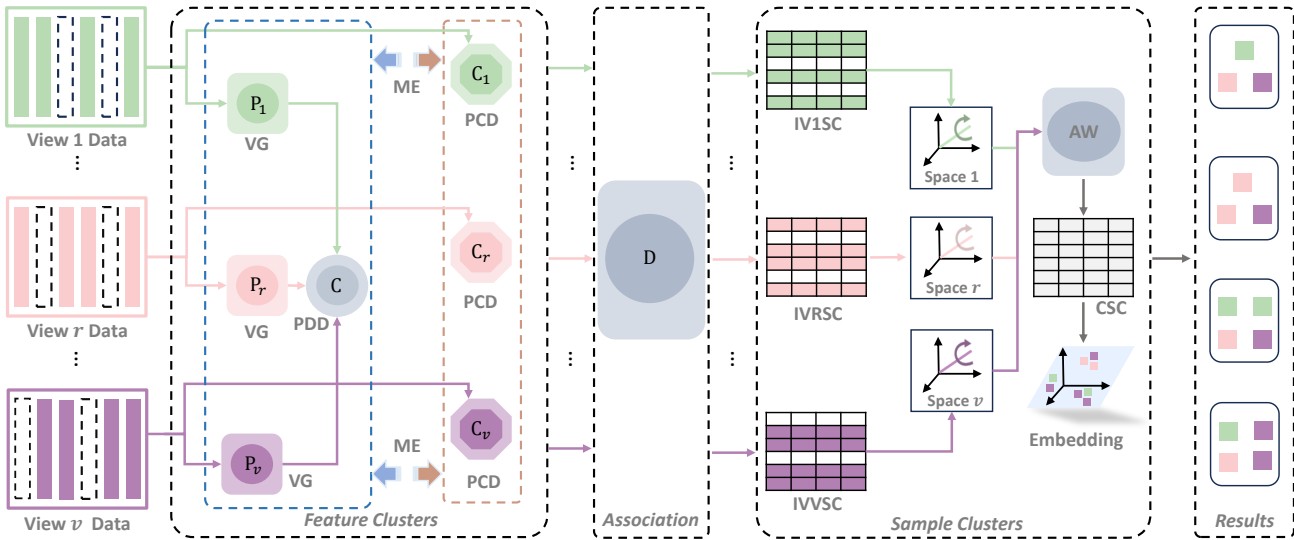

*Figure 1.* Pipeline of proposed BACDL. It firstly bifurcates feature clusters on each incomplete view and magnifies the discrimination between determinants via mutual exclusion learning. Then, it couples view guidance into feature clusters through distribution learning to alleviate the dimension inconsistency. Subsequently, it unifies the association between feature clusters and sample clusters to bridge all views. Further, all sample clusters are reordered in latent subspace and, following adaptive weighting, mapped to a common one to constitute clustering embedding. VG: view guidance; PDD: perspective-shared determinant; PCD: perspective-specific determinant; ME: mutual exclusion; D: association between feature clusters and sample clusters; AW: adaptive weighting; IV1SC: incomplete view 1 sample clusters; IVRSC: incomplete view $r$ sample clusters; IVVSC: incomplete view $v$ sample clusters; CSC: common sample clusters.

guidance into feature clusters to eliminate the inconsistent dimensions. Further, relying on the fact that samples in one common cluster are with similar marginal distribution and conditional distribution, we unify the association between feature clusters and sample clusters to bridge all views. Afterwards, we reorder sample clusters by space rotation and map them after adaptive weighting onto a common one to form clustering embedding. Then, for efficiently minimizing the formulated objective function, we give a nine-step updating rule with overall linear overhead and theoretical convergence. For verifying the effectiveness of BACDL, we organize extensive comparison experiments under multiple missing ratios. To sum up, in this manuscript we (1) propose a new IMC learning paradigm, which achieves the simultaneous exploration of both perspective-shared determinants and perspective-specific determinants via coupled distribution learning; (2) design a updating rule with overall linear overhead, enhancing the practicality (3) conduct comprehensive experiments, demonstrating the effectiveness and merits of presented BACDL method from multiple aspects.

## 2. Related Works

To effectively handle IMC problems, in recent years researchers have proposed many prominent algorithms. Xu et al. (2023) impose an adaptive projection in feature space to evade imputation, and formulate cluster structures by si-

multaneously enlarging mutual information and shrinking mean discrepancy. Ren et al. (2024) advance imputation fidelity in an unsupervised manner and conduct data matching across clients by utilizing the sample homogeneity and view multi-functionality. Inspired by contrastive learning, Yang et al. (2022) utilize observed pairs as positives and randomly-selected cross-view samples as negatives to eliminate the partial unalignment and alleviate the noisy impact. Zhao et al. (2023) introduce consistency constraint to maintain the similarity between graphs on different views and utilize the tensor means to extract graph correlations in a manifold space. Orthogonal to them, He et al. (2023) learn asymmetric similarities based on structural anchors to replace distance-based weighting and extend late fusion to general scenarios for affinity construction. Sun et al. (2023) generate uniform probability representation by relaxing standard spectrum to increase the stability and integrate a balance constraint to adaptively exploit intra-view features. Li et al. (2023a) design a dual-stream mechanism to model prototype-sample similarity and utilize the relationship between available views to conduct sample recovery. With the idea of neighbor group, Wong et al. (2023) produce neighbor sets for each sample pair to partition structure embedding and encode view-missing position to guide the fusion of individual graph. Zhang et al. (2024) introduce kernelized subspace to extract intrinsic structure between views and perform low-rank learning and affinity construction jointly.

## 3. Preliminary

Non-negative matrix factorization (Ding et al., 2006; Long et al., 2012; Ding et al., 2010) is usually deemed as a powerful means to tackle multi-view data. The basic idea is formulated as

$$\min_{\mathbf{C}_r \geq 0, \mathbf{D}_r \geq 0, \mathbf{E}_r \geq 0} \sum_{r=1}^{v} \left\| \mathbf{X}_r - \mathbf{C}_r \mathbf{D}_r \mathbf{E}_r^\top \right\|_F^2, \quad (1)$$

where $\mathbf{X}_r \in \mathbb{R}^{d_r \times n}$ denotes the (complete) data matrix on view $r$. $d_r$ is the feature dimension on view $r$, and $n$ is the number of samples. $\mathbf{C}_r \in \mathbb{R}^{d_r \times w}$ denotes $w$ feature clusters. Each column of $\mathbf{C}_r$ represents a probability distribution on $d_r$ features. $t^* = \arg\max_t (\mathbf{C}_r)_{s,t}$ indicates that feature $s$ is affiliated to feature cluster $t^*$ where $(\mathbf{C}_r)_{s,t}$ is the element in the $s$-th row and $t$-th column of $\mathbf{C}_r$. The matrix $\mathbf{E}_r \in \mathbb{R}^{n \times z}$ denotes $z$ sample clusters. Each row of $\mathbf{E}_r$ represents a probability distribution. $t^* = \arg\max_t (\mathbf{E}_r)_{s,t}$ indicates that sample $s$ is affiliated to sample cluster $t^*$. $\mathbf{D}_r \in \mathbb{R}^{w \times z}$ denotes the association matrix between $\mathbf{C}_r$ and $\mathbf{E}_r$. The element $(\mathbf{D}_r)_{s,t}$ expresses the association probability between feature cluster $s$ and sample cluster $t$.

Besides, the formula $\min_{\mathbf{C}_r, \mathbf{D}_r, \mathbf{E}_r} \left\| \mathbf{X}_r - \mathbf{C}_r \mathbf{D}_r \mathbf{E}_r^\top \right\|_F^2$ can be equivalently deemed as $\min_{\mathbf{C}_r, \mathbf{Z}_r} \left\| \mathbf{X}_r - \mathbf{C}_r \mathbf{Z}_r \right\|_F^2$ and $\min_{\mathbf{D}_r, \mathbf{E}_r} \left\| \mathbf{Z}_r - \mathbf{D}_r \mathbf{E}_r^\top \right\|_F^2$. $\mathbf{C}_r$ maps the original data $\mathbf{X}_r$ to the potential space $\mathbf{Z}_r$. Correspondingly, the marginal distribution $P(\mathbf{x}_r)$ is transformed to $P(\mathbf{z}_r)$. $\mathbf{C}_r$ learns the marginal distribution. Similarly, $\mathbf{D}_r$ maps the potential space $\mathbf{Z}_r$ to the sample clusters $\mathbf{E}_r$. Correspondingly, the conditional distribution $P(y|\mathbf{x}_r)$ is transformed to $P(y|\mathbf{z}_r)$. $\mathbf{D}_r$ learns the conditional distribution.

## 4. Proposed Model

Firstly, for incomplete dataset $\{\mathbf{X}_r\}_{r=1}^{v}$, in order to work with the incompleteness, we introduce the index matrix $\mathbf{G}_r \in \mathbb{R}^{n \times n_r}$ where $(\mathbf{G}_r)_{i,j} = 1$ if $(\mathbf{w}_r)_j == i$ otherwise $(\mathbf{G}_r)_{i,j} = 0$ for any $j \in [1, 2, \cdots, n_r]$. $\mathbf{w}_r \in \mathbb{R}^{n_r \times 1}$ is the indicator vector. $n_r$ is the number of samples observed on view $r$. Accordingly, $\mathbf{X}_r \mathbf{G}_r \in \mathbb{R}^{d_r \times n_r}$ denotes the available samples on the $r$-th view.

Then, to achieve double-determinant exploration, we bifurcate the feature clusters. Specifically, we attempt to jointly utilize $\mathbf{C} \in \mathbb{R}^{k \times k}$ and $\mathbf{C}_r \in \mathbb{R}^{d_r \times k}$ to exploit perspective-shared determinants and perspective-specific determinants respectively where $k$ is the number of clusters. However, due to the feature dimension inconsistency, it is hard to embed $\mathbf{C}$ into feature clusters. To alleviate this dilemma, we introduce view guidance $\mathbf{P}_r \in \mathbb{R}^{d_r \times k}$ to assist $\mathbf{C}$ exploring inter-view characteristics. Accordingly, the feature clusters are coupled as $[\mathbf{P}_r \mathbf{C} | \mathbf{C}_r]$. Additionally, given the probability distribution characteristic of feature clusters, it

needs satisfying the constraint $[\mathbf{P}_r \mathbf{C} | \mathbf{C}_r]^\top \mathbf{1}_{d_r} = \mathbf{1}_{2k}$. So, the loss can be defined as

$$\min_{\mathbf{C}, \mathbf{C}_r, \mathbf{D}_r, \mathbf{E}_r, \mathbf{P}_r,} \sum_{r=1}^{v} \left\| \mathbf{X}_r \mathbf{G}_r - [\mathbf{P}_r \mathbf{C} | \mathbf{C}_r] \mathbf{D}_r \mathbf{E}_r^\top \mathbf{G}_r \right\|_F^2$$

$$\text{s.t. } \mathbf{C} \geq 0, \mathbf{C}_r \geq 0, \mathbf{D}_r \geq 0, \mathbf{E}_r \geq 0, \mathbf{P}_r \geq 0,$$

$$[\mathbf{P}_r \mathbf{C} | \mathbf{C}_r]^\top \mathbf{1}_{d_r} = \mathbf{1}_{2k}. \quad (2)$$

Further, in conjunction with the probability distribution characteristic of sample clusters, we have that it needs satisfying $\mathbf{G}_r^\top \mathbf{E}_r \mathbf{1}_k = \mathbf{1}_{n_r}$. On the other hand, in light of the fact that samples in one common cluster are with similar marginal distribution and conditional distribution, we unify the association between feature clusters and sample clusters to bridge all views. Besides, considering that views could have different levels of contributions, we assign a weight for each view to automatically balance them. Consequently, the loss can be designed as

$$\mathcal{L}_0 = \min_{\Theta} \sum_{r=1}^{v} a_r^2 \left\| \mathbf{X}_r \mathbf{G}_r - [\mathbf{P}_r \mathbf{C} | \mathbf{C}_r] \mathbf{D} \mathbf{E}_r^\top \mathbf{G}_r \right\|_F^2$$

$$\text{s.t. } \mathbf{C} \geq 0, \mathbf{C}_r \geq 0, \mathbf{D} \geq 0, \mathbf{E}_r \geq 0, \mathbf{P}_r \geq 0, a_r \geq 0,$$

$$\sum_{r=1}^{v} a_r = 1, [\mathbf{P}_r \mathbf{C} | \mathbf{C}_r]^\top \mathbf{1}_{d_r} = \mathbf{1}_{2k}, \mathbf{G}_r^\top \mathbf{E}_r \mathbf{1}_k = \mathbf{1}_{n_r}, \quad (3)$$

where $\Theta = \{\mathbf{C}, \mathbf{C}_r, \mathbf{D}, \mathbf{E}_r, \mathbf{P}_r, a_r\}$. $[\cdot|\cdot]$ denotes the matrix concatenation operation.

Thereafter, to enhance the discrimination between double determinants, we make $\mathbf{P}_r \mathbf{C}$ and $\mathbf{C}_r$ alienated from each other. Specially, by point-to-point alienation, we utilize the matrix inner product $\langle \mathbf{P}_r \mathbf{C}, \mathbf{C}_r \rangle$ to render them as dissimilar as possible. So, we have the following loss,

$$\mathcal{L}_1 = \min_{\mathbf{C}, \mathbf{C}_r, \mathbf{P}_r,} \langle \mathbf{P}_r \mathbf{C}, \mathbf{C}_r \rangle \quad (4)$$

$$\text{s.t. } \mathbf{C} \geq 0, \mathbf{C}_r \geq 0, \mathbf{P}_r \geq 0, [\mathbf{P}_r \mathbf{C} | \mathbf{C}_r]^\top \mathbf{1}_{d_r} = \mathbf{1}_{2k}.$$

Subsequently, sample clusters on different views could be misregistered due to the unsupervised property, and we associate a transformation space for the sample clusters on each view to reorder them. Further, we map reordered sample clusters to a common one so as to formulate full clustering embedding. Consequently, the loss is defined as

$$\mathcal{L}_2 = \max_{\mathbf{E}, \mathbf{F}_r, b_r} \text{Tr} \left( \mathbf{E}^\top \sum_{r=1}^{v} b_r \mathbf{E}_r \mathbf{F}_r \right) \quad (5)$$

$$\text{s.t. } \mathbf{E}^\top \mathbf{E} = \mathbf{I}_k, \mathbf{F}_r \mathbf{F}_r^\top = \mathbf{I}_k, \sum_{r=1}^{v} b_r^2 = 1, b_r \geq 0,$$

where $b_r$ is a weight variable.

Therefore, the loss of proposed BACDL is formulated as

$$\mathcal{L} = \mathcal{L}_0 + \lambda \mathcal{L}_1 - \beta \mathcal{L}_2. \quad (6)$$

## 5. Updating Rule

▶ When updating $\mathbf{E}_r$, the loss equivalently becomes

$$\min_{\mathbf{E}_r} a_r^2 \left\| \mathbf{X}_r \mathbf{G}_r - \widehat{\mathbf{A}}_r \mathbf{E}_r^\top \mathbf{G}_r \right\|_F^2 - \beta b_r \operatorname{Tr}\left( \mathbf{E}^\top \mathbf{E}_r \mathbf{F}_r \right)$$
$$\text{s.t. } \mathbf{E}_r \geq 0, \mathbf{G}_r^\top \mathbf{E}_r \mathbf{1}_k = \mathbf{1}_{n_r}, \tag{7}$$

where $\widehat{\mathbf{A}}_r = [\mathbf{P}_r \mathbf{C} | \mathbf{C}_r] \mathbf{D}$.

We can derive the following updating rule for $\mathbf{E}_r$,

$$(\mathbf{E}_r)_{i,j} \leftarrow (\mathbf{E}_r)_{i,j} \left[ \frac{\left( \widehat{\mathbf{B}}_r \mathbf{X}_r^\top \widehat{\mathbf{A}}_r + \frac{\beta}{2} b_r (\widehat{\mathbf{C}}_r)_{pos} \right)_{i,j}}{\left( \widehat{\mathbf{B}}_r \mathbf{E}_r \widehat{\mathbf{A}}_r^\top \widehat{\mathbf{A}}_r + \frac{\beta}{2} b_r (\widehat{\mathbf{C}}_r)_{neg} \right)_{i,j}} \right]^{\frac{1}{2}} \tag{8}$$

where $\widehat{\mathbf{B}}_r = a_r^2 \mathbf{G}_r \mathbf{G}_r^\top$, $\widehat{\mathbf{C}}_r = \mathbf{E} \mathbf{F}_r^\top$. $(\cdot)_{pos}$ and $(\cdot)_{neg}$ denote the positive part and negative part respectively.

Denote the loss (7) as $\mathcal{L}(\mathbf{E}_r)$. We give the auxiliary function definition of $\mathcal{L}(\mathbf{E}_r)$.

**Definition 1.** $\mathcal{F}(\mathbf{E}_r, \widetilde{\mathbf{E}}_r)$ is an auxiliary function of $\mathcal{L}(\mathbf{E}_r)$ when for any $\mathbf{E}_r$ and $\widetilde{\mathbf{E}}_r$ it satisfies $\mathcal{F}(\mathbf{E}_r, \widetilde{\mathbf{E}}_r) \geq \mathcal{L}(\mathbf{E}_r)$ and $\mathcal{F}(\mathbf{E}_r, \mathbf{E}_r) = \mathcal{L}(\mathbf{E}_r)$ (Ding et al., 2010).

Then, we have the following theorem holds,

**Theorem 1.** *Eq. (9) is an auxiliary function of $\mathcal{L}(\mathbf{E}_r)$ and also convex.*

$$\mathcal{F}(\mathbf{E}_r, \widetilde{\mathbf{E}}_r) = a_r^2 \sum (\mathbf{G}_r \mathbf{G}_r^\top \widetilde{\mathbf{E}}_r \widehat{\mathbf{A}}_r^\top \widehat{\mathbf{A}}_r)_{i,j} (\widehat{\mathbf{H}}_r)_{i,j}$$
$$- 2 a_r^2 \sum (\widetilde{\mathbf{E}}_r)_{i,j} (\mathbf{G}_r \mathbf{G}_r^\top \mathbf{X}_r^\top \widehat{\mathbf{A}}_r)_{i,j} (\widehat{\mathbf{D}}_r)_{i,j}$$
$$- \beta b_r \sum (\widetilde{\mathbf{E}}_r)_{i,j} \left( (\widehat{\mathbf{C}}_r)_{pos} \right)_{i,j} (\widehat{\mathbf{D}}_r)_{i,j}$$
$$- 2 \sum (\widetilde{\mathbf{E}}_r)_{i,j} (\mathbf{G}_r \Phi_r^\top \mathbf{1}_{n_r} \mathbf{1}_k^\top)_{i,j} (\widehat{\mathbf{D}}_r)_{i,j} \tag{9}$$
$$+ \sum (\mathbf{G}_r \Phi_r \mathbf{G}_r^\top \widetilde{\mathbf{E}}_r \mathbf{1}_k \mathbf{1}_k^\top)_{i,j} (\widehat{\mathbf{H}}_r)_{i,j}$$
$$+ \frac{\beta b_r}{2} \sum \left( (\widehat{\mathbf{C}}_r)_{neg} \right)_{i,j} (\widehat{\mathbf{J}}_r)_{i,j},$$

*where* $(\widehat{\mathbf{D}}_r)_{i,j} = 1 + \log\left( (\mathbf{E}_r)_{i,j}/(\widetilde{\mathbf{E}}_r)_{i,j} \right)$, $(\widehat{\mathbf{H}}_r)_{i,j} = (\mathbf{E}_r)_{i,j}^2/(\widetilde{\mathbf{E}}_r)_{i,j}$, $(\widehat{\mathbf{J}}_r)_{i,j} = \left( (\widetilde{\mathbf{E}}_r)_{i,j}^2 + (\mathbf{E}_r)_{i,j}^2 \right)/(\widetilde{\mathbf{E}}_r)_{i,j}$.

On the basis of **Theorem** 1, we further have the following theorem holds,

**Theorem 2.** *Under the rule Eq. (8), the loss Eq. (6) is monotonically decreasing.*

▶ When updating $\mathbf{C}$, the loss can be simplified as

$$\min_{\mathbf{C}} \sum_{r=1}^{v} a_r^2 \left\| \mathbf{X}_r \mathbf{G}_r - \widehat{\mathbf{A}}_r \mathbf{E}_r^\top \mathbf{G}_r \right\|_F^2 + \lambda \operatorname{Tr}\left( \mathbf{C}^\top \mathbf{P}_r^\top \mathbf{C}_r \right)$$
$$\text{s.t. } \mathbf{C} \geq 0, \mathbf{C}^\top \mathbf{P}_r^\top \mathbf{1}_{d_r} = \mathbf{1}_k. \tag{10}$$

Then, we can get the following updating rule,

$$(\mathbf{C})_{i,j} \leftarrow (\mathbf{C})_{i,j} \left[ \frac{\left( \sum_{r=1}^{v} a_r^2 \mathbf{P}_r^\top \mathbf{X}_r \widehat{\mathbf{K}}_r \right)_{i,j}}{\left( \sum_{r=1}^{v} a_r^2 \mathbf{P}_r^\top \widehat{\mathbf{A}}_r \mathbf{E}_r^\top \widehat{\mathbf{K}}_r + \frac{\lambda}{2} \mathbf{M} \right)_{i,j}} \right]^{\frac{1}{2}} \tag{11}$$

where $\widehat{\mathbf{K}}_r = \mathbf{G}_r \mathbf{G}_r^\top \mathbf{E}_r \mathbf{D}_\gamma^\top$, $\mathbf{D}_\gamma = \mathbf{D}_{1:k,:}$, $\mathbf{M} = \sum_{r=1}^{v} \mathbf{P}_r^\top \mathbf{C}_r$. Further, we have the following theorem,

**Theorem 3.** *With column-normalized* $\mathbf{P}_r$, $\mathbf{P}_r \mathbf{C}$ *satisfies column normalization only if* $\mathbf{C}$ *is also column-normalized.*

▶ When updating $\mathbf{C}_r$, the loss is transformed as

$$\min_{\mathbf{C}_r} a_r^2 \left\| \mathbf{X}_r \mathbf{G}_r - \widehat{\mathbf{A}}_r \mathbf{E}_r^\top \mathbf{G}_r \right\|_F^2 + \lambda \operatorname{Tr}\left( \mathbf{C}^\top \mathbf{P}_r^\top \mathbf{C}_r \right)$$
$$\text{s.t. } \mathbf{C}_r \geq 0, \mathbf{C}_r^\top \mathbf{1}_{d_r} = \mathbf{1}_k. \tag{12}$$

We have the following updating rule,

$$(\mathbf{C}_r)_{i,j} \leftarrow (\mathbf{C}_r)_{i,j} \left[ \frac{\left( \mathbf{X}_r \widehat{\mathbf{B}}_r \mathbf{E}_r \mathbf{D}_\psi^\top \right)_{i,j}}{\left( \widehat{\mathbf{A}}_r \mathbf{E}_r^\top \widehat{\mathbf{B}}_r \mathbf{E}_r \mathbf{D}_\psi^\top + \frac{\lambda}{2} \mathbf{P}_r \mathbf{C} \right)_{i,j}} \right]^{\frac{1}{2}} \tag{13}$$

where $\mathbf{D}_\varphi = \mathbf{D}_{k+1:2k,:}$.

▶ When updating $\mathbf{D}$, the loss can be formulated as

$$\min_{\mathbf{D}} \sum_{r=1}^{v} a_r^2 \left\| \mathbf{X}_r \mathbf{G}_r - [\mathbf{P}_r \mathbf{C} | \mathbf{C}_r] \mathbf{D} \mathbf{E}_r^\top \mathbf{G}_r \right\|_F^2 \tag{14}$$
$$\text{s.t. } \mathbf{D} \geq 0.$$

We can obtain the following updating rule for $\mathbf{D}$,

$$\mathbf{D}_{i,j} \leftarrow \mathbf{D}_{i,j} \left[ \frac{\left( \sum_{r=1}^{v} a_r^2 \widehat{\mathbf{L}}_r^\top \mathbf{X}_r \mathbf{G}_r \mathbf{G}_r^\top \mathbf{E}_r \right)_{i,j}}{\left( \sum_{r=1}^{v} a_r^2 \widehat{\mathbf{L}}_r^\top \widehat{\mathbf{L}}_r \mathbf{D} \mathbf{E}_r^\top \mathbf{G}_r \mathbf{G}_r^\top \mathbf{E}_r \right)_{i,j}} \right]^{\frac{1}{2}} \tag{15}$$

where $\widehat{\mathbf{L}}_r = [\mathbf{P}_r \mathbf{C} | \mathbf{C}_r]$.

▶ When updating $\mathbf{P}_r$, the loss can be formulated as

$$\min_{\mathbf{P}_r} a_r^2 \left\| \mathbf{X}_r \mathbf{G}_r - \widehat{\mathbf{A}}_r \mathbf{E}_r^\top \mathbf{G}_r \right\|_F^2 + \lambda \operatorname{Tr}\left( \mathbf{C}^\top \mathbf{P}_r^\top \mathbf{C}_r \right)$$
$$\text{s.t. } \mathbf{P}_r \geq 0, \mathbf{C}^\top \mathbf{P}_r^\top \mathbf{1}_{d_r} = \mathbf{1}_k. \tag{16}$$

We have the following updating rule,

$$(\mathbf{P}_r)_{i,j} \leftarrow (\mathbf{P}_r)_{i,j} \left[ \frac{\left( a_r^2 \mathbf{X}_r \widehat{\mathbf{K}}_r \mathbf{C}^\top \right)_{i,j}}{\left( a_r^2 \widehat{\mathbf{A}}_r \mathbf{E}_r^\top \widehat{\mathbf{K}}_r \mathbf{C}^\top + \frac{\lambda}{2} \mathbf{C}_r \mathbf{C}^\top \right)_{i,j}} \right]^{\frac{1}{2}} \tag{17}$$

▶ When updating $\mathbf{E}$, the loss equivalently becomes

$$\min_{\mathbf{E}} - \operatorname{Tr}\left( \mathbf{E}^\top \sum_{r=1}^{v} b_r \mathbf{E}_r \mathbf{F}_r \right) \text{ s.t. } \mathbf{E}^\top \mathbf{E} = \mathbf{I}_k. \tag{18}$$

Its optimal solution can be obtained by setting $\mathbf{E}$ as the product of $\mathbf{U}$ and $\mathbf{V}^\top$ where $\mathbf{U}$ and $\mathbf{V}^\top$ are the singular matrices of the term $\sum_{r=1}^{v} b_r \mathbf{E}_r \mathbf{F}_r$.

▶ When updating $\mathbf{F}_r$, the loss equivalently becomes

$$\min_{\mathbf{F}_r} -\operatorname{Tr}\left(\mathbf{E}^\top \sum_{r=1}^{v} b_r \mathbf{E}_r \mathbf{F}_r\right) \text{ s.t. } \mathbf{F}_r \mathbf{F}_r^\top = \mathbf{I}_k. \quad (19)$$

The optimal solution is $\mathbf{U}\mathbf{V}^\top$. $\mathbf{U}$ and $\mathbf{V}$ are the singular matrices of $\mathbf{E}_r^\top \mathbf{E}$.

▶ When updating $a_r$, the loss can be simplified as

$$\min_{a_r} \sum_{r=1}^{v} a_r^2 \left\|\mathbf{X}_r \mathbf{G}_r - \widehat{\mathbf{A}}_r \mathbf{E}_r^\top \mathbf{G}_r\right\|_F^2 \text{ s.t. } \sum_{r=1}^{v} a_r = 1, a_r \geq 0. \quad (20)$$

The optimal solution is

$$a_r = \frac{\frac{1}{\left\|\mathbf{X}_r \mathbf{G}_r - \widehat{\mathbf{A}}_r \mathbf{E}_r^\top \mathbf{G}_r\right\|_F^2}}{\sum_{r=1}^{v} \frac{1}{\left\|\mathbf{X}_r \mathbf{G}_r - \widehat{\mathbf{A}}_r \mathbf{E}_r^\top \mathbf{G}_r\right\|_F^2}}. \quad (21)$$

**Remark 1.** *The value of $\|\mathbf{X}_r \mathbf{G}_r - \widehat{\mathbf{A}}_r \mathbf{E}_r^\top \mathbf{G}_r\|_F^2$ equals to that of $\|\mathbf{X}_r \mathbf{G}_r \mathbf{G}_r^\top - \widehat{\mathbf{A}}_r \mathbf{E}_r^\top \mathbf{G}_r \mathbf{G}_r^\top\|_F^2$. The former needs at least $\mathcal{O}(nn_r)$ computing overhead while the latter can be calculated within $\mathcal{O}(n)$ overhead.*

▶ When updating $b_r$, the loss can be simplified as

$$\min_{b_r} -\operatorname{Tr}\left(\mathbf{E}^\top \sum_{r=1}^{v} b_r \mathbf{E}_r \mathbf{F}_r\right) \text{ s.t. } \sum_{r=1}^{v} b_r^2 = 1, b_r \geq 0. \quad (22)$$

The optimal solution is

$$b_r = \frac{\widehat{M_r}}{\sqrt{\sum_{r=1}^{v} \left(\widehat{M_r}\right)^2}}. \quad (23)$$

where $\widehat{M_r} = \operatorname{Tr}\left(\mathbf{E}^\top \mathbf{E}_r \mathbf{F}_r\right)$.

**Algorithm** 1 summarizes the overall procedure of proposed BACDL where $g^h$ denotes the loss value at the $h$-th iteration.

For the lower bound, we have the following theorem holds,

**Theorem 4.** *The loss value is lower-bounded by $-\beta\sqrt{vnk}$.*

On the basis of **Theorem** 2 and **Theorem** 4, we further have

**Theorem 5.** *The proposed BACDL algorithm is convergent.*

About the complexity of BACDL, we have

**Theorem 6.** *BACDL is with the time and space overhead linear to the sample size $n$.*

**Remark 2** *In virtue of the overall linear overhead, BACDL is scalable to large-scale scenarios.*

---

**Algorithm 1** Proposed BACDL Algorithm

**Input:** Data matrix $\mathbf{X}_r$, indicator vector $\mathbf{w}_r$, hyper-parameters $\lambda$ and $\beta$;

1: Constructing index matrix $\mathbf{G}_r$;
2: Doing distribution normalization on $\mathbf{X}_r \mathbf{G}_r$;
3: **while** $(g^h - g^{h+1})/g^h \geq 1e-3$ **do**
4:     Updating the feature clusters $\mathbf{E}_r$ by (8);
5:     Doing row-normalization on $\mathbf{E}_r$;
6:     Updating the perspective-shared matrix $\mathbf{C}$ by (11);
7:     Doing column-normalization on $\mathbf{C}$;
8:     Updating the perspective-specific matrix $\mathbf{C}_r$ by (13);
9:     Doing column-normalization on $\mathbf{C}_r$;
10:     Updating the association $\mathbf{D}$ by (15);
11:     Updating the view guidance $\mathbf{P}_r$ by (17);
12:     Doing column-normalization on $\mathbf{P}_r$;
13:     Updating the common sample clusters $\mathbf{E}$ by (18);
14:     Updating the space rotation matrix $\mathbf{F}_r$ by (19);
15:     Updating the view weight $a_r$ by (21);
16:     Updating the sample cluster weight $b_r$ by (23);
17:     $h = h + 1$;
18: **end while**
**Output:** Spectral clustering on $\mathbf{E}$;

---

*Table 1.* Details of Multi-view Datasets Used in Experiments

| Dataset | Samples | Views | Clusters | Dimensions |
|---------|---------|-------|----------|------------|
| FLOEVEN | 1360 | 7 | 17 | 5376/1239/512/5376/5376/5376/5376 |
| SYNTHREED | 600 | 3 | 3 | 3/2003/3 |
| DEOLOG | 358 | 2 | 6 | 22/12 |
| YALTHREE | 165 | 3 | 15 | 3304/4096/6750 |
| BGFEA | 2500 | 3 | 5 | 1000/500/250 |
| AWTEN | 5814 | 6 | 10 | 2000/2688/2000/2000/252/2000 |
| HDIGTWO | 10000 | 2 | 10 | 256/784 |
| YOUFOURV | 38654 | 4 | 10 | 512/944/576/640 |

## 6. Experiments and Analysis

### 6.1. Benchmark and Baseline

The following IMC algorithms are utilized as the baselines: Sparse Consensus Affinity Construction (**LRTL** (Chen et al., 2023)), Interview Graph Connectivity Learning (**TCIMC** (Xia et al., 2022)), Between-view Inferring and Within-view Preservation (**AGCIM** (Wen et al., 2021a)), Local Graph Embedding Generation and Unified Representation Learning (**LSIMV** (Liu et al., 2023b)), Local Geometric Similarity Conservation (**GIMC** (Wen et al., 2021b)), Individual Consensus Structure Set Exploration (**IMVCI** (Tang et al., 2024)), Projective Graph Regularization Balance Learning (**PIMVC** (Deng et al., 2024)), Similar Several-Neighbor Confidence Representation Generation (**HCCGL** (Wen et al., 2023a)), Multiple Spectral Connection Relationship Fusion (**USETL** (Chen et al., 2024)), Multi-matrix-factorization Geometry-preserving Learning (**LBIMV** (Wen et al., 2023b)), Feature Inferring and Cross-view Correlation Guidance (**UIMC** (Lin et al., 2024)).

*Table 2.* Clustering Results of Different IMC Algorithms

| DATASET | FLOEVEN | | | | | | | | | SYNTHREED | | | | | | | | |
|---|---|---|---|---|---|---|---|---|---|---|---|---|---|---|---|---|---|---|
| RATIO | 0.2 | | | 0.5 | | | 0.8 | | | 0.2 | | | 0.5 | | | 0.8 | | |
| METRIC | PUR | ACC | FSC | PUR | ACC | FSC | PUR | ACC | FSC | PUR | ACC | FSC | PUR | ACC | FSC | PUR | ACC | FSC |
| LRTL | 11.08 | 9.27 | 7.85 | 10.83 | 10.85 | 7.60 | 11.38 | 9.41 | 7.21 | 39.87 | 40.08 | 31.14 | 38.67 | 40.67 | 31.47 | 35.83 | 39.83 | 31.88 |
| TCIMC | 10.93 | 10.35 | 8.21 | 10.63 | 9.63 | 6.87 | 9.88 | 9.48 | 5.69 | 37.57 | 38.92 | 30.76 | 37.53 | 35.72 | 31.23 | 36.83 | 36.23 | 29.87 |
| AGCIM | 11.02 | 10.07 | 8.43 | 9.34 | 8.50 | 7.22 | 10.67 | 10.32 | 7.19 | 33.50 | 33.50 | 27.79 | 34.00 | 34.00 | 28.75 | 34.17 | 34.17 | 29.15 |
| LSIMV | 8.74 | 9.47 | 5.17 | 8.21 | 9.67 | 6.44 | 7.88 | 8.43 | 7.38 | 37.74 | 39.37 | 32.21 | 36.14 | 36.73 | 35.13 | 34.78 | 33.89 | 31.33 |
| GIMC | 8.32 | 9.38 | 5.97 | 8.63 | 7.94 | 6.34 | 7.26 | 7.94 | 5.25 | 36.89 | 35.72 | 32.67 | 35.89 | 33.73 | 34.01 | 33.84 | 32.75 | 33.84 |
| IMVCI | 10.37 | 10.86 | 6.40 | 10.95 | 10.21 | 6.51 | 11.35 | 10.17 | 5.08 | 40.38 | 40.38 | 32.80 | 39.77 | 40.68 | 32.29 | 38.93 | 33.53 | 30.41 |
| PIMVC | 11.11 | 9.28 | 4.32 | 11.23 | 9.36 | 7.32 | 10.78 | 8.56 | 6.35 | 40.27 | 40.67 | 30.89 | 39.50 | 39.50 | 31.04 | 37.02 | 35.02 | 30.82 |
| HCCGL | 7.13 | 6.68 | 5.38 | 7.06 | 6.91 | 5.64 | 7.28 | 7.06 | 5.31 | 40.45 | 39.95 | 32.19 | 39.67 | 39.67 | 34.51 | 37.01 | 37.01 | 31.46 |
| USETL | 9.99 | 9.43 | 7.22 | 10.19 | 9.56 | 7.67 | 9.32 | 8.90 | 7.25 | 38.96 | 38.96 | 28.86 | 36.15 | 35.51 | 29.56 | 36.28 | 36.28 | 29.08 |
| LBIMV | 10.56 | 10.31 | 5.09 | 11.07 | 9.85 | 5.18 | 10.01 | 9.06 | 5.23 | 40.27 | 40.34 | 27.45 | 40.33 | 40.07 | 30.16 | 35.67 | 35.67 | 27.04 |
| UIMC | 10.88 | 9.22 | 5.80 | 11.12 | 9.96 | 5.99 | 10.51 | 9.22 | 5.83 | 39.33 | 38.33 | 32.13 | 36.17 | 40.17 | 33.26 | 38.83 | 37.50 | 33.86 |
| OURS | 11.17 | 10.80 | 5.88 | 11.25 | 10.53 | 5.88 | 11.54 | 10.74 | 5.88 | 42.50 | 42.12 | 35.21 | 42.83 | 42.83 | 35.24 | 41.04 | 41.04 | 34.59 |

| DATASET | DEOLOG | | | | | | | | | YALTHREE | | | | | | | | |
|---|---|---|---|---|---|---|---|---|---|---|---|---|---|---|---|---|---|---|
| METRIC | PUR | ACC | FSC | PUR | ACC | FSC | PUR | ACC | FSC | PUR | ACC | FSC | PUR | ACC | FSC | PUR | ACC | FSC |
| LRTL | 31.43 | 21.59 | 15.29 | 30.71 | 19.87 | 16.54 | 31.37 | 20.90 | 15.93 | 21.02 | 21.52 | 8.59 | 18.88 | 20.17 | 6.35 | 19.39 | 18.85 | 6.61 |
| TCIMC | 30.86 | 19.13 | 16.43 | 30.73 | 19.26 | 15.74 | 33.47 | 20.32 | 15.88 | 21.35 | 17.98 | 7.79 | 21.51 | 17.87 | 6.98 | 20.21 | 17.79 | 5.89 |
| AGCIM | 30.45 | 21.45 | 18.64 | 29.55 | 22.44 | 17.88 | 32.01 | 22.89 | 17.32 | 22.12 | 18.12 | 7.88 | 18.18 | 17.97 | 5.87 | 18.48 | 17.67 | 6.42 |
| LSIMV | 31.26 | 18.43 | 14.37 | 31.69 | 18.57 | 15.32 | 32.94 | 18.74 | 14.57 | 18.84 | 16.42 | 6.73 | 18.96 | 16.52 | 6.26 | 18.34 | 16.43 | 7.46 |
| GIMC | 31.64 | 19.32 | 17.13 | 32.64 | 17.36 | 15.63 | 31.53 | 18.57 | 16.43 | 19.46 | 16.72 | 6.84 | 17.83 | 17.96 | 6.72 | 19.43 | 17.43 | 8.82 |
| IMVCI | 30.98 | 21.58 | 17.43 | 30.93 | 21.93 | 17.98 | 30.82 | 24.82 | 18.23 | 21.15 | 21.74 | 7.12 | 19.15 | 20.55 | 7.42 | 20.36 | 20.17 | 6.78 |
| PIMVC | 31.33 | 22.04 | 17.32 | 29.36 | 20.24 | 17.54 | 31.62 | 23.03 | 18.97 | 20.64 | 17.24 | 7.34 | 18.12 | 15.52 | 6.26 | 18.36 | 15.42 | 7.56 |
| HCCGL | 29.05 | 20.05 | 17.57 | 30.58 | 19.72 | 17.16 | 29.32 | 24.20 | 17.77 | 21.91 | 21.71 | 7.13 | 21.15 | 20.18 | 7.16 | 20.61 | 20.36 | 9.23 |
| USETL | 30.84 | 18.99 | 18.13 | 28.19 | 22.61 | 17.98 | 30.94 | 22.22 | 17.98 | 19.94 | 21.85 | 8.21 | 17.33 | 19.85 | 6.11 | 20.85 | 19.88 | 7.43 |
| LBIMV | 30.09 | 20.09 | 17.71 | 30.93 | 22.93 | 18.07 | 29.27 | 21.42 | 17.07 | 22.94 | 19.12 | 6.29 | 20.94 | 19.91 | 6.18 | 21.15 | 19.73 | 5.51 |
| UIMC | 30.83 | 19.89 | 16.23 | 31.40 | 20.82 | 16.57 | 31.01 | 22.07 | 16.17 | 20.03 | 19.82 | 6.39 | 18.18 | 17.58 | 4.86 | 21.64 | 17.82 | 7.30 |
| OURS | 31.84 | 22.32 | 18.45 | 31.84 | 23.74 | 18.41 | 31.28 | 24.25 | 19.56 | 24.94 | 23.55 | 7.89 | 21.73 | 20.61 | 6.30 | 21.55 | 20.70 | 6.70 |

| DATASET | BGFEA | | | | | | | | | AWTEN | | | | | | | | |
|---|---|---|---|---|---|---|---|---|---|---|---|---|---|---|---|---|---|---|
| METRIC | PUR | ACC | FSC | PUR | ACC | FSC | PUR | ACC | FSC | PUR | ACC | FSC | PUR | ACC | FSC | PUR | ACC | FSC |
| LRTL | 17.37 | 20.21 | 18.16 | 18.27 | 20.62 | 19.01 | 18.22 | 19.22 | 18.07 | 17.13 | 9.67 | 8.43 | 18.03 | 11.28 | 8.42 | 19.53 | 9.40 | 8.38 |
| TCIMC | 19.63 | 20.46 | 18.26 | 19.43 | 20.88 | 18.33 | 20.16 | 19.69 | 17.93 | 16.78 | 10.71 | 10.63 | 16.67 | 11.48 | 9.37 | 17.84 | 9.28 | 9.13 |
| AGCIM | 20.56 | 20.43 | 16.32 | 21.28 | 20.20 | 16.43 | 21.36 | 20.32 | 15.33 | 13.56 | 10.28 | 10.66 | 14.59 | 10.32 | 9.47 | 13.47 | 10.73 | 9.89 |
| LSIMV | 19.73 | 18.24 | 20.11 | 18.03 | 18.32 | 18.26 | 18.48 | 18.21 | 18.11 | 19.47 | 10.37 | 9.72 | 17.39 | 10.58 | 9.56 | 17.69 | 10.57 | 8.92 |
| GIMC | 21.26 | 19.26 | 19.42 | 18.29 | 17.97 | 19.17 | 18.97 | 18.57 | 18.32 | 20.47 | 10.98 | 9.83 | 20.89 | 10.24 | 9.46 | 18.36 | 9.43 | 8.74 |
| IMVCI | 19.58 | 20.48 | 16.32 | 21.02 | 20.02 | 16.73 | 18.42 | 21.92 | 17.32 | 20.21 | 9.69 | 11.54 | 20.29 | 8.69 | 9.86 | 19.55 | 8.07 | 9.35 |
| PIMVC | 22.58 | 20.52 | 16.28 | 20.66 | 20.87 | 17.25 | 20.76 | 20.57 | 15.73 | 20.82 | 10.64 | 10.85 | 20.44 | 9.87 | 10.24 | 19.87 | 9.19 | 9.27 |
| HCCGL | 20.68 | 18.32 | 18.46 | 21.32 | 19.32 | 19.32 | 20.60 | 17.36 | 18.43 | 20.26 | 10.18 | 7.83 | 20.21 | 9.64 | 8.23 | 20.13 | 9.67 | 7.38 |
| USETL | 21.32 | 18.01 | 16.93 | 19.75 | 18.56 | 16.89 | 20.21 | 18.08 | 15.62 | 20.16 | 7.31 | 8.89 | 20.07 | 8.26 | 9.87 | 20.01 | 8.88 | 10.21 |
| LBIMV | 21.08 | 20.11 | 18.21 | 22.18 | 20.73 | 16.16 | 21.20 | 20.52 | 16.11 | 20.43 | 8.95 | 8.03 | 20.12 | 9.42 | 8.59 | 20.02 | 9.23 | 8.69 |
| UIMC | 21.53 | 18.69 | 18.75 | 20.78 | 19.46 | 19.33 | 19.74 | 17.84 | 17.82 | 19.32 | 10.25 | 9.43 | 19.85 | 10.49 | 8.78 | 19.56 | 10.23 | 9.64 |
| OURS | 22.08 | 22.08 | 20.08 | 22.64 | 22.40 | 20.16 | 22.48 | 22.28 | 20.20 | 20.74 | 12.24 | 11.01 | 20.17 | 12.01 | 10.97 | 20.18 | 12.06 | 10.94 |

| DATASET | HDIGTWO | | | | | | | | | YOUFOURV | | | | | | | | |
|---|---|---|---|---|---|---|---|---|---|---|---|---|---|---|---|---|---|---|
| METRIC | PUR | ACC | FSC | PUR | ACC | FSC | PUR | ACC | FSC | PUR | ACC | FSC | PUR | ACC | FSC | PUR | ACC | FSC |
| LRTL | 11.62 | 10.02 | 8.43 | 11.62 | 10.61 | 8.92 | 9.78 | 10.48 | 8.87 | - | - | - | - | - | - | - | - | - |
| TCIMC | 11.48 | 10.62 | 9.42 | 10.32 | 10.63 | 8.62 | 9.57 | 9.32 | 7.63 | - | - | - | - | - | - | - | - | - |
| AGCIM | 11.82 | 10.28 | 7.88 | 11.59 | 10.43 | 9.39 | 10.68 | 10.49 | 8.13 | - | - | - | - | - | - | - | - | - |
| LSIMV | 10.73 | 9.36 | 8.43 | 12.67 | 8.93 | 8.56 | 9.63 | 9.17 | 7.53 | - | - | - | - | - | - | - | - | - |
| GIMC | 10.85 | 8.23 | 7.84 | 11.32 | 8.42 | 8.32 | 9.32 | 9.63 | 6.89 | 12.56 | 9.45 | 9.28 | 11.36 | 8.99 | 8.66 | 11.32 | 8.79 | 8.76 |
| IMVCI | 10.83 | 10.35 | 8.50 | 11.51 | 11.05 | 9.36 | 11.18 | 10.65 | 9.09 | - | - | - | - | - | - | - | - | - |
| PIMVC | 10.93 | 8.44 | 8.69 | 11.73 | 9.15 | 9.28 | 10.73 | 8.94 | 7.43 | 15.67 | 9.77 | 8.87 | 14.12 | 9.79 | 9.33 | 15.36 | 9.94 | 8.62 |
| HCCGL | 10.42 | 11.27 | 8.44 | 10.64 | 11.16 | 8.52 | 8.77 | 10.32 | 8.11 | - | - | - | - | - | - | - | - | - |
| USETL | 10.28 | 9.22 | 8.52 | 9.74 | 9.73 | 7.98 | 8.37 | 9.92 | 7.09 | 11.32 | 8.94 | 9.68 | 10.43 | 9.35 | 9.42 | 10.42 | 8.75 | 8.47 |
| LBIMV | 9.77 | 10.71 | 8.10 | 9.55 | 9.35 | 9.26 | 9.16 | 10.93 | 9.54 | 15.32 | 10.54 | 8.99 | 16.03 | 10.36 | 9.16 | 14.32 | 10.21 | 9.14 |
| UIMC | 10.74 | 11.32 | 8.49 | 9.73 | 10.56 | 9.23 | 8.42 | 10.42 | 8.52 | - | - | - | - | - | - | - | - | - |
| OURS | 11.73 | 11.51 | 10.03 | 11.75 | 11.60 | 10.11 | 11.79 | 11.70 | 10.22 | 15.99 | 10.91 | 10.48 | 15.95 | 10.85 | 10.47 | 15.95 | 10.80 | 10.47 |

All algorithms are evaluated on the following eight multi-view benchmark datasets: FLOEVEN, SYNTHREED, DE-OLOG, YALTHREE, BGFEA, AWTEN, HDIGTWO, and YOUFOURV. Table 1 presents their details.

### 6.2. Results and Discussions

We test the clustering performance under diverse missing ratios, i.e., 0.2, 0.5 and 0.8 respectively. The results are reported in Table 2. We can draw that
(1) Our BACDL makes more favorable results than multiple competitors. For example, on SYNTHREED, we receive the most desirable results; On BGFEA, HDIGTWO and YOUFOURV, we are consistently in Top-2; On DEOLOG and AWTEN, there are only two sub-optimal results totally. This suggests that BACDL is effective to handle IMC issues.

(2) Algorithms LRTL, TCIMC, AGCIM, LSIMV, IMVCI, HCCGL and UIMC are unable to deal with the large-scale dataset YOUFOURV due to the complexity limitation. Orthogonal to them, not only do we normally work but provide competitive results, which illustrates that the proposed BACDL is more widely applicable.

(3) There are some inferior results on FLOEVEN and YALTHREE, possibly because we generate the cluster la-

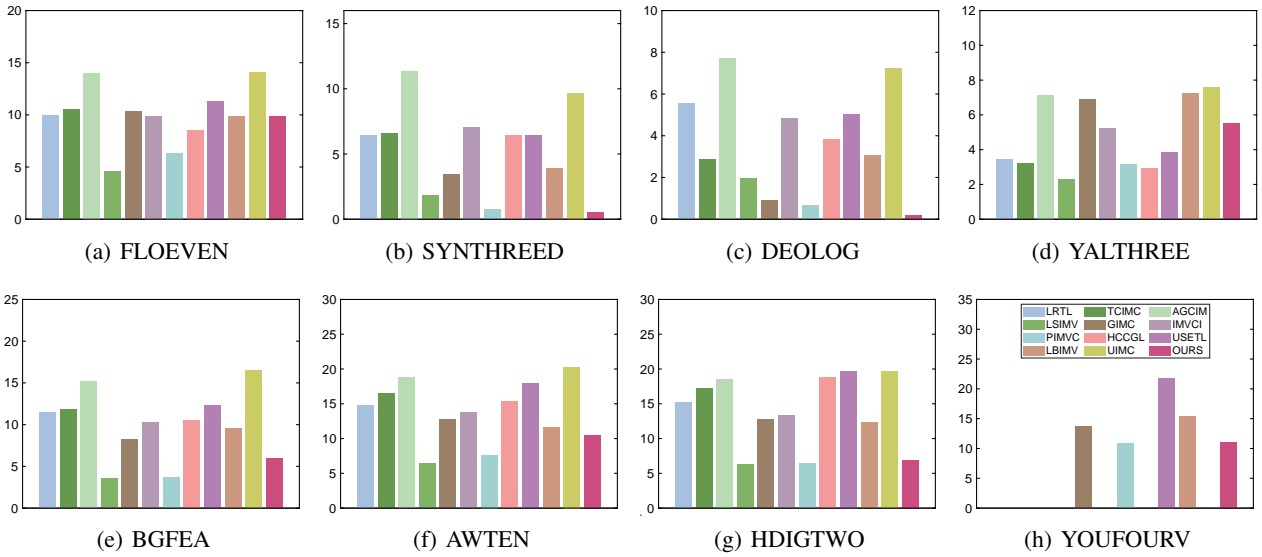

*Figure 2.* Running Time of Different IMC Algorithms

*Table 3.* Memory Overhead (GB) of Different IMC Algorithms

| Method | DAT1 | DAT2 | DAT3 | DAT4 | DAT5 | DAT6 | DAT7 | DAT8 |
|---|---|---|---|---|---|---|---|---|
| LRTL | 1.61 | 0.12 | 0.05 | 0.07 | 1.85 | 19.49 | 19.59 | - |
| TCIMC | 2.56 | 0.23 | 0.04 | 0.11 | 3.45 | 22.73 | 37.70 | - |
| AGCIM | 0.77 | 0.06 | 0.08 | 0.06 | 0.84 | 7.13 | 14.65 | - |
| LSIMV | 1.60 | 0.06 | 0.05 | 0.08 | 0.48 | 4.65 | 4.39 | - |
| GIMC | 1.22 | 0.09 | 0.03 | 0.41 | 0.50 | 4.63 | 5.47 | 98.86 |
| IMVCI | 0.99 | 0.07 | 0.03 | 0.07 | 0.78 | 5.96 | 11.41 | - |
| PIMVC | 3.89 | 0.05 | 0.05 | 3.66 | 0.45 | 4.14 | 4.77 | 80.51 |
| HCCGL | 1.32 | 0.10 | 0.05 | 0.07 | 1.66 | 13.02 | 21.46 | - |
| USETL | 0.72 | 0.04 | 0.08 | 0.07 | 1.05 | 5.85 | 15.27 | 84.66 |
| LBIMV | 0.81 | 0.04 | 0.03 | 0.37 | 0.35 | 3.33 | 3.94 | 82.68 |
| UIMC | 3.81 | 0.13 | 0.04 | 0.15 | 2.68 | 28.29 | 28.34 | - |
| OURS | 1.07 | 0.03 | 0.04 | 0.11 | 0.13 | 1.82 | 0.35 | 3.37 |

*Table 4.* Ablation for Perspective-shared Determinants

| A/B | 0.2 | | | 0.5 | | | 0.8 | | |
|---|---|---|---|---|---|---|---|---|---|
| | PUR | ACC | FSC | PUR | ACC | FSC | PUR | ACC | FSC |
| **FLOEVEN** | | | | | | | | | |
| OPDD | 9.86 | 9.53 | 4.52 | 9.57 | 9.36 | 4.46 | 9.81 | 9.47 | 4.62 |
| WPDD | **11.17** | **10.80** | **5.89** | **11.25** | **10.53** | **5.88** | **11.54** | **10.74** | **5.83** |
| **SYNTHREED** | | | | | | | | | |
| OPDD | 36.14 | 35.84 | 33.21 | 37.17 | 37.17 | 33.39 | 37.25 | 37.25 | 33.43 |
| WPDD | **42.50** | **42.12** | **35.21** | **42.83** | **42.83** | **35.24** | **41.04** | **41.04** | **34.59** |
| **DEOLOG** | | | | | | | | | |
| OPDD | 29.53 | 20.04 | 16.13 | 29.02 | 20.41 | 16.34 | 29.02 | 21.64 | 17.58 |
| WPDD | **31.84** | **22.32** | **18.45** | **31.84** | **23.74** | **18.41** | **31.28** | **24.25** | **19.56** |
| **YALTHREE** | | | | | | | | | |
| OPDD | 21.27 | 21.27 | 6.93 | 21.36 | 18.97 | 5.87 | 19.03 | 19.26 | 6.06 |
| WPDD | **24.94** | **23.55** | **7.89** | **21.73** | **20.61** | **6.30** | **21.55** | **20.70** | **6.70** |
| **BGFEA** | | | | | | | | | |
| OPDD | 19.58 | 19.22 | 17.57 | 18.90 | 18.74 | 17.42 | 18.90 | 18.74 | 17.42 |
| WPDD | **22.08** | **22.08** | **20.08** | **22.64** | **22.40** | **20.16** | **22.48** | **22.28** | **20.20** |
| **AWTEN** | | | | | | | | | |
| OPDD | 18.42 | 10.07 | 9.25 | 18.42 | 10.11 | 9.25 | 18.42 | 10.32 | 9.30 |
| WPDD | **20.74** | **12.24** | **11.01** | **20.17** | **12.01** | **10.97** | **20.18** | **12.06** | **10.94** |
| **HDIGTWO** | | | | | | | | | |
| OPDD | 9.35 | 9.29 | 8.07 | 9.56 | 9.31 | 8.10 | 9.32 | 9.18 | 8.08 |
| WPDD | **11.73** | **11.51** | **10.03** | **11.75** | **11.60** | **10.11** | **11.79** | **11.70** | **10.22** |
| **YOUFOURV** | | | | | | | | | |
| OPDD | 13.48 | 8.30 | 7.99 | 13.47 | 8.33 | 8.01 | 13.51 | 8.27 | 8.01 |
| WPDD | **15.99** | **10.91** | **10.48** | **15.95** | **10.85** | **10.47** | **15.95** | **10.80** | **10.47** |

bels from formed spectrum rather than directly from original data. This could, to a certain extent, degenerate the diversity of view data, weakening the clustering performance.

### 6.3. Time and Space Overhead

*Time Overhead:* To validate the efficiency, we record the running time of each algorithm, as suggested in Fig. 2 (The $y$-axis denotes $\log_2(\cdot)$ seconds plus a constant.) According to this figure, one can conclude that

(1) Our BACDL requires relatively lower time overhead than multiple competitors. For instance, it takes the least time on SYNTHREED, DEOLOG, HDIGTWO and YOUFOURV. On FLOEVEN, YALTHREE, BGFEA and AWTEN, its running speed is still comparable. This demonstrates that the proposed BACDL is computationally-efficient.

(2) LSIMV, PIMVC and HCCGL run faster than BACDL in some cases, potentially because LSIMV introduces sparse

factors and integrates low-dimensional embedding into local graphs to form uniform representation, PIMVC utilizes a group of projections to leverage view diversities in a consensus low-dimensional manifold space, HCCGL learns only one affinity across views using a small-sized confidence graph and refines structures directly from original similarity. In spite of the time-saving benefits, they typically do

*Table 5.* Ablation for Perspective-specific Determinants

| A/B | 0.2 | | | 0.5 | | | 0.8 | | |
|---|---|---|---|---|---|---|---|---|---|
| | PUR | ACC | FSC | PUR | ACC | FSC | PUR | ACC | FSC |
| **FLOEVEN** | | | | | | | | | |
| OPCD | 9.43 | 9.07 | 4.32 | 9.80 | 9.30 | 4.43 | 9.26 | 9.07 | 4.42 |
| WPCD | **11.17** | **10.80** | **5.89** | **11.25** | **10.53** | **5.88** | **11.54** | **10.74** | **5.83** |
| **SYNTHREED** | | | | | | | | | |
| OPCD | 35.75 | 35.71 | 32.57 | 37.50 | 37.42 | 33.27 | 36.77 | 36.57 | 32.49 |
| WPCD | **42.50** | **42.12** | **35.21** | **42.83** | **42.83** | **35.24** | **41.04** | **41.04** | **34.59** |
| **DEOLOG** | | | | | | | | | |
| OPCD | 28.93 | 18.16 | 15.07 | 28.08 | 18.09 | 14.85 | 28.59 | 17.89 | 15.54 |
| WPCD | **31.84** | **22.32** | **18.45** | **31.84** | **23.74** | **18.41** | **31.28** | **24.25** | **19.56** |
| **YALTHREE** | | | | | | | | | |
| OPCD | 20.76 | 18.13 | 4.48 | 20.64 | 18.04 | 4.41 | 19.49 | 18.13 | 5.46 |
| WPCD | **24.94** | **23.55** | **7.89** | **21.73** | **20.61** | **6.30** | **21.55** | **20.70** | **6.70** |
| **BGFEA** | | | | | | | | | |
| OPCD | 20.12 | 19.54 | 17.32 | 19.42 | 19.34 | 17.31 | 19.42 | 19.14 | 17.36 |
| WPCD | **22.08** | **22.08** | **20.08** | **22.64** | **22.40** | **20.16** | **22.48** | **22.28** | **20.20** |
| **AWTEN** | | | | | | | | | |
| OPCD | 18.44 | 10.24 | 9.03 | 18.34 | 10.18 | 9.32 | 18.44 | 10.17 | 9.29 |
| WPCD | **20.74** | **12.24** | **11.01** | **20.17** | **12.01** | **10.97** | **20.18** | **12.06** | **10.94** |
| **HDIGTWO** | | | | | | | | | |
| OPCD | 9.29 | 9.20 | 7.99 | 9.13 | 8.95 | 7.95 | 9.18 | 9.06 | 7.95 |
| WPCD | **11.73** | **11.51** | **10.03** | **11.75** | **11.60** | **10.11** | **11.79** | **11.70** | **10.22** |
| **YOUFOURV** | | | | | | | | | |
| OPCD | 13.15 | 7.89 | 7.87 | 13.16 | 8.32 | 7.68 | 13.18 | 8.42 | 7.69 |
| WPCD | **15.99** | **10.91** | **10.48** | **15.95** | **10.85** | **10.47** | **15.95** | **10.80** | **10.47** |

*Table 6.* Ablation for the Alienating Action

| A/B | 0.2 | | | 0.5 | | | 0.8 | | |
|---|---|---|---|---|---|---|---|---|---|
| | PUR | ACC | FSC | PUR | ACC | FSC | PUR | ACC | FSC |
| **FLOEVEN** | | | | | | | | | |
| OALG | 9.95 | 9.57 | 4.96 | 10.16 | 9.56 | 5.03 | 10.56 | 9.62 | 4.98 |
| WALG | **11.17** | **10.80** | **5.89** | **11.25** | **10.53** | **5.88** | **11.54** | **10.74** | **5.83** |
| **SYNTHREED** | | | | | | | | | |
| OALG | 35.26 | 34.93 | 33.12 | 35.33 | 35.17 | 33.14 | 35.83 | 35.33 | 33.27 |
| WALG | **42.50** | **42.12** | **35.21** | **42.83** | **42.83** | **35.24** | **41.04** | **41.04** | **34.59** |
| **DEOLOG** | | | | | | | | | |
| OALG | 30.14 | 21.39 | 17.44 | 30.14 | 20.12 | 17.08 | 30.41 | 22.23 | 17.61 |
| WALG | **31.84** | **22.32** | **18.45** | **31.84** | **23.74** | **18.41** | **31.28** | **24.25** | **19.56** |
| **YALTHREE** | | | | | | | | | |
| OALG | 22.92 | 21.92 | 6.58 | 19.58 | 19.01 | 4.55 | 19.80 | 19.65 | 5.36 |
| WALG | **24.94** | **23.55** | **7.89** | **21.73** | **20.61** | **6.30** | **21.55** | **20.70** | **6.70** |
| **BGFEA** | | | | | | | | | |
| OALG | 21.29 | 21.29 | 19.23 | 21.73 | 21.49 | 19.36 | **22.65** | 21.41 | 19.35 |
| WALG | **22.08** | **22.08** | **20.08** | **22.64** | **22.40** | **20.16** | 22.48 | **22.28** | **20.20** |
| **AWTEN** | | | | | | | | | |
| OALG | 18.72 | 10.82 | 9.63 | 18.72 | 10.60 | 9.58 | 18.72 | 10.58 | 9.54 |
| WALG | **20.74** | **12.24** | **11.01** | **20.17** | **12.01** | **10.97** | **20.18** | **12.06** | **10.94** |
| **HDIGTWO** | | | | | | | | | |
| OALG | 10.03 | 9.84 | 8.40 | 10.09 | 9.91 | 8.46 | 10.10 | 10.03 | 8.53 |
| WALG | **11.73** | **11.51** | **10.03** | **11.75** | **11.60** | **10.11** | **11.79** | **11.70** | **10.22** |
| **YOUFOURV** | | | | | | | | | |
| OALG | 13.90 | 8.85 | 8.44 | 13.86 | 8.81 | 8.44 | 13.82 | 8.75 | 8.45 |
| WALG | **15.99** | **10.91** | **10.48** | **15.95** | **10.85** | **10.47** | **15.95** | **10.80** | **10.47** |

not explore double-determinants, producing sub-optimal clustering outcomes.

*Space Overhead:* To demonstrate the space-friendly characteristic, we count the memory consumption, as presented in Table 3 where DAT1~DAT8 are the alternative names of datasets in Table 1. One can observe that

(1) Our BACDL consumes the least amount of memory on SYNTHREED, BGFEA, AWTEN, HDIGTWO and YOUFOURV. Especially, on YOUFOURV, it is clearly the lowest. On FLOEVEN, DEOLOG and YALTHREE, it still compares favorably with the optimal one. This gives evidence that BACDL is memory-efficient.

(2) USETL, IMVCI and AGCIM achieve slightly less overhead in some cases. The reasons could be that USETL constructs low-rank spectrum and decreases the information redundancy by multi-level partition, IMVCI generates consensus structure and skips the procedure of seeking eigenvector, AGCIM integrates the graph restoration into cluster structure and jointly conducts uniform representation construction and similarity completion. Despite resource-saving, they usually do not take into account view-specific characteristics, harming the representation diversities.

### 6.4. Ablation

Table 4, 5 and 6 summarize relevant ablation (A/B) results.

According to Table 4 where OPDD and WPDD represent the results without/with perspective-shared determinants, one can observe that WPDD consistently surpasses OPDD, confirming that the perspective-shared determinants are beneficial for performance increment.

Similarly, Table 5 where OPCD and WPCD represent the results without/with perspective-specific determinants suggests that the perspective-specific determinants indeed help improve the clustering results.

Both Table 4 and 5 illustrate that our dual-determinant paradigm can facilitate the clustering performance.

Besides, we alienate them to enhance the discrimination. Table 6 where OALG and WALG are the results without/with alienation indicates that our alienating action is functional.

Due to space limit, other ablations are located in Section L.

## 7. Concluding Remarks

In the manuscript, we devise a dual-determinant exploration paradigm for IMC issues. It bifurcates features clusters and alienates them via mutual exclusion mechanism. Together with coupled distribution learning, it effectively alleviates the dimension inconsistency. Then, it bridges all views through unified association. All sample clusters are reordered and mapped to formulate full clustering embedding. The overall linear overhead further enlarges its applicability.

## Impact Statement

This paper presents work whose goal is to advance the field of Machine Learning. There are many potential societal consequences of our work, none which we feel must be specifically highlighted here.

## Acknowledgments

This work was supported in part by the Research Grants Council (RGC) Joint Research Scheme under grant: N_HKBU214/21, the General Research Fund of RGC under the grants: 12202622 and 12202924, the RGC Senior Research Fellow Scheme under grant: SRFS2324-2S02, and the National Natural Science Foundation of China under Grant No. 62406329, 62476280, 62441618, 62325604, 62276271.

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

# Appendix

## A. Solving Procedure

We split the entire original optimization problem Eq. (6) into nine sub-problems, and manage to provide a solving scheme for each sub-problem respectively.

### A.1. $\mathbf{E}_r$ Sub-problem

Under given $\mathbf{C}$, $\mathbf{C}_r$, $\mathbf{D}$, $\mathbf{P}_r$, $\mathbf{E}$, $\mathbf{F}_r$, $a_r$ and $b_r$, the original optimization problem is equivalently transformed as

$$
\min_{\mathbf{E}_r} \sum_{r=1}^{v} a_r^2 \left\| \mathbf{X}_r \mathbf{G}_r - [\mathbf{P}_r \mathbf{C} | \mathbf{C}_r] \mathbf{D} \mathbf{E}_r^\top \mathbf{G}_r \right\|_F^2 - \beta \operatorname{Tr}\left( \mathbf{E}^\top \sum_{r=1}^{v} b_r \mathbf{E}_r \mathbf{F}_r \right)
$$
$$
\text{s.t. } \mathbf{E}_r \geq 0, \mathbf{G}_r^\top \mathbf{E}_r \mathbf{1}_k = \mathbf{1}_{n_r}.
$$
(24)

Owing to the fact that $\mathbf{E}_r$ on each view is independent of each other, we equivalently have the following optimization problem,

$$
\min_{\mathbf{E}_r} a_r^2 \left\| \mathbf{X}_r \mathbf{G}_r - [\mathbf{P}_r \mathbf{C} | \mathbf{C}_r] \mathbf{D} \mathbf{E}_r^\top \mathbf{G}_r \right\|_F^2 - \beta b_r \operatorname{Tr}\left( \mathbf{E}^\top \mathbf{E}_r \mathbf{F}_r \right)
$$
$$
\text{s.t. } \mathbf{E}_r \geq 0, \mathbf{G}_r^\top \mathbf{E}_r \mathbf{1}_k = \mathbf{1}_{n_r}.
$$
(25)

To optimize this problem, we firstly design its Lagrange function as

$$
\mathcal{L}(\mathbf{E}_r, \Phi_r) = a_r^2 \left\| \mathbf{X}_r \mathbf{G}_r - [\mathbf{P}_r \mathbf{C} | \mathbf{C}_r] \mathbf{D} \mathbf{E}_r^\top \mathbf{G}_r \right\|_F^2 - \beta b_r \operatorname{Tr}\left( \mathbf{E}^\top \mathbf{E}_r \mathbf{F}_r \right)
$$
$$
+ \operatorname{Tr}\left( \Phi_r \left( \mathbf{G}_r^\top \mathbf{E}_r \mathbf{1}_k - \mathbf{1}_{n_r} \right) \left( \mathbf{G}_r^\top \mathbf{E}_r \mathbf{1}_k - \mathbf{1}_{n_r} \right)^\top \right),
$$
(26)

where $\Phi_r \in \mathbb{R}^{n_r \times n_r}$ is the Lagrange multiplier matrix.

Expanding $F$-norm and removing irrelevant items to $\mathbf{E}_r$, we can equivalently get

$$
\mathcal{L}(\mathbf{E}_r, \Phi_r) = a_r^2 \operatorname{Tr}\left( \mathbf{G}_r^\top \mathbf{E}_r \mathbf{D}^\top [\mathbf{P}_r \mathbf{C} | \mathbf{C}_r]^\top [\mathbf{P}_r \mathbf{C} | \mathbf{C}_r] \mathbf{D} \mathbf{E}_r^\top \mathbf{G}_r - 2 \mathbf{G}_r^\top \mathbf{X}_r^\top [\mathbf{P}_r \mathbf{C} | \mathbf{C}_r] \mathbf{D} \mathbf{E}_r^\top \mathbf{G}_r \right)
$$
$$
- \beta b_r \operatorname{Tr}\left( \mathbf{E}^\top \mathbf{E}_r \mathbf{F}_r \right) + \operatorname{Tr}\left( \Phi_r \mathbf{G}_r^\top \mathbf{E}_r \mathbf{1}_k \mathbf{1}_k^\top \mathbf{E}_r^\top \mathbf{G}_r - 2 \Phi_r \mathbf{G}_r^\top \mathbf{E}_r \mathbf{1}_k \mathbf{1}_{n_r}^\top \right).
$$
(27)

Combined with the trace cyclic property and transposition invariance property, we further have

$$
\mathcal{L}(\mathbf{E}_r, \Phi_r) = \operatorname{Tr}\left( a_r^2 \mathbf{G}_r \mathbf{G}_r^\top \mathbf{E}_r \mathbf{D}^\top [\mathbf{P}_r \mathbf{C} | \mathbf{C}_r]^\top [\mathbf{P}_r \mathbf{C} | \mathbf{C}_r] \mathbf{D} \mathbf{E}_r^\top - 2 a_r^2 \mathbf{G}_r \mathbf{G}_r^\top \mathbf{X}_r^\top [\mathbf{P}_r \mathbf{C} | \mathbf{C}_r] \mathbf{D} \mathbf{E}_r^\top \right.
$$
$$
\left. - \beta b_r \mathbf{E} \mathbf{F}_r^\top \mathbf{E}_r^\top + \mathbf{G}_r \Phi_r \mathbf{G}_r^\top \mathbf{E}_r \mathbf{1}_k \mathbf{1}_k^\top \mathbf{E}_r^\top - 2 \mathbf{G}_r \Phi_r^\top \mathbf{1}_{n_r} \mathbf{1}_k^\top \mathbf{E}_r^\top \right).
$$
(28)

Let $\frac{\partial \mathcal{L}}{\partial \mathbf{E}_r} = 0$, and we have

$$
2 a_r^2 \mathbf{G}_r \mathbf{G}_r^\top \mathbf{E}_r \mathbf{D}^\top [\mathbf{P}_r \mathbf{C} | \mathbf{C}_r]^\top [\mathbf{P}_r \mathbf{C} | \mathbf{C}_r] \mathbf{D} - 2 a_r^2 \mathbf{G}_r \mathbf{G}_r^\top \mathbf{X}_r^\top [\mathbf{P}_r \mathbf{C} | \mathbf{C}_r] \mathbf{D} - \beta b_r \mathbf{E} \mathbf{F}_r^\top
$$
$$
+ \mathbf{G}_r \Phi_r \mathbf{G}_r^\top \mathbf{E}_r \mathbf{1}_k \mathbf{1}_k^\top + \mathbf{G}_r \Phi_r^\top \mathbf{G}_r^\top \mathbf{E}_r \mathbf{1}_k \mathbf{1}_k^\top - 2 \mathbf{G}_r \Phi_r^\top \mathbf{1}_{n_r} \mathbf{1}_k^\top = 0.
$$
(29)

Given that $\Phi_r$ is a diagonal matrix, we further have

$$
a_r^2 \mathbf{G}_r \mathbf{G}_r^\top \mathbf{E}_r \mathbf{D}^\top [\mathbf{P}_r \mathbf{C} | \mathbf{C}_r]^\top [\mathbf{P}_r \mathbf{C} | \mathbf{C}_r] \mathbf{D} - a_r^2 \mathbf{G}_r \mathbf{G}_r^\top \mathbf{X}_r^\top [\mathbf{P}_r \mathbf{C} | \mathbf{C}_r] \mathbf{D}
$$
$$
- \frac{\beta}{2} b_r \mathbf{E} \mathbf{F}_r^\top + \mathbf{G}_r \Phi_r \mathbf{G}_r^\top \mathbf{E}_r \mathbf{1}_k \mathbf{1}_k^\top - \mathbf{G}_r \Phi_r \mathbf{1}_{n_r} \mathbf{1}_k^\top = 0.
$$
(30)

In conjunction with complementary slackness condition for non-negative $\mathbf{E}_r$, we can get

$$
\left( a_r^2 \mathbf{G}_r \mathbf{G}_r^\top \mathbf{E}_r \mathbf{D}^\top [\mathbf{P}_r \mathbf{C} | \mathbf{C}_r]^\top [\mathbf{P}_r \mathbf{C} | \mathbf{C}_r] \mathbf{D} - a_r^2 \mathbf{G}_r \mathbf{G}_r^\top \mathbf{X}_r^\top [\mathbf{P}_r \mathbf{C} | \mathbf{C}_r] \mathbf{D} - \frac{\beta}{2} b_r \mathbf{E} \mathbf{F}_r^\top \right)_{i,j} (\mathbf{E}_r)_{i,j}
$$
$$
+ \left( \mathbf{G}_r \Phi_r \mathbf{G}_r^\top \mathbf{E}_r \mathbf{1}_k \mathbf{1}_k^\top - \mathbf{G}_r \Phi_r \mathbf{1}_{n_r} \mathbf{1}_k^\top \right)_{i,j} (\mathbf{E}_r)_{i,j} = 0.
$$
(31)

Then, to eliminate the multiplier matrix $\Phi_r$, we do normalization on the rows of $\mathbf{G}_r^\top \mathbf{E}_r \in \mathbb{R}^{n_r \times k}$ at each step optimization. Accordingly, we can obtain that $\sum_{j=1}^{k} (\mathbf{G}_r^\top \mathbf{E}_r)_{i,j} = 1$ holds for $i = 1, 2, \cdots, n_r$ where the notation $(\cdot)_{i,j}$ represents the element located in the i-th row and j-th column of the matrix. That is, $\mathbf{G}_r^\top \mathbf{E}_r \mathbf{1}_k = \mathbf{1}_{n_r}$ holds. Thus, we have

$$\mathbf{G}_r \Phi_r \mathbf{G}_r^\top \mathbf{E}_r \mathbf{1}_k \mathbf{1}_k^\top = \mathbf{G}_r \Phi_r \mathbf{1}_{n_r} \mathbf{1}_k^\top. \tag{32}$$

Combining Eq. (31) and Eq. (32) yields

$$\left( a_r^2 \mathbf{G}_r \mathbf{G}_r^\top \mathbf{E}_r \mathbf{D}^\top [\mathbf{P}_r \mathbf{C} | \mathbf{C}_r]^\top [\mathbf{P}_r \mathbf{C} | \mathbf{C}_r] \mathbf{D} - a_r^2 \mathbf{G}_r \mathbf{G}_r^\top \mathbf{X}_r^\top [\mathbf{P}_r \mathbf{C} | \mathbf{C}_r] \mathbf{D} - \frac{\beta}{2} b_r \mathbf{E} \mathbf{F}_r^\top \right)_{i,j} (\mathbf{E}_r)_{i,j} = 0. \tag{33}$$

Further, noticed that $\mathbf{E}$ and $\mathbf{F}_r$ are orthogonal matrices, and accordingly there exist negative elements in the product of $\mathbf{E}$ and $\mathbf{F}_r^\top$. In response, we partition $\mathbf{E} \mathbf{F}_r^\top$ into two parts, $\left( \mathbf{E} \mathbf{F}_r^\top \right)_{pos}$ and $\left( \mathbf{E} \mathbf{F}_r^\top \right)_{neg}$. That is, $\mathbf{E} \mathbf{F}_r^\top = \left( \mathbf{E} \mathbf{F}_r^\top \right)_{pos} - \left( \mathbf{E} \mathbf{F}_r^\top \right)_{neg}$ where $(\mathbf{E} \mathbf{F}_r^\top)_{pos}$ and $(\mathbf{E} \mathbf{F}_r^\top)_{neg}$ represent the positive part and negative part of $\mathbf{E} \mathbf{F}_r^\top$ respectively. Then, we have

$$\left( a_r^2 \mathbf{G}_r \mathbf{G}_r^\top \mathbf{E}_r \mathbf{D}^\top [\mathbf{P}_r \mathbf{C} | \mathbf{C}_r]^\top [\mathbf{P}_r \mathbf{C} | \mathbf{C}_r] \mathbf{D} - a_r^2 \mathbf{G}_r \mathbf{G}_r^\top \mathbf{X}_r^\top [\mathbf{P}_r \mathbf{C} | \mathbf{C}_r] \mathbf{D} \right.$$
$$\left. - \frac{\beta}{2} b_r \left( \mathbf{E} \mathbf{F}_r^\top \right)_{pos} + \frac{\beta}{2} b_r \left( \mathbf{E} \mathbf{F}_r^\top \right)_{neg} \right)_{i,j} (\mathbf{E}_r)_{i,j} = 0. \tag{34}$$

Consequently, we can get the following updating rule about $\mathbf{E}_r$,

$$(\mathbf{E}_r)_{i,j} \leftarrow (\mathbf{E}_r)_{i,j} \left[ \frac{\left( a_r^2 \mathbf{G}_r \mathbf{G}_r^\top \mathbf{X}_r^\top [\mathbf{P}_r \mathbf{C} | \mathbf{C}_r] \mathbf{D} + \frac{\beta}{2} b_r \left( \mathbf{E} \mathbf{F}_r^\top \right)_{pos} \right)_{i,j}}{\left( a_r^2 \mathbf{G}_r \mathbf{G}_r^\top \mathbf{E}_r \mathbf{D}^\top [\mathbf{P}_r \mathbf{C} | \mathbf{C}_r]^\top [\mathbf{P}_r \mathbf{C} | \mathbf{C}_r] \mathbf{D} + \frac{\beta}{2} b_r \left( \mathbf{E} \mathbf{F}_r^\top \right)_{neg} \right)_{i,j}} \right]^{\frac{1}{2}}. \tag{35}$$

### A.2. C Sub-problem

Under given $\mathbf{E}_r$, $\mathbf{C}_r$, $\mathbf{D}$, $\mathbf{P}_r$, $\mathbf{E}$, $\mathbf{F}_r$, $a_r$ and $b_r$, the original optimization problem is equivalently transformed as

$$\min_{\mathbf{C}} \sum_{r=1}^{v} a_r^2 \left\| \mathbf{X}_r \mathbf{G}_r - [\mathbf{P}_r \mathbf{C} | \mathbf{C}_r] \mathbf{D} \mathbf{E}_r^\top \mathbf{G}_r \right\|_F^2 + \lambda \operatorname{Tr} \left( \mathbf{C}^\top \mathbf{P}_r^\top \mathbf{C}_r \right) \tag{36}$$
$$\text{s.t. } \mathbf{C} \geq 0, \mathbf{C}^\top \mathbf{P}_r^\top \mathbf{1}_{d_r} = \mathbf{1}_k.$$

Due to the coupled property, to separate out $\mathbf{C}$, we first denote $\mathbf{D}_\gamma = \mathbf{D}_{1:k,:}$ and $\mathbf{D}_\varphi = \mathbf{D}_{k+1:2k,:}$. Then, we equivalently transform Eq. (36) the following optimization problem,

$$\min_{\mathbf{C}} \sum_{r=1}^{v} a_r^2 \left\| \mathbf{H}_r - \mathbf{P}_r \mathbf{C} \mathbf{D}_\gamma \mathbf{E}_r^\top \mathbf{G}_r \right\|_F^2 + \lambda \operatorname{Tr} \left( \mathbf{C}^\top \mathbf{P}_r^\top \mathbf{C}_r \right) \tag{37}$$
$$\text{s.t. } \mathbf{C} \geq 0, \mathbf{C}^\top \mathbf{P}_r^\top \mathbf{1}_{d_r} = \mathbf{1}_k,$$

where $\mathbf{H}_r = \mathbf{X}_r \mathbf{G}_r - \mathbf{C}_r \mathbf{D}_\varphi \mathbf{E}_r^\top \mathbf{G}_r$.

After removing irrelevant items and looping matrices, we can equivalently simplify Eq. (37) as

$$\min_{\mathbf{C}} \operatorname{Tr} \left( \sum_{r=1}^{v} a_r^2 \mathbf{P}_r^\top \mathbf{P}_r \mathbf{C} \mathbf{D}_\gamma \mathbf{E}_r^\top \mathbf{G}_r \mathbf{G}_r^\top \mathbf{E}_r \mathbf{D}_\gamma^\top \mathbf{C}^\top - 2 \left( \sum_{r=1}^{v} a_r^2 \mathbf{P}_r^\top \mathbf{H}_r \mathbf{G}_r^\top \mathbf{E}_r \mathbf{D}_\gamma^\top \right) \mathbf{C}^\top + \lambda \mathbf{C}^\top \left( \sum_{r=1}^{v} \mathbf{P}_r^\top \mathbf{C}_r \right) \right) \tag{38}$$
$$\text{s.t. } \mathbf{C} \geq 0, \mathbf{C}^\top \mathbf{P}_r^\top \mathbf{1}_{d_r} = \mathbf{1}_k.$$

Subsequently, its Lagrange function can be designed as

$$\mathcal{L}(\mathbf{C}, \Psi) = \operatorname{Tr} \left( \sum_{r=1}^{v} a_r^2 \mathbf{P}_r^\top \mathbf{P}_r \mathbf{C} \mathbf{J}_r \mathbf{C}^\top - 2 \mathbf{L} \mathbf{C}^\top + \lambda \mathbf{C}^\top \mathbf{M} + \sum_{r=1}^{v} \Psi_r \left( \mathbf{C}^\top \mathbf{P}_r^\top \mathbf{1}_{d_r} - \mathbf{1}_k \right) \left( \mathbf{C}^\top \mathbf{P}_r^\top \mathbf{1}_{d_r} - \mathbf{1}_k \right)^\top \right), \tag{39}$$

where $\mathbf{J}_r = \mathbf{D}_\gamma \mathbf{E}_r^\top \mathbf{G}_r \mathbf{G}_r^\top \mathbf{E}_r \mathbf{D}_\gamma^\top$, $\mathbf{L} = \sum_{r=1}^v a_r^2 \mathbf{P}_r^\top \mathbf{H}_r \mathbf{G}_r^\top \mathbf{E}_r \mathbf{D}_\gamma^\top$ and $\mathbf{M} = \sum_{r=1}^v \mathbf{P}_r^\top \mathbf{C}_r$. $\Psi_r \in \mathbb{R}^{k \times k}$ is the Lagrange multiplier matrix.

Combined with the zero partial derivative, we have

$$\sum_{r=1}^v a_r^2 \left( \mathbf{P}_r^\top \mathbf{P}_r \mathbf{C} \mathbf{J}_r + \mathbf{P}_r^\top \mathbf{P}_r \mathbf{C} \mathbf{J}_r^\top \right) - 2\mathbf{L} + \lambda \mathbf{M} + \sum_{r=1}^v \mathbf{P}_r^\top \mathbf{1}_{d_r} \mathbf{1}_{d_r}^\top \mathbf{P}_r \mathbf{C} \left( \Psi_r + \Psi_r^\top \right) - 2\mathbf{P}_r^\top \mathbf{1}_{d_r} \mathbf{1}_k^\top \Psi_r = 0. \quad (40)$$

Note that $\mathbf{J}_r$ and $\Psi_r$ are symmetric matrices, we can equivalently obtain

$$2\sum_{r=1}^v a_r^2 \mathbf{P}_r^\top \mathbf{P}_r \mathbf{C} \mathbf{J}_r - 2\mathbf{L} + \lambda \mathbf{M} + \sum_{r=1}^v 2\mathbf{P}_r^\top \mathbf{1}_{d_r} \mathbf{1}_{d_r}^\top \mathbf{P}_r \mathbf{C} \Psi_r - 2\mathbf{P}_r^\top \mathbf{1}_{d_r} \mathbf{1}_k^\top \Psi_r = 0. \quad (41)$$

Combined with the complementary slackness property, we further have

$$\left( \sum_{r=1}^v a_r^2 \mathbf{P}_r^\top \mathbf{P}_r \mathbf{C} \mathbf{J}_r - \mathbf{L} + \frac{\lambda}{2}\mathbf{M} + \sum_{r=1}^v \mathbf{P}_r^\top \mathbf{1}_{d_r} \mathbf{1}_{d_r}^\top \mathbf{P}_r \mathbf{C} \Psi_r - \mathbf{P}_r^\top \mathbf{1}_{d_r} \mathbf{1}_k^\top \Psi_r \right)_{i,j} \mathbf{C}_{i,j} = 0. \quad (42)$$

Therefore, we can obtain the following updating rule for $\mathbf{C}$,

$$(\mathbf{C})_{i,j} \leftarrow (\mathbf{C})_{i,j} \left[ \frac{\left( \mathbf{L} + \mathbf{P}_r^\top \mathbf{1}_{d_r} \mathbf{1}_k^\top \Psi_r \right)_{i,j}}{\left( \sum_{r=1}^v a_r^2 \mathbf{P}_r^\top \mathbf{P}_r \mathbf{C} \mathbf{J}_r + \frac{\lambda}{2}\mathbf{M} + \sum_{r=1}^v \mathbf{P}_r^\top \mathbf{1}_{d_r} \mathbf{1}_{d_r}^\top \mathbf{P}_r \mathbf{C} \Psi_r \right)_{i,j}} \right]^{\frac{1}{2}}. \quad (43)$$

Afterwards, to eliminate $\Psi_r$, we perform normalization on the columns of $\mathbf{P}_r \mathbf{C}$ so as to make $\sum_{i=1}^{d_r} (\mathbf{P}_r \mathbf{C})_{i,j}$ equal to 1 for any $j \in \{1, 2, \cdots, k\}$. That is, $\mathbf{C}^\top \mathbf{P}_r^\top \mathbf{1}_{d_r} = \mathbf{1}_k$ holds. Then, we have

$$\mathbf{1}_{d_r} \mathbf{1}_{d_r}^\top \mathbf{P}_r \mathbf{C} = \mathbf{1}_{d_r} \mathbf{1}_k^\top. \quad (44)$$

In conjunction with Eq. (42), we further have

$$(\mathbf{C})_{i,j} \leftarrow (\mathbf{C})_{i,j} \left[ \frac{(\mathbf{L})_{i,j}}{\left( \sum_{r=1}^v a_r^2 \mathbf{P}_r^\top \mathbf{P}_r \mathbf{C} \mathbf{J}_r + \frac{\lambda}{2}\mathbf{M} \right)_{i,j}} \right]^{\frac{1}{2}}. \quad (45)$$

Subsequently, based on the facts that

$$\mathbf{L} = \sum_{r=1}^v a_r^2 \mathbf{P}_r^\top \mathbf{H}_r \mathbf{G}_r^\top \mathbf{E}_r \mathbf{D}_\gamma^\top = \sum_{r=1}^v a_r^2 \left( \mathbf{P}_r^\top \mathbf{X}_r \mathbf{G}_r \mathbf{G}_r^\top \mathbf{E}_r \mathbf{D}_\gamma^\top - \mathbf{P}_r^\top \mathbf{C}_r \mathbf{D}_\varphi \mathbf{E}_r^\top \mathbf{G}_r \mathbf{G}_r^\top \mathbf{E}_r \mathbf{D}_\gamma^\top \right)$$
$$= \sum_{r=1}^v a_r^2 \mathbf{P}_r^\top \mathbf{X}_r \mathbf{G}_r \mathbf{G}_r^\top \mathbf{E}_r \mathbf{D}_\gamma^\top - \sum_{r=1}^v a_r^2 \mathbf{P}_r^\top \mathbf{C}_r \mathbf{D}_\varphi \mathbf{E}_r^\top \mathbf{G}_r \mathbf{G}_r^\top \mathbf{E}_r \mathbf{D}_\gamma^\top \quad (46)$$

and that

$$\sum_{r=1}^v a_r^2 \mathbf{P}_r^\top \mathbf{P}_r \mathbf{C} \mathbf{J}_r + \sum_{r=1}^v a_r^2 \mathbf{P}_r^\top \mathbf{C}_r \mathbf{D}_\varphi \mathbf{E}_r^\top \mathbf{G}_r \mathbf{G}_r^\top \mathbf{E}_r \mathbf{D}_\gamma^\top = \sum_{r=1}^v a_r^2 \mathbf{P}_r^\top \mathbf{P}_r \mathbf{C} \mathbf{D}_\gamma \mathbf{E}_r^\top \mathbf{G}_r \mathbf{G}_r^\top \mathbf{E}_r \mathbf{D}_\gamma^\top +$$
$$a_r^2 \mathbf{P}_r^\top \mathbf{C}_r \mathbf{D}_\varphi \mathbf{E}_r^\top \mathbf{G}_r \mathbf{G}_r^\top \mathbf{E}_r \mathbf{D}_\gamma^\top \quad (47)$$
$$= \sum_{r=1}^v a_r^2 \mathbf{P}_r^\top [\mathbf{P}_r \mathbf{C} | \mathbf{C}_r] \mathbf{D} \mathbf{E}_r^\top \mathbf{G}_r \mathbf{G}_r^\top \mathbf{E}_r \mathbf{D}_\gamma^\top$$

and Eqs. (42), (45), we can obtain the following rule,

$$(\mathbf{C})_{i,j} \leftarrow (\mathbf{C})_{i,j} \left[ \frac{\left( \sum_{r=1}^v a_r^2 \mathbf{P}_r^\top \mathbf{X}_r \mathbf{G}_r \mathbf{G}_r^\top \mathbf{E}_r \mathbf{D}_\gamma^\top \right)_{i,j}}{\left( \sum_{r=1}^v a_r^2 \mathbf{P}_r^\top [\mathbf{P}_r \mathbf{C} | \mathbf{C}_r] \mathbf{D} \mathbf{E}_r^\top \mathbf{G}_r \mathbf{G}_r^\top \mathbf{E}_r \mathbf{D}_\gamma^\top + \frac{\lambda}{2}\mathbf{M} \right)_{i,j}} \right]^{\frac{1}{2}}, \quad (48)$$

where $\mathbf{M} = \sum_{r=1}^v \mathbf{P}_r^\top \mathbf{C}_r$.

### A.3. $\mathbf{C}_r$ Sub-problem

Under given $\mathbf{E}_r$, $\mathbf{C}$, $\mathbf{D}$, $\mathbf{P}_r$, $\mathbf{E}$, $\mathbf{F}_r$, $a_r$ and $b_r$, the original optimization problem is equivalently transformed as

$$\min_{\mathbf{C}_r} \sum_{r=1}^{v} a_r^2 \left\| \mathbf{X}_r \mathbf{G}_r - [\mathbf{P}_r \mathbf{C} | \mathbf{C}_r] \mathbf{D} \mathbf{E}_r^\top \mathbf{G}_r \right\|_F^2 + \lambda \operatorname{Tr}\left( \mathbf{C}^\top \mathbf{P}_r^\top \mathbf{C}_r \right) \tag{49}$$

$$\text{s.t. } \mathbf{C}_r \geq 0, \mathbf{C}_r^\top \mathbf{1}_{d_r} = \mathbf{1}_k.$$

According to the fact that $\{\mathbf{C}_r\}_{r=1}^{v}$ are mutually independent, we can equivalently transform the above optimization problem as

$$\min_{\mathbf{C}_r} a_r^2 \left\| \mathbf{X}_r \mathbf{G}_r - [\mathbf{P}_r \mathbf{C} | \mathbf{C}_r] \mathbf{D} \mathbf{E}_r^\top \mathbf{G}_r \right\|_F^2 + \lambda \operatorname{Tr}\left( \mathbf{C}^\top \mathbf{P}_r^\top \mathbf{C}_r \right) \tag{50}$$

$$\text{s.t. } \mathbf{C}_r \geq 0, \mathbf{C}_r^\top \mathbf{1}_{d_r} = \mathbf{1}_k.$$

Splitting $\mathbf{D}$ into $\mathbf{D}_\gamma$ and $\mathbf{D}_\psi$, $\mathbf{D}_\gamma = \mathbf{D}_{1:k,:}$ and $\mathbf{D}_\psi = \mathbf{D}_{k+1:2k,:}$, we can get

$$\min_{\mathbf{C}_r} a_r^2 \left\| \mathbf{X}_r \mathbf{G}_r - \mathbf{P}_r \mathbf{C} \mathbf{D}_\gamma \mathbf{E}_r^\top \mathbf{G}_r - \mathbf{C}_r \mathbf{D}_\varphi \mathbf{E}_r^\top \mathbf{G}_r \right\|_F^2 + \lambda \operatorname{Tr}\left( \mathbf{C}^\top \mathbf{P}_r^\top \mathbf{C}_r \right) \tag{51}$$

$$\text{s.t. } \mathbf{C}_r \geq 0, \mathbf{C}_r^\top \mathbf{1}_{d_r} = \mathbf{1}_k.$$

Unfolding $F$-norm via trace operation and deleting unrelated items yield

$$\min_{\mathbf{C}_r} \operatorname{Tr}\left( a_r^2 \mathbf{C}_r \mathbf{D}_\psi \mathbf{E}_r^\top \mathbf{G}_r \mathbf{G}_r^\top \mathbf{E}_r \mathbf{D}_\psi^\top \mathbf{C}_r^\top - 2a_r^2 \mathbf{X}_r \mathbf{G}_r \mathbf{G}_r^\top \mathbf{E}_r \mathbf{D}_\psi^\top \mathbf{C}_r^\top + 2a_r^2 \mathbf{P}_r \mathbf{C} \mathbf{D}_\gamma \mathbf{E}_r^\top \mathbf{G}_r \mathbf{G}_r^\top \mathbf{E}_r \mathbf{D}_\psi^\top \mathbf{C}_r^\top + \lambda \mathbf{C}^\top \mathbf{P}_r^\top \mathbf{C}_r \right)$$

$$\text{s.t. } \mathbf{C}_r \geq 0, \mathbf{C}_r^\top \mathbf{1}_{d_r} = \mathbf{1}_k. \tag{52}$$

The Lagrange function can be written as

$$\mathcal{L}(\mathbf{C}_r, \Omega_r) = \operatorname{Tr}\left( a_r^2 \mathbf{C}_r \mathbf{D}_\psi \mathbf{E}_r^\top \mathbf{G}_r \mathbf{G}_r^\top \mathbf{E}_r \mathbf{D}_\psi^\top \mathbf{C}_r^\top - 2a_r^2 \mathbf{X}_r \mathbf{G}_r \mathbf{G}_r^\top \mathbf{E}_r \mathbf{D}_\psi^\top \mathbf{C}_r^\top + \right.$$
$$\left. 2a_r^2 \mathbf{P}_r \mathbf{C} \mathbf{D}_\gamma \mathbf{E}_r^\top \mathbf{G}_r \mathbf{G}_r^\top \mathbf{E}_r \mathbf{D}_\psi^\top \mathbf{C}_r^\top + \lambda \mathbf{C}^\top \mathbf{P}_r^\top \mathbf{C}_r + \Omega_r (\mathbf{C}_r^\top \mathbf{1}_{d_r} - \mathbf{1}_k)(\mathbf{C}_r^\top \mathbf{1}_{d_r} - \mathbf{1}_k)^\top \right), \tag{53}$$

where $\Omega_r \in \mathbb{R}^{k \times k}$ is the multiplier matrix.

Thus, we have

$$2a_r^2 \mathbf{C}_r \mathbf{D}_\psi \mathbf{E}_r^\top \mathbf{G}_r \mathbf{G}_r^\top \mathbf{E}_r \mathbf{D}_\psi^\top - 2a_r^2 \mathbf{X}_r \mathbf{G}_r \mathbf{G}_r^\top \mathbf{E}_r \mathbf{D}_\psi^\top +$$
$$2a_r^2 \mathbf{P}_r \mathbf{C} \mathbf{D}_\gamma \mathbf{E}_r^\top \mathbf{G}_r \mathbf{G}_r^\top \mathbf{E}_r \mathbf{D}_\psi^\top + \lambda \mathbf{P}_r \mathbf{C} + \mathbf{1}_{d_r} \mathbf{1}_{d_r}^\top \mathbf{C}_r \Omega_r + \mathbf{1}_{d_r} \mathbf{1}_{d_r}^\top \mathbf{C}_r \Omega_r^\top - 2 \cdot \mathbf{1}_{d_r} \mathbf{1}_k^\top \Omega_r = 0. \tag{54}$$

Further, combined with the complementary condition, we can obtain

$$\left( a_r^2 \mathbf{C}_r \mathbf{D}_\psi \mathbf{E}_r^\top \mathbf{G}_r \mathbf{G}_r^\top \mathbf{E}_r \mathbf{D}_\psi^\top + a_r^2 \mathbf{P}_r \mathbf{C} \mathbf{D}_\gamma \mathbf{E}_r^\top \mathbf{G}_r \mathbf{G}_r^\top \mathbf{E}_r \mathbf{D}_\psi^\top + \frac{\lambda}{2} \mathbf{P}_r \mathbf{C} \right)_{i,j} (\mathbf{C}_r)_{i,j}$$
$$- \left( a_r^2 \mathbf{X}_r \mathbf{G}_r \mathbf{G}_r^\top \mathbf{E}_r \mathbf{D}_\psi^\top \right)_{i,j} (\mathbf{C}_r)_{i,j} + \left( \mathbf{1}_{d_r} \mathbf{1}_{d_r}^\top \mathbf{C}_r \Omega_r - \mathbf{1}_{d_r} \mathbf{1}_k^\top \Omega_r \right)_{i,j} (\mathbf{C}_r)_{i,j} = 0. \tag{55}$$

To eliminate $\Omega_r$, we conduct normalization on the columns of $\mathbf{C}_r$ at each step optimization so as to make $\mathbf{C}_r^\top \mathbf{1}_{d_r} = \mathbf{1}_k$ hold. Based on this, we can obtain $\mathbf{1}_{d_r} \mathbf{1}_{d_r}^\top \mathbf{C}_r = \mathbf{1}_{d_r} \mathbf{1}_k^\top$. Further, we have

$$\left( a_r^2 \mathbf{C}_r \mathbf{D}_\psi \mathbf{E}_r^\top \mathbf{G}_r \mathbf{G}_r^\top \mathbf{E}_r \mathbf{D}_\psi^\top + a_r^2 \mathbf{P}_r \mathbf{C} \mathbf{D}_\gamma \mathbf{E}_r^\top \mathbf{G}_r \mathbf{G}_r^\top \mathbf{E}_r \mathbf{D}_\psi^\top + \frac{\lambda}{2} \mathbf{P}_r \mathbf{C} \right)_{i,j} (\mathbf{C}_r)_{i,j} = \left( a_r^2 \mathbf{X}_r \mathbf{G}_r \mathbf{G}_r^\top \mathbf{E}_r \mathbf{D}_\psi^\top \right)_{i,j} (\mathbf{C}_r)_{i,j}. \tag{56}$$

Thus, we can have the following updating rule for $\mathbf{C}_r$,

$$(\mathbf{C}_r)_{i,j} \leftarrow (\mathbf{C}_r)_{i,j} \left[ \frac{\left( a_r^2 \mathbf{X}_r \mathbf{G}_r \mathbf{G}_r^\top \mathbf{E}_r \mathbf{D}_\psi^\top \right)_{i,j}}{\left( a_r^2 \mathbf{C}_r \mathbf{D}_\psi \mathbf{E}_r^\top \mathbf{G}_r \mathbf{G}_r^\top \mathbf{E}_r \mathbf{D}_\psi^\top + a_r^2 \mathbf{P}_r \mathbf{C} \mathbf{D}_\gamma \mathbf{E}_r^\top \mathbf{G}_r \mathbf{G}_r^\top \mathbf{E}_r \mathbf{D}_\psi^\top + \frac{\lambda}{2} \mathbf{P}_r \mathbf{C} \right)_{i,j}} \right]^{\frac{1}{2}}. \tag{57}$$

Combined with the fact that $\mathbf{C}_r\mathbf{D}_\psi + \mathbf{P}_r\mathbf{C}\mathbf{D}_\gamma = [\mathbf{P}_r\mathbf{C}|\mathbf{C}_r]\mathbf{D}$, we can equivalently have

$$(\mathbf{C}_r)_{i,j} \leftarrow (\mathbf{C}_r)_{i,j} \left[ \frac{\left(a_r^2\mathbf{X}_r\mathbf{G}_r\mathbf{G}_r^\top\mathbf{E}_r\mathbf{D}_\psi^\top\right)_{i,j}}{\left(a_r^2[\mathbf{P}_r\mathbf{C}|\mathbf{C}_r]\mathbf{D}\mathbf{E}_r^\top\mathbf{G}_r\mathbf{G}_r^\top\mathbf{E}_r\mathbf{D}_\psi^\top + \frac{\lambda}{2}\mathbf{P}_r\mathbf{C}\right)_{i,j}} \right]^{\frac{1}{2}}. \tag{58}$$

### A.4. D Sub-problem

Under given $\mathbf{E}_r$, $\mathbf{C}$, $\mathbf{C}_r$, $\mathbf{P}_r$, $\mathbf{E}$, $\mathbf{F}_r$, $a_r$ and $b_r$, the original optimization problem is equivalently transformed as

$$\min_{\mathbf{D}} \sum_{r=1}^{v} a_r^2 \left\| \mathbf{X}_r\mathbf{G}_r - [\mathbf{P}_r\mathbf{C}|\mathbf{C}_r]\mathbf{D}\mathbf{E}_r^\top\mathbf{G}_r \right\|_F^2 \tag{59}$$

$$\text{s.t. } \mathbf{D} \geq 0.$$

After norm unfolding and unrelated item removing, we can equivalently obtain the following optimization problem,

$$\min_{\mathbf{D}} \sum_{r=1}^{v} a_r^2 \operatorname{Tr}\left([\mathbf{P}_r\mathbf{C}|\mathbf{C}_r]\mathbf{D}\mathbf{E}_r^\top\mathbf{G}_r\mathbf{G}_r^\top\mathbf{E}_r\mathbf{D}^\top[\mathbf{P}_r\mathbf{C}|\mathbf{C}_r]^\top - 2\mathbf{X}_r\mathbf{G}_r\mathbf{G}_r^\top\mathbf{E}_r\mathbf{D}^\top[\mathbf{P}_r\mathbf{C}|\mathbf{C}_r]^\top\right) \tag{60}$$

$$\text{s.t. } \mathbf{D} \geq 0.$$

By means of the trace cyclic characteristic, we can further transform the above optimization problem as

$$\min_{\mathbf{D}} \sum_{r=1}^{v} a_r^2 \operatorname{Tr}\left([\mathbf{P}_r\mathbf{C}|\mathbf{C}_r]^\top[\mathbf{P}_r\mathbf{C}|\mathbf{C}_r]\mathbf{D}\mathbf{E}_r^\top\mathbf{G}_r\mathbf{G}_r^\top\mathbf{E}_r\mathbf{D}^\top - 2[\mathbf{P}_r\mathbf{C}|\mathbf{C}_r]^\top\mathbf{X}_r\mathbf{G}_r\mathbf{G}_r^\top\mathbf{E}_r\mathbf{D}^\top\right) \tag{61}$$

$$\text{s.t. } \mathbf{D} \geq 0.$$

Let the partial derivation of objective function equal to zero, we have

$$\sum_{r=1}^{v} a_r^2 \left([\mathbf{P}_r\mathbf{C}|\mathbf{C}_r]^\top[\mathbf{P}_r\mathbf{C}|\mathbf{C}_r]\mathbf{D}\mathbf{E}_r^\top\mathbf{G}_r\mathbf{G}_r^\top\mathbf{E}_r - [\mathbf{P}_r\mathbf{C}|\mathbf{C}_r]^\top\mathbf{X}_r\mathbf{G}_r\mathbf{G}_r^\top\mathbf{E}_r\right) = 0. \tag{62}$$

Further, in conjunction with the complementary condition, we can obtain

$$\left(\sum_{r=1}^{v} a_r^2[\mathbf{P}_r\mathbf{C}|\mathbf{C}_r]^\top[\mathbf{P}_r\mathbf{C}|\mathbf{C}_r]\mathbf{D}\mathbf{E}_r^\top\mathbf{G}_r\mathbf{G}_r^\top\mathbf{E}_r\right)_{i,j}\mathbf{D}_{i,j} = \left(\sum_{r=1}^{v} a_r^2[\mathbf{P}_r\mathbf{C}|\mathbf{C}_r]^\top\mathbf{X}_r\mathbf{G}_r\mathbf{G}_r^\top\mathbf{E}_r\right)_{i,j}\mathbf{D}_{i,j}. \tag{63}$$

Therefore, we have the following updating rule for the variable $\mathbf{D}$,

$$\mathbf{D}_{i,j} \leftarrow \mathbf{D}_{i,j} \left[ \frac{\left(\sum_{r=1}^{v} a_r^2[\mathbf{P}_r\mathbf{C}|\mathbf{C}_r]^\top\mathbf{X}_r\mathbf{G}_r\mathbf{G}_r^\top\mathbf{E}_r\right)_{i,j}}{\left(\sum_{r=1}^{v} a_r^2[\mathbf{P}_r\mathbf{C}|\mathbf{C}_r]^\top[\mathbf{P}_r\mathbf{C}|\mathbf{C}_r]\mathbf{D}\mathbf{E}_r^\top\mathbf{G}_r\mathbf{G}_r^\top\mathbf{E}_r\right)_{i,j}} \right]^{\frac{1}{2}}. \tag{64}$$

### A.5. $\mathbf{P}_r$ Sub-problem

Under given $\mathbf{E}_r$, $\mathbf{C}$, $\mathbf{C}_r$, $\mathbf{D}$, $\mathbf{E}$, $\mathbf{F}_r$, $a_r$ and $b_r$, the original optimization problem is equivalently transformed as

$$\min_{\mathbf{P}_r} \sum_{r=1}^{v} a_r^2 \left\| \mathbf{X}_r\mathbf{G}_r - [\mathbf{P}_r\mathbf{C}|\mathbf{C}_r]\mathbf{D}\mathbf{E}_r^\top\mathbf{G}_r \right\|_F^2 + \lambda \operatorname{Tr}\left(\mathbf{C}^\top\mathbf{P}_r^\top\mathbf{C}_r\right) \tag{65}$$

$$\text{s.t. } \mathbf{P}_r \geq 0, \mathbf{C}^\top\mathbf{P}_r^\top\mathbf{1}_{d_r} = \mathbf{1}_k.$$

Based on the fact that guidance matrices $\{\mathbf{P}_r\}_{r=1}^{v}$ on different views are independent of each other, we can equivalently obtain the following optimization problem,

$$\min_{\mathbf{P}_r} a_r^2 \left\| \mathbf{X}_r\mathbf{G}_r - [\mathbf{P}_r\mathbf{C}|\mathbf{C}_r]\mathbf{D}\mathbf{E}_r^\top\mathbf{G}_r \right\|_F^2 + \lambda \operatorname{Tr}\left(\mathbf{C}^\top\mathbf{P}_r^\top\mathbf{C}_r\right) \tag{66}$$

$$\text{s.t. } \mathbf{P}_r \geq 0, \mathbf{C}^\top\mathbf{P}_r^\top\mathbf{1}_{d_r} = \mathbf{1}_k.$$

After removing irrelevant terms and combining trace cyclic property, we can equivalently have

$$\min_{\mathbf{P}_r} \text{Tr}\left(a_r^2 \mathbf{P}_r \mathbf{C} \mathbf{D}_\gamma \mathbf{E}_r^\top \mathbf{G}_r \mathbf{G}_r^\top \mathbf{E}_r \mathbf{D}_\gamma^\top \mathbf{C}^\top \mathbf{P}_r^\top - 2a_r^2 \left(\mathbf{X}_r \mathbf{G}_r - \mathbf{C}_r \mathbf{D}_\varphi \mathbf{E}_r^\top \mathbf{G}_r\right) \mathbf{G}_r^\top \mathbf{E}_r \mathbf{D}_\gamma^\top \mathbf{C}^\top \mathbf{P}_r^\top + \lambda \mathbf{P}_r^\top \mathbf{C}_r \mathbf{C}^\top\right)$$

$$\mathbf{P}_r \geq 0, \mathbf{C}^\top \mathbf{P}_r^\top \mathbf{1}_{d_r} = \mathbf{1}_k, \tag{67}$$

where the last item is based on the fact that $\text{Tr}\left(\mathbf{C}^\top \mathbf{P}_r^\top \mathbf{C}_r\right)$ is equal to $\text{Tr}\left(\mathbf{P}_r^\top \mathbf{C}_r \mathbf{C}^\top\right)$.

Its Lagrange function can be formulated as

$$\mathcal{L}(\mathbf{P}_r, \Gamma_r) = \text{Tr}\left(a_r^2 \mathbf{P}_r \mathbf{C} \mathbf{D}_\gamma \mathbf{E}_r^\top \mathbf{G}_r \mathbf{G}_r^\top \mathbf{E}_r \mathbf{D}_\gamma^\top \mathbf{C}^\top \mathbf{P}_r^\top - 2a_r^2 \left(\mathbf{X}_r \mathbf{G}_r - \mathbf{C}_r \mathbf{D}_\varphi \mathbf{E}_r^\top \mathbf{G}_r\right) \mathbf{G}_r^\top \mathbf{E}_r \mathbf{D}_\gamma^\top \mathbf{C}^\top \mathbf{P}_r^\top + \right.$$

$$\left. \lambda \mathbf{P}_r^\top \mathbf{C}_r \mathbf{C}^\top + \Gamma_r(\mathbf{C}^\top \mathbf{P}_r^\top \mathbf{1}_{d_r} - \mathbf{1}_k)(\mathbf{C}^\top \mathbf{P}_r^\top \mathbf{1}_{d_r} - \mathbf{1}_k)^\top\right), \tag{68}$$

where $\Gamma_r \in \mathbb{R}^{k \times k}$ denotes the multiplier matrix.

Given the zero partial derivation, we can get

$$2a_r^2 \mathbf{P}_r \mathbf{C} \mathbf{D}_\gamma \mathbf{E}_r^\top \mathbf{G}_r \mathbf{G}_r^\top \mathbf{E}_r \mathbf{D}_\gamma^\top \mathbf{C}^\top - 2a_r^2 \left(\mathbf{X}_r \mathbf{G}_r - \mathbf{C}_r \mathbf{D}_\varphi \mathbf{E}_r^\top \mathbf{G}_r\right) \mathbf{G}_r^\top \mathbf{E}_r \mathbf{D}_\gamma^\top \mathbf{C}^\top$$

$$+ \lambda \mathbf{C}_r \mathbf{C}^\top + \mathbf{1}_{d_r} \mathbf{1}_{d_r}^\top \mathbf{P}_r \mathbf{C}(\Gamma_r + \Gamma_r^\top)\mathbf{C}^\top - 2 \cdot \mathbf{1}_{d_r} \mathbf{1}_k^\top \Gamma_r \mathbf{C}^\top = 0. \tag{69}$$

Combined with the symmetry of $\Gamma_r$ and complementarity of $\mathbf{P}_r$, we have

$$\left(a_r^2 \mathbf{P}_r \mathbf{C} \mathbf{D}_\gamma \mathbf{E}_r^\top \mathbf{G}_r \mathbf{G}_r^\top \mathbf{E}_r \mathbf{D}_\gamma^\top \mathbf{C}^\top - a_r^2 \left(\mathbf{X}_r \mathbf{G}_r - \mathbf{C}_r \mathbf{D}_\varphi \mathbf{E}_r^\top \mathbf{G}_r\right) \mathbf{G}_r^\top \mathbf{E}_r \mathbf{D}_\gamma^\top \mathbf{C}^\top\right.$$

$$\left. + \frac{\lambda}{2} \mathbf{C}_r \mathbf{C}^\top + \mathbf{1}_{d_r} \mathbf{1}_{d_r}^\top \mathbf{P}_r \mathbf{C} \Gamma_r \mathbf{C}^\top - \mathbf{1}_{d_r} \mathbf{1}_k^\top \Gamma_r \mathbf{C}^\top\right)_{i,j} (\mathbf{P}_r)_{i,j} = 0. \tag{70}$$

Further, we conduct normalization operation on the columns of $\mathbf{P}_r \mathbf{C}$ to ensure $\mathbf{C}^\top \mathbf{P}_r^\top \mathbf{1}_{d_r} = \mathbf{1}_k$ hold. Accordingly, we can obtain

$$\left(a_r^2 \mathbf{P}_r \mathbf{C} \mathbf{D}_\gamma \mathbf{E}_r^\top \mathbf{G}_r \mathbf{G}_r^\top \mathbf{E}_r \mathbf{D}_\gamma^\top \mathbf{C}^\top + a_r^2 \mathbf{C}_r \mathbf{D}_\varphi \mathbf{E}_r^\top \mathbf{G}_r \mathbf{G}_r^\top \mathbf{E}_r \mathbf{D}_\gamma^\top \mathbf{C}^\top + \frac{\lambda}{2} \mathbf{C}_r \mathbf{C}^\top\right)_{i,j} (\mathbf{P}_r)_{i,j} =$$

$$\left(a_r^2 \mathbf{X}_r \mathbf{G}_r \mathbf{G}_r^\top \mathbf{E}_r \mathbf{D}_\gamma^\top \mathbf{C}^\top\right) (\mathbf{P}_r)_{i,j}. \tag{71}$$

Kindly note that $\mathbf{P}_r \mathbf{C} \mathbf{D}_\gamma \mathbf{E}_r^\top \mathbf{G}_r \mathbf{G}_r^\top \mathbf{E}_r \mathbf{D}_\gamma^\top \mathbf{C}^\top + \mathbf{C}_r \mathbf{D}_\varphi \mathbf{E}_r^\top \mathbf{G}_r \mathbf{G}_r^\top \mathbf{E}_r \mathbf{D}_\gamma^\top \mathbf{C}^\top = [\mathbf{P}_r \mathbf{C} | \mathbf{C}_r] \mathbf{D} \mathbf{E}_r^\top \mathbf{G}_r \mathbf{G}_r^\top \mathbf{E}_r \mathbf{D}_\gamma^\top \mathbf{C}^\top$, and therefore we can obtain the following updating rule for $\mathbf{P}_r$,

$$(\mathbf{P}_r)_{i,j} \leftarrow (\mathbf{P}_r)_{i,j} \left[\frac{\left(a_r^2 \mathbf{X}_r \mathbf{G}_r \mathbf{G}_r^\top \mathbf{E}_r \mathbf{D}_\gamma^\top \mathbf{C}^\top\right)_{i,j}}{\left(a_r^2 [\mathbf{P}_r \mathbf{C} | \mathbf{C}_r] \mathbf{D} \mathbf{E}_r^\top \mathbf{G}_r \mathbf{G}_r^\top \mathbf{E}_r \mathbf{D}_\gamma^\top \mathbf{C}^\top + \frac{\lambda}{2} \mathbf{C}_r \mathbf{C}^\top\right)_{i,j}}\right]^{\frac{1}{2}}. \tag{72}$$

### A.6. E Sub-problem

Under given $\mathbf{E}_r$, $\mathbf{C}$, $\mathbf{C}_r$, $\mathbf{D}$, $\mathbf{P}_r$, $\mathbf{F}_r$, $a_r$ and $b_r$, the original optimization problem is equivalently transformed as

$$\max_{\mathbf{E}} \text{Tr}\left(\mathbf{E}^\top \sum_{r=1}^v b_r \mathbf{E}_r \mathbf{F}_r\right) \tag{73}$$

$$\text{s.t. } \mathbf{E}^\top \mathbf{E} = \mathbf{I}_k.$$

Let $\sum_{r=1}^v b_r \mathbf{E}_r \mathbf{F}_r$ equal to $\mathbf{U} \boldsymbol{\Sigma} \mathbf{V}^\top$ where $\mathbf{U}$, $\boldsymbol{\Sigma}$ and $\mathbf{V}$ denote its left singular matrix, singular value matrix, right singular matrix, respectively. Then, the optimal solution of $\mathbf{E}$ can be acquired by setting $\mathbf{E} = \mathbf{U} \mathbf{V}^\top$.

### A.7. $\mathbf{F}_r$ Sub-problem

Under given $\mathbf{E}_r$, $\mathbf{C}$, $\mathbf{C}_r$, $\mathbf{D}$, $\mathbf{P}_r$, $\mathbf{E}$, $a_r$ and $b_r$, the original optimization problem is equivalently transformed as

$$\max_{\mathbf{F}_r} \text{Tr}\left(\mathbf{E}^\top \sum_{r=1}^v b_r \mathbf{E}_r \mathbf{F}_r\right) \tag{74}$$

$$\text{s.t. } \mathbf{F}_r \mathbf{F}_r^\top = \mathbf{I}_k.$$

Owing to $\{\mathbf{F}_r\}_{r=1}^v$ being independent of each other, we can further transform the above optimization problem as

$$\max_{\mathbf{F}_r} \mathrm{Tr}\left(\mathbf{E}^\top \mathbf{E}_r \mathbf{F}_r\right)$$
$$\text{s.t. } \mathbf{F}_r \mathbf{F}_r^\top = \mathbf{I}_k. \tag{75}$$

Denote $\mathbf{S}_r^\top$ as $\mathbf{F}_r$, then, we equivalently have the following problem,

$$\max_{\mathbf{S}_r} \mathrm{Tr}\left(\mathbf{S}_r^\top \mathbf{E}^\top \mathbf{E}_r\right)$$
$$\text{s.t. } \mathbf{S}_r^\top \mathbf{S}_r = \mathbf{I}_k. \tag{76}$$

Therefore, the optimal solution of variable $\mathbf{F}_r$ is $\mathbf{U}\mathbf{V}^\top$ where $\mathbf{U}$ and $\mathbf{V}$ represent the left and right singular matrices of $\mathbf{E}_r^\top \mathbf{E}$ respectively.

### A.8. $a_r$ Sub-problem

Under given $\mathbf{E}_r$, $\mathbf{C}$, $\mathbf{C}_r$, $\mathbf{D}$, $\mathbf{P}_r$, $\mathbf{E}$, $\mathbf{F}_r$ and $b_r$, the original optimization problem is equivalently transformed as

$$\min_{a_r} \sum_{r=1}^v a_r^2 \left\| \mathbf{X}_r \mathbf{G}_r - [\mathbf{P}_r \mathbf{C} | \mathbf{C}_r] \mathbf{D} \mathbf{E}_r^\top \mathbf{G}_r \right\|_F^2$$
$$\text{s.t. } \sum_{r=1}^v a_r = 1, a_r \geq 0. \tag{77}$$

To solve it, we formulate its Lagrange function as

$$\mathcal{L}(a_r, \zeta) = \sum_{r=1}^v a_r^2 \left\| \mathbf{X}_r \mathbf{G}_r - [\mathbf{P}_r \mathbf{C} | \mathbf{C}_r] \mathbf{D} \mathbf{E}_r^\top \mathbf{G}_r \right\|_F^2 + \zeta \left( \sum_{r=1}^v a_r - 1 \right). \tag{78}$$

Then, we have

$$2a_r \left\| \mathbf{X}_r \mathbf{G}_r - [\mathbf{P}_r \mathbf{C} | \mathbf{C}_r] \mathbf{D} \mathbf{E}_r^\top \mathbf{G}_r \right\|_F^2 + \zeta = 0, r = 1, 2, \cdots, v; a_1 + a_2 + \cdots + a_v - 1 = 0. \tag{79}$$

Thus, we can further obtain

$$a_r = -\frac{\zeta}{2 \left\| \mathbf{X}_r \mathbf{G}_r - [\mathbf{P}_r \mathbf{C} | \mathbf{C}_r] \mathbf{D} \mathbf{E}_r^\top \mathbf{G}_r \right\|_F^2}. \tag{80}$$

Combined with $a_1 + a_2 + \cdots + a_v - 1 = 0$, we can get

$$\zeta = -\frac{2}{\sum_{r=1}^v \frac{1}{\left\| \mathbf{X}_r \mathbf{G}_r - [\mathbf{P}_r \mathbf{C} | \mathbf{C}_r] \mathbf{D} \mathbf{E}_r^\top \mathbf{G}_r \right\|_F^2}}. \tag{81}$$

Plugging Eq. (81) into Eq. (80), consequently, we have

$$a_r = \frac{\frac{\sum_{r=1}^v \frac{1}{\left\| \mathbf{X}_r \mathbf{G}_r - [\mathbf{P}_r \mathbf{C} | \mathbf{C}_r] \mathbf{D} \mathbf{E}_r^\top \mathbf{G}_r \right\|_F^2}}{\left\| \mathbf{X}_r \mathbf{G}_r - [\mathbf{P}_r \mathbf{C} | \mathbf{C}_r] \mathbf{D} \mathbf{E}_r^\top \mathbf{G}_r \right\|_F^2}}{} = \frac{\frac{1}{\left\| \mathbf{X}_r \mathbf{G}_r - [\mathbf{P}_r \mathbf{C} | \mathbf{C}_r] \mathbf{D} \mathbf{E}_r^\top \mathbf{G}_r \right\|_F^2}}{\sum_{r=1}^v \frac{1}{\left\| \mathbf{X}_r \mathbf{G}_r - [\mathbf{P}_r \mathbf{C} | \mathbf{C}_r] \mathbf{D} \mathbf{E}_r^\top \mathbf{G}_r \right\|_F^2}}. \tag{82}$$

### A.9. $b_r$ Sub-problem

Under given $\mathbf{E}_r$, $\mathbf{C}$, $\mathbf{C}_r$, $\mathbf{D}$, $\mathbf{P}_r$, $\mathbf{E}$, $\mathbf{F}_r$ and $a_r$, the original optimization problem is equivalently transformed as

$$\max_{b_r} \sum_{r=1}^v b_r \mathrm{Tr}\left(\mathbf{E}^\top \mathbf{E}_r \mathbf{F}_r\right)$$
$$\text{s.t. } \sum_{r=1}^v b_r^2 = 1, b_r \geq 0. \tag{83}$$

We provide two solving solutions for it. Using Lagrange function method, we can get

$$\mathcal{L}(b_r, \eta) = \sum_{r=1}^{v} b_r \operatorname{Tr}\left(\mathbf{E}^\top \mathbf{E}_r \mathbf{F}_r\right) + \eta \left(\sum_{r=1}^{v} b_r^2 - 1\right). \tag{84}$$

Then, we have

$$\operatorname{Tr}\left(\mathbf{E}^\top \mathbf{E}_r \mathbf{F}_r\right) + 2\eta b_r = 0, \sum_{r=1}^{v} b_r^2 - 1 = 0. \tag{85}$$

Accordingly, we can obtain $b_r = -\frac{\operatorname{Tr}\left(\mathbf{E}^\top \mathbf{E}_r \mathbf{F}_r\right)}{2\eta}$. In conjunction with $\sum_{r=1}^{v} b_r^2 - 1 = 0$, we have $\eta = \pm\frac{1}{2}\left(\sum_{r=1}^{v}\left(\operatorname{Tr}\left(\mathbf{E}^\top \mathbf{E}_r \mathbf{F}_r\right)\right)^2\right)^{\frac{1}{2}}$. Further, combined with the non-negative constraint, we have

$$b_r = \frac{\operatorname{Tr}\left(\mathbf{E}^\top \mathbf{E}_r \mathbf{F}_r\right)}{\left(\sum_{r=1}^{v}\left(\operatorname{Tr}\left(\mathbf{E}^\top \mathbf{E}_r \mathbf{F}_r\right)\right)^2\right)^{\frac{1}{2}}}. \tag{86}$$

Using Cauchy inequality $\left(\sum_{i=1}^{n} x_i y_i\right)^2 \leq \left(\sum_{i=1}^{n} x_i^2\right)\left(\sum_{i=1}^{n} y_i^2\right)$, we have

$$\left(\sum_{r=1}^{v} b_r \operatorname{Tr}\left(\mathbf{E}^\top \mathbf{E}_r \mathbf{F}_r\right)\right)^2 \leq \sum_{r=1}^{v}\left(\operatorname{Tr}\left(\mathbf{E}^\top \mathbf{E}_r \mathbf{F}_r\right)\right)^2, \tag{87}$$

where the equality holds if and only if $\frac{b_1}{\operatorname{Tr}(\mathbf{E}^\top \mathbf{E}_1 \mathbf{F}_1)} = \frac{b_2}{\operatorname{Tr}(\mathbf{E}^\top \mathbf{E}_2 \mathbf{F}_2)} = \cdots = \frac{b_v}{\operatorname{Tr}(\mathbf{E}^\top \mathbf{E}_v \mathbf{F}_v)}$.

Let $\frac{b_r}{\operatorname{Tr}(\mathbf{E}^\top \mathbf{E}_r \mathbf{F}_r)} = c$ where $c$ is a constant, and then we have $b_r = c \operatorname{Tr}\left(\mathbf{E}^\top \mathbf{E}_r \mathbf{F}_r\right)$. Further, according to $\sum_{r=1}^{v} b_r^2 = 1$, we can obtain $\sum_{r=1}^{v} c^2 \left(\operatorname{Tr}\left(\mathbf{E}^\top \mathbf{E}_r \mathbf{F}_r\right)\right)^2 = 1$. So, $c = \frac{1}{\sqrt{\sum_{r=1}^{v}(\operatorname{Tr}(\mathbf{E}^\top \mathbf{E}_r \mathbf{F}_r))^2}}$. Accordingly, the optimal solution of $b_r$ is $\frac{\operatorname{Tr}\left(\mathbf{E}^\top \mathbf{E}_r \mathbf{F}_r\right)}{\sqrt{\sum_{r=1}^{v}(\operatorname{Tr}(\mathbf{E}^\top \mathbf{E}_r \mathbf{F}_r))^2}}$.

## B. Proof of Theorem 1

*Proof.* During optimizing $\mathbf{E}_r$, on the basis of Eq. (28), for the term $\operatorname{Tr}\left(\mathbf{E} \mathbf{F}_r^\top \mathbf{E}_r^\top\right)$, due to the orthogonal properties of $\mathbf{E}$ and $\mathbf{F}_r$, the elements of $\mathbf{E}\mathbf{F}_r^\top$ could not be non-negative. In view of this, we split $\mathbf{E}\mathbf{F}_r^\top$ into $(\mathbf{E}\mathbf{F}_r^\top)_{pos}$ and $(\mathbf{E}\mathbf{F}_r^\top)_{neg}$, i.e., $\mathbf{E}\mathbf{F}_r^\top = (\mathbf{E}\mathbf{F}_r^\top)_{pos} - (\mathbf{E}\mathbf{F}_r^\top)_{neg}$ where $(\mathbf{E}\mathbf{F}_r^\top)_{pos}$ and $(\mathbf{E}\mathbf{F}_r^\top)_{neg}$ denote the positive part and negative part of $\mathbf{E}\mathbf{F}_r^\top$ respectively. Therefore, we have $\operatorname{Tr}\left(\mathbf{E}\mathbf{F}_r^\top \mathbf{E}_r^\top\right) = \operatorname{Tr}\left((\mathbf{E}\mathbf{F}_r^\top)_{pos}\mathbf{E}_r^\top\right) - \operatorname{Tr}\left((\mathbf{E}\mathbf{F}_r^\top)_{neg}\mathbf{E}_r^\top\right)$.

For the term $\operatorname{Tr}\left((\mathbf{E}\mathbf{F}_r^\top)_{pos}\mathbf{E}_r^\top\right)$, combined with the non-negative property of $\mathbf{E}_r$, we have

$$\operatorname{Tr}\left((\mathbf{E}\mathbf{F}_r^\top)_{pos}\mathbf{E}_r^\top\right) = \sum (\mathbf{E}_r)_{i,j}\left((\mathbf{E}\mathbf{F}_r^\top)_{pos}\right)_{i,j} \geq \sum (\widetilde{\mathbf{E}}_r)_{i,j}\left((\mathbf{E}\mathbf{F}_r^\top)_{pos}\right)_{i,j}\left(1 + \log\frac{(\mathbf{E}_r)_{i,j}}{(\widetilde{\mathbf{E}}_r)_{i,j}}\right), \tag{88}$$

where the inequality holds based on the fact that $x \geq 1 + \log x$ in which $x$ requires to be non-negative.

Then, for the term $\operatorname{Tr}\left((\mathbf{E}\mathbf{F}_r^\top)_{neg}\mathbf{E}_r^\top\right)$, via element expanding, we have

$$\operatorname{Tr}\left((\mathbf{E}\mathbf{F}_r^\top)_{neg}\mathbf{E}_r^\top\right) = \sum (\mathbf{E}_r)_{i,j}\left((\mathbf{E}\mathbf{F}_r^\top)_{neg}\right)_{i,j} \leq \frac{1}{2}\sum\left((\mathbf{E}\mathbf{F}_r^\top)_{neg}\right)_{i,j}\frac{(\widetilde{\mathbf{E}}_r)_{i,j}^2 + (\mathbf{E}_r)_{i,j}^2}{(\widetilde{\mathbf{E}}_r)_{i,j}}. \tag{89}$$

Therefore, for the term $\operatorname{Tr}\left(\mathbf{E}\mathbf{F}_r^\top \mathbf{E}_r^\top\right)$ we have

$$\operatorname{Tr}\left(\mathbf{E}\mathbf{F}_r^\top \mathbf{E}_r^\top\right) \geq \sum (\widetilde{\mathbf{E}}_r)_{i,j}\left((\mathbf{E}\mathbf{F}_r^\top)_{pos}\right)_{i,j}\left(1 + \log\frac{(\mathbf{E}_r)_{i,j}}{(\widetilde{\mathbf{E}}_r)_{i,j}}\right) - \frac{1}{2}\sum\left((\mathbf{E}\mathbf{F}_r^\top)_{neg}\right)_{i,j}\frac{(\widetilde{\mathbf{E}}_r)_{i,j}^2 + (\mathbf{E}_r)_{i,j}^2}{(\widetilde{\mathbf{E}}_r)_{i,j}}. \tag{90}$$

Subsequently, for the term $\mathrm{Tr}\left(\mathbf{G}_r\mathbf{G}_r^\top\mathbf{X}_r^\top[\mathbf{P}_r\mathbf{C}|\mathbf{C}_r]\mathbf{D}\mathbf{E}_r^\top\right)$ in Eq. (28), in conjunction with the non-negative property of $\mathbf{G}_r\mathbf{G}_r^\top\mathbf{X}_r^\top[\mathbf{P}_r\mathbf{C}|\mathbf{C}_r]\mathbf{D}$, we can obtain

$$\mathrm{Tr}\left(\mathbf{G}_r\mathbf{G}_r^\top\mathbf{X}_r^\top[\mathbf{P}_r\mathbf{C}|\mathbf{C}_r]\mathbf{D}\mathbf{E}_r^\top\right) \geq \sum(\widetilde{\mathbf{E}}_r)_{i,j}(\mathbf{G}_r\mathbf{G}_r^\top\mathbf{X}_r^\top[\mathbf{P}_r\mathbf{C}|\mathbf{C}_r]\mathbf{D})_{i,j}\left(1+\log\frac{(\mathbf{E}_r)_{i,j}}{(\widetilde{\mathbf{E}}_r)_{i,j}}\right). \tag{91}$$

For the matrix $\mathbf{G}_r\Phi_r^\top\mathbf{1}_{n_r}\mathbf{1}_k^\top\mathbf{E}_r^\top$, likewise, we have

$$\mathrm{Tr}\left(\mathbf{G}_r\Phi_r^\top\mathbf{1}_{n_r}\mathbf{1}_k^\top\mathbf{E}_r^\top\right) = \sum(\mathbf{E}_r)_{i,j}(\mathbf{G}_r\Phi_r^\top\mathbf{1}_{n_r}\mathbf{1}_k^\top)_{i,j} \geq \sum(\widetilde{\mathbf{E}}_r)_{i,j}(\mathbf{G}_r\Phi_r^\top\mathbf{1}_{n_r}\mathbf{1}_k^\top)_{i,j}\left(1+\log\frac{(\mathbf{E}_r)_{i,j}}{(\widetilde{\mathbf{E}}_r)_{i,j}}\right). \tag{92}$$

Further, for the term $\mathrm{Tr}\left(\mathbf{G}_r\mathbf{G}_r^\top\mathbf{E}_r\mathbf{D}^\top[\mathbf{P}_r\mathbf{C}|\mathbf{C}_r]^\top[\mathbf{P}_r\mathbf{C}|\mathbf{C}_r]\mathbf{D}\mathbf{E}_r^\top\right)$, by utilizing element unfolding and matrix symmetry, we can get

$$\begin{aligned}\mathrm{Tr}\left(\mathbf{G}_r\mathbf{G}_r^\top\mathbf{E}_r\mathbf{D}^\top[\mathbf{P}_r\mathbf{C}|\mathbf{C}_r]^\top[\mathbf{P}_r\mathbf{C}|\mathbf{C}_r]\mathbf{D}\mathbf{E}_r^\top\right) &= \sum\sum(\mathbf{G}_r\mathbf{G}_r^\top)_{i,s}(\mathbf{E}_r)_{s,t}(\mathbf{D}^\top[\mathbf{P}_r\mathbf{C}|\mathbf{C}_r]^\top[\mathbf{P}_r\mathbf{C}|\mathbf{C}_r]\mathbf{D})_{t,j}(\mathbf{E}_r)_{i,j} \\ &= \sum\sum(\mathbf{G}_r\mathbf{G}_r^\top)_{s,i}(\mathbf{E}_r)_{i,j}(\mathbf{D}^\top[\mathbf{P}_r\mathbf{C}|\mathbf{C}_r]^\top[\mathbf{P}_r\mathbf{C}|\mathbf{C}_r]\mathbf{D})_{j,t}(\mathbf{E}_r)_{s,t}.\end{aligned} \tag{93}$$

Suppose the elements of $\mathbf{E}_r$ are related to that of $\widetilde{\mathbf{E}}_r$ with certain scale factors, i.e., $(\mathbf{E}_r)_{i,j} = z_{i,j}(\widetilde{\mathbf{E}}_r)_{i,j}$ where $z_{i,j}$ is a constant, then we have

$$\begin{aligned}&\sum\sum(\mathbf{G}_r\mathbf{G}_r^\top)_{i,s}(\mathbf{E}_r)_{s,t}(\mathbf{D}^\top[\mathbf{P}_r\mathbf{C}|\mathbf{C}_r]^\top[\mathbf{P}_r\mathbf{C}|\mathbf{C}_r]\mathbf{D})_{t,j}(\mathbf{E}_r)_{i,j} = \\ &\sum\sum(\mathbf{G}_r\mathbf{G}_r^\top)_{i,s}(\widetilde{\mathbf{E}}_r)_{s,t}(\mathbf{D}^\top[\mathbf{P}_r\mathbf{C}|\mathbf{C}_r]^\top[\mathbf{P}_r\mathbf{C}|\mathbf{C}_r]\mathbf{D})_{t,j}(\widetilde{\mathbf{E}}_r)_{i,j}z_{s,t}z_{i,j} \leq \\ &\frac{1}{2}\sum\sum(\mathbf{G}_r\mathbf{G}_r^\top)_{i,s}(\widetilde{\mathbf{E}}_r)_{s,t}(\mathbf{D}^\top[\mathbf{P}_r\mathbf{C}|\mathbf{C}_r]^\top[\mathbf{P}_r\mathbf{C}|\mathbf{C}_r]\mathbf{D})_{t,j}(\widetilde{\mathbf{E}}_r)_{i,j}\left(z_{s,t}^2+z_{i,j}^2\right)\end{aligned} \tag{94}$$

and

$$\begin{aligned}&\sum\sum(\mathbf{G}_r\mathbf{G}_r^\top)_{s,i}(\mathbf{E}_r)_{i,j}(\mathbf{D}^\top[\mathbf{P}_r\mathbf{C}|\mathbf{C}_r]^\top[\mathbf{P}_r\mathbf{C}|\mathbf{C}_r]\mathbf{D})_{j,t}(\mathbf{E}_r)_{s,t} = \\ &\sum\sum(\mathbf{G}_r\mathbf{G}_r^\top)_{s,i}(\widetilde{\mathbf{E}}_r)_{i,j}(\mathbf{D}^\top[\mathbf{P}_r\mathbf{C}|\mathbf{C}_r]^\top[\mathbf{P}_r\mathbf{C}|\mathbf{C}_r]\mathbf{D})_{j,t}(\widetilde{\mathbf{E}}_r)_{s,t}z_{i,j}z_{s,t} \leq \\ &\frac{1}{2}\sum\sum(\mathbf{G}_r\mathbf{G}_r^\top)_{s,i}(\widetilde{\mathbf{E}}_r)_{i,j}(\mathbf{D}^\top[\mathbf{P}_r\mathbf{C}|\mathbf{C}_r]^\top[\mathbf{P}_r\mathbf{C}|\mathbf{C}_r]\mathbf{D})_{j,t}(\widetilde{\mathbf{E}}_r)_{s,t}\left(z_{i,j}^2+z_{s,t}^2\right).\end{aligned} \tag{95}$$

Further, using element folding, we can get

$$\begin{aligned}&\sum\sum(\mathbf{G}_r\mathbf{G}_r^\top)_{s,i}(\widetilde{\mathbf{E}}_r)_{i,j}(\mathbf{D}^\top[\mathbf{P}_r\mathbf{C}|\mathbf{C}_r]^\top[\mathbf{P}_r\mathbf{C}|\mathbf{C}_r]\mathbf{D})_{j,t}(\widetilde{\mathbf{E}}_r)_{s,t}z_{s,t}^2 = \\ &\sum(\mathbf{G}_r\mathbf{G}_r^\top\widetilde{\mathbf{E}}_r\mathbf{D}^\top[\mathbf{P}_r\mathbf{C}|\mathbf{C}_r]^\top[\mathbf{P}_r\mathbf{C}|\mathbf{C}_r]\mathbf{D})_{s,t}(\widetilde{\mathbf{E}}_r)_{s,t}z_{s,t}^2 = \\ &\sum(\mathbf{G}_r\mathbf{G}_r^\top\widetilde{\mathbf{E}}_r\mathbf{D}^\top[\mathbf{P}_r\mathbf{C}|\mathbf{C}_r]^\top[\mathbf{P}_r\mathbf{C}|\mathbf{C}_r]\mathbf{D})_{s,t}(\mathbf{E}_r)_{s,t}z_{s,t} = \\ &\sum\frac{(\mathbf{G}_r\mathbf{G}_r^\top\widetilde{\mathbf{E}}_r\mathbf{D}^\top[\mathbf{P}_r\mathbf{C}|\mathbf{C}_r]^\top[\mathbf{P}_r\mathbf{C}|\mathbf{C}_r]\mathbf{D})_{s,t}(\mathbf{E}_r)_{s,t}^2}{(\widetilde{\mathbf{E}}_r)_{s,t}}\end{aligned} \tag{96}$$

and

$$\begin{aligned}&\sum\sum(\mathbf{G}_r\mathbf{G}_r^\top)_{i,s}(\widetilde{\mathbf{E}}_r)_{s,t}(\mathbf{D}^\top[\mathbf{P}_r\mathbf{C}|\mathbf{C}_r]^\top[\mathbf{P}_r\mathbf{C}|\mathbf{C}_r]\mathbf{D})_{t,j}(\widetilde{\mathbf{E}}_r)_{i,j}z_{i,j}^2 = \\ &\sum\frac{(\mathbf{G}_r\mathbf{G}_r^\top\widetilde{\mathbf{E}}_r\mathbf{D}^\top[\mathbf{P}_r\mathbf{C}|\mathbf{C}_r]^\top[\mathbf{P}_r\mathbf{C}|\mathbf{C}_r]\mathbf{D})_{i,j}(\mathbf{E}_r)_{i,j}^2}{(\widetilde{\mathbf{E}}_r)_{i,j}}.\end{aligned} \tag{97}$$

Additionally, for $\sum\sum(\mathbf{G}_r\mathbf{G}_r^\top)_{s,i}(\widetilde{\mathbf{E}}_r)_{i,j}(\mathbf{D}^\top[\mathbf{P}_r\mathbf{C}|\mathbf{C}_r]^\top[\mathbf{P}_r\mathbf{C}|\mathbf{C}_r]\mathbf{D})_{j,t}(\widetilde{\mathbf{E}}_r)_{s,t}z_{i,j}^2$ , in virtue of element transfer and

element folding, we can get

$$
\begin{aligned}
\sum\sum(\mathbf{G}_r\mathbf{G}_r^\top)_{s,i}(\widetilde{\mathbf{E}}_r)_{i,j}(\mathbf{D}^\top[\mathbf{P}_r\mathbf{C}|\mathbf{C}_r]^\top[\mathbf{P}_r\mathbf{C}|\mathbf{C}_r]\mathbf{D})_{j,t}(\widetilde{\mathbf{E}}_r)_{s,t}z_{i,j}^2 &= \\
\sum\sum(\mathbf{G}_r\mathbf{G}_r^\top)_{s,i}(\widetilde{\mathbf{E}}_r)_{s,t}(\mathbf{D}^\top[\mathbf{P}_r\mathbf{C}|\mathbf{C}_r]^\top[\mathbf{P}_r\mathbf{C}|\mathbf{C}_r]\mathbf{D})_{j,t}(\widetilde{\mathbf{E}}_r)_{i,j}z_{i,j}^2 &= \\
\sum\sum(\mathbf{G}_r\mathbf{G}_r^\top)_{i,s}(\widetilde{\mathbf{E}}_r)_{s,t}(\mathbf{D}^\top[\mathbf{P}_r\mathbf{C}|\mathbf{C}_r]^\top[\mathbf{P}_r\mathbf{C}|\mathbf{C}_r]\mathbf{D})_{t,j}(\widetilde{\mathbf{E}}_r)_{i,j}z_{i,j}^2 &= \\
\sum\sum(\mathbf{G}_r\mathbf{G}_r^\top)_{i,s}(\widetilde{\mathbf{E}}_r)_{s,t}(\mathbf{D}^\top[\mathbf{P}_r\mathbf{C}|\mathbf{C}_r]^\top[\mathbf{P}_r\mathbf{C}|\mathbf{C}_r]\mathbf{D})_{t,j}(\mathbf{E}_r)_{i,j}z_{i,j}. &
\end{aligned}
\tag{98}
$$

In conjunction with the element relationship between $\mathbf{E}_r$ and $\widetilde{\mathbf{E}}_r$, we can further have

$$
\begin{aligned}
\sum\sum(\mathbf{G}_r\mathbf{G}_r^\top)_{s,i}(\widetilde{\mathbf{E}}_r)_{i,j}(\mathbf{D}^\top[\mathbf{P}_r\mathbf{C}|\mathbf{C}_r]^\top[\mathbf{P}_r\mathbf{C}|\mathbf{C}_r]\mathbf{D})_{j,t}(\widetilde{\mathbf{E}}_r)_{s,t}z_{i,j}^2 &= \\
\sum\sum(\mathbf{G}_r\mathbf{G}_r^\top)_{i,s}(\widetilde{\mathbf{E}}_r)_{s,t}(\mathbf{D}^\top[\mathbf{P}_r\mathbf{C}|\mathbf{C}_r]^\top[\mathbf{P}_r\mathbf{C}|\mathbf{C}_r]\mathbf{D})_{t,j}\frac{(\mathbf{E}_r)_{i,j}^2}{(\widetilde{\mathbf{E}}_r)_{i,j}} &= \\
\sum(\mathbf{G}_r\mathbf{G}_r^\top\widetilde{\mathbf{E}}_r\mathbf{D}^\top[\mathbf{P}_r\mathbf{C}|\mathbf{C}_r]^\top[\mathbf{P}_r\mathbf{C}|\mathbf{C}_r]\mathbf{D})_{i,j}\frac{(\mathbf{E}_r)_{i,j}^2}{(\widetilde{\mathbf{E}}_r)_{i,j}}. &
\end{aligned}
\tag{99}
$$

Similarly, for the term $\sum\sum(\mathbf{G}_r\mathbf{G}_r^\top)_{i,s}(\widetilde{\mathbf{E}}_r)_{s,t}(\mathbf{D}^\top[\mathbf{P}_r\mathbf{C}|\mathbf{C}_r]^\top[\mathbf{P}_r\mathbf{C}|\mathbf{C}_r]\mathbf{D})_{t,j}(\widetilde{\mathbf{E}}_r)_{i,j}z_{s,t}^2$, we can get

$$
\begin{aligned}
\sum\sum(\mathbf{G}_r\mathbf{G}_r^\top)_{i,s}(\widetilde{\mathbf{E}}_r)_{s,t}(\mathbf{D}^\top[\mathbf{P}_r\mathbf{C}|\mathbf{C}_r]^\top[\mathbf{P}_r\mathbf{C}|\mathbf{C}_r]\mathbf{D})_{t,j}(\widetilde{\mathbf{E}}_r)_{i,j}z_{s,t}^2 &= \\
\sum(\mathbf{G}_r\mathbf{G}_r^\top\widetilde{\mathbf{E}}_r\mathbf{D}^\top[\mathbf{P}_r\mathbf{C}|\mathbf{C}_r]^\top[\mathbf{P}_r\mathbf{C}|\mathbf{C}_r]\mathbf{D})_{s,t}\frac{(\mathbf{E}_r)_{s,t}^2}{(\widetilde{\mathbf{E}}_r)_{s,t}}. &
\end{aligned}
\tag{100}
$$

Combined with Eqs. (93), (94), (95), (96), (97), (99) and (100), we have

$$
\begin{aligned}
\mathrm{Tr}\left(\mathbf{G}_r\mathbf{G}_r^\top\mathbf{E}_r\mathbf{D}^\top[\mathbf{P}_r\mathbf{C}|\mathbf{C}_r]^\top[\mathbf{P}_r\mathbf{C}|\mathbf{C}_r]\mathbf{D}\mathbf{E}_r^\top\right) &\leq \\
\frac{1}{4}\sum\sum(\mathbf{G}_r\mathbf{G}_r^\top)_{i,s}(\widetilde{\mathbf{E}}_r)_{s,t}(\mathbf{D}^\top[\mathbf{P}_r\mathbf{C}|\mathbf{C}_r]^\top[\mathbf{P}_r\mathbf{C}|\mathbf{C}_r]\mathbf{D})_{t,j}(\widetilde{\mathbf{E}}_r)_{i,j}\left(z_{s,t}^2+z_{i,j}^2\right) &+ \\
\frac{1}{4}\sum\sum(\mathbf{G}_r\mathbf{G}_r^\top)_{s,i}(\widetilde{\mathbf{E}}_r)_{i,j}(\mathbf{D}^\top[\mathbf{P}_r\mathbf{C}|\mathbf{C}_r]^\top[\mathbf{P}_r\mathbf{C}|\mathbf{C}_r]\mathbf{D})_{j,t}(\widetilde{\mathbf{E}}_r)_{s,t}\left(z_{i,j}^2+z_{s,t}^2\right) & \\
= \frac{1}{2}\sum(\mathbf{G}_r\mathbf{G}_r^\top\widetilde{\mathbf{E}}_r\mathbf{D}^\top[\mathbf{P}_r\mathbf{C}|\mathbf{C}_r]^\top[\mathbf{P}_r\mathbf{C}|\mathbf{C}_r]\mathbf{D})_{i,j}\frac{(\mathbf{E}_r)_{i,j}^2}{(\widetilde{\mathbf{E}}_r)_{i,j}} &+ \\
\frac{1}{2}\sum(\mathbf{G}_r\mathbf{G}_r^\top\widetilde{\mathbf{E}}_r\mathbf{D}^\top[\mathbf{P}_r\mathbf{C}|\mathbf{C}_r]^\top[\mathbf{P}_r\mathbf{C}|\mathbf{C}_r]\mathbf{D})_{s,t}\frac{(\mathbf{E}_r)_{s,t}^2}{(\widetilde{\mathbf{E}}_r)_{s,t}}. &
\end{aligned}
\tag{101}
$$

Consequently, we can obtain

$$
\mathrm{Tr}\left(\mathbf{G}_r\mathbf{G}_r^\top\mathbf{E}_r\mathbf{D}^\top[\mathbf{P}_r\mathbf{C}|\mathbf{C}_r]^\top[\mathbf{P}_r\mathbf{C}|\mathbf{C}_r]\mathbf{D}\mathbf{E}_r^\top\right) \leq \sum\frac{(\mathbf{G}_r\mathbf{G}_r^\top\widetilde{\mathbf{E}}_r\mathbf{D}^\top[\mathbf{P}_r\mathbf{C}|\mathbf{C}_r]^\top[\mathbf{P}_r\mathbf{C}|\mathbf{C}_r]\mathbf{D})_{i,j}(\mathbf{E}_r)_{i,j}^2}{(\widetilde{\mathbf{E}}_r)_{i,j}}.
\tag{102}
$$

For the term $\mathrm{Tr}\left(\mathbf{G}_r\Phi_r\mathbf{G}_r^\top\mathbf{E}_r\mathbf{1}_k\mathbf{1}_k^\top\mathbf{E}_r^\top\right)$ in Eq. (28), we have

$$
\mathrm{Tr}\left(\mathbf{G}_r\Phi_r\mathbf{G}_r^\top\mathbf{E}_r\mathbf{1}_k\mathbf{1}_k^\top\mathbf{E}_r^\top\right) = \sum\sum(\mathbf{G}_r\Phi_r\mathbf{G}_r^\top)_{i,t}(\mathbf{E}_r)_{t,l}(\mathbf{1}_k\mathbf{1}_k^\top)_{l,j}(\mathbf{E}_r)_{i,j} \leq \sum(\mathbf{G}_r\Phi_r\mathbf{G}_r^\top\widetilde{\mathbf{E}}_r\mathbf{1}_k\mathbf{1}_k^\top)_{i,j}\frac{(\mathbf{E}_r)_{i,j}^2}{(\widetilde{\mathbf{E}}_r)_{i,j}}.
\tag{103}
$$

Combined with Eqs. (90), (91), (92), (102) and (103), therefore, we can have

$$
\begin{aligned}
\mathcal{F}(\mathbf{E}_r, \widetilde{\mathbf{E}}_r) =& a_r^2 \sum (\mathbf{G}_r \mathbf{G}_r^\top \widetilde{\mathbf{E}}_r \mathbf{D}^\top [\mathbf{P}_r \mathbf{C} | \mathbf{C}_r]^\top [\mathbf{P}_r \mathbf{C} | \mathbf{C}_r] \mathbf{D})_{i,j} \frac{(\mathbf{E}_r)_{i,j}^2}{(\widetilde{\mathbf{E}}_r)_{i,j}} \\
& - 2a_r^2 \sum (\widetilde{\mathbf{E}}_r)_{i,j} (\mathbf{G}_r \mathbf{G}_r^\top \mathbf{X}_r^\top [\mathbf{P}_r \mathbf{C} | \mathbf{C}_r] \mathbf{D})_{i,j} \left( 1 + \log \frac{(\mathbf{E}_r)_{i,j}}{(\widetilde{\mathbf{E}}_r)_{i,j}} \right) \\
& - \beta b_r \sum (\widetilde{\mathbf{E}}_r)_{i,j} \left( \left(\mathbf{E}\mathbf{F}_r^\top\right)_{pos} \right)_{i,j} \left( 1 + \log \frac{(\mathbf{E}_r)_{i,j}}{(\widetilde{\mathbf{E}}_r)_{i,j}} \right) \\
& + \frac{\beta b_r}{2} \sum \left( \left(\mathbf{E}\mathbf{F}_r^\top\right)_{neg} \right)_{i,j} \frac{(\widetilde{\mathbf{E}}_r)_{i,j}^2 + (\mathbf{E}_r)_{i,j}^2}{(\widetilde{\mathbf{E}}_r)_{i,j}} \\
& + \sum (\mathbf{G}_r \Phi_r \mathbf{G}_r^\top \widetilde{\mathbf{E}}_r \mathbf{1}_k \mathbf{1}_k^\top)_{i,j} \frac{(\mathbf{E}_r)_{i,j}^2}{(\widetilde{\mathbf{E}}_r)_{i,j}} \\
& - 2 \sum (\widetilde{\mathbf{E}}_r)_{i,j} (\mathbf{G}_r \Phi_r^\top \mathbf{1}_{n_r} \mathbf{1}_k^\top)_{i,j} \left( 1 + \log \frac{(\mathbf{E}_r)_{i,j}}{(\widetilde{\mathbf{E}}_r)_{i,j}} \right).
\end{aligned}
\tag{104}
$$

According to the bound of each branch, we have that $\mathcal{F}(\mathbf{E}_r, \widetilde{\mathbf{E}}_r) \geq \mathcal{L}(\mathbf{E}_r)$ and $\mathcal{F}(\mathbf{E}_r, \mathbf{E}_r) = \mathcal{L}(\mathbf{E}_r)$. Therefore, it is an auxiliary function of $\mathcal{L}(\mathbf{E}_r)$. Further, its Hessian matrix is semi-positive, and consequently it is convex. The global solution can be acquired by setting its derivative equaling to zero. That is,

$$
(\mathbf{E}_r)_{i,j} = \left[ \frac{\left( a_r^2 \mathbf{G}_r \mathbf{G}_r^\top \mathbf{X}_r^\top [\mathbf{P}_r \mathbf{C} | \mathbf{C}_r] \mathbf{D} + \frac{\beta}{2} b_r \left(\mathbf{E}\mathbf{F}_r^\top\right)_{pos} + \mathbf{G}_r \Phi_r \mathbf{1}_{n_r} \mathbf{1}_k^\top \right)_{i,j}}{\left( a_r^2 \mathbf{G}_r \mathbf{G}_r^\top \widetilde{\mathbf{E}}_r \mathbf{D}^\top [\mathbf{P}_r \mathbf{C} | \mathbf{C}_r]^\top [\mathbf{P}_r \mathbf{C} | \mathbf{C}_r] \mathbf{D} + \frac{\beta}{2} b_r \left(\mathbf{E}\mathbf{F}_r^\top\right)_{neg} + \mathbf{G}_r \Phi_r \mathbf{G}_r^\top \widetilde{\mathbf{E}}_r \mathbf{1}_k \mathbf{1}_k^\top \right)_{i,j}} \right]^{\frac{1}{2}}.
\tag{105}
$$

After eliminating $\Phi_r$, we can equivalently obtain the updating rule Eq. (8).

$\square$

## C. Proof of Theorem 2

*Proof.* Let $\mathbf{E}_r^{(h+1)} = \arg \min_{\mathbf{E}_r} \mathcal{F}(\mathbf{E}_r, \mathbf{E}_r^{(h)})$, and then we have $\mathcal{F}(\mathbf{E}_r^{(h+1)}, \mathbf{E}_r^{(h)}) \leq \mathcal{F}(\mathbf{E}_r^{(h)}, \mathbf{E}_r^{(h)})$. In conjunction with **Theorem** 1, we have that $\mathcal{F}(\mathbf{E}_r, \mathbf{E}_r^{(h)})$ is convex and the global optimum can be achieved. Further, combined with the definition of auxiliary function, we have

$$
\mathcal{F}(\mathbf{E}_r^{(h)}, \mathbf{E}_r^{(h)}) = \mathcal{L}(\mathbf{E}_r^{(h)})
\tag{106}
$$

and

$$
\mathcal{F}(\mathbf{E}_r^{(h+1)}, \mathbf{E}_r^{(h)}) \geq \mathcal{L}(\mathbf{E}_r^{(h+1)}).
\tag{107}
$$

Therefore, we can obtain

$$
\mathcal{L}(\mathbf{E}_r^{(h+1)}) \leq \mathcal{L}(\mathbf{E}_r^{(h)}),
\tag{108}
$$

which demonstrates that it is steadily decreasing.

Accordingly, we have the following inequality holds,

$$
\mathcal{J}\left(\mathbf{E}_r^{(h+1)}, \mathbf{C}^{(h)}, \mathbf{C}_r^{(h)}, \mathbf{D}^{(h)}, \mathbf{P}_r^{(h)}, \mathbf{E}^{(h)}, \mathbf{F}_r^{(h)}, a_r^{(h)}, b_r^{(h)}\right) \leq \mathcal{J}\left(\mathbf{E}_r^{(h)}, \mathbf{C}^{(h)}, \mathbf{C}_r^{(h)}, \mathbf{D}^{(h)}, \mathbf{P}_r^{(h)}, \mathbf{E}^{(h)}, \mathbf{F}_r^{(h)}, a_r^{(h)}, b_r^{(h)}\right),
\tag{109}
$$

where $\mathcal{J}$ is the objective value of Eq. (6).

$\square$

## D. Proof of Theorem 3

*Proof.* For the $j$-th column of $\mathbf{P}_r\mathbf{C}$, we have

$$\sum_{i=1}^{n}(\mathbf{P}_r\mathbf{C})_{i,j} = \sum_{i=1}^{n}\left(\sum_{l=1}^{k}(\mathbf{P}_r)_{i,l}\,\mathbf{C}_{l,j}\right). \tag{110}$$

After exchanging the order of summation, we can obtain

$$\sum_{i=1}^{n}(\mathbf{P}_r\mathbf{C})_{i,j} = \sum_{l=1}^{k}\left(\sum_{i=1}^{n}(\mathbf{P}_r)_{i,l}\right)\mathbf{C}_{l,j}. \tag{111}$$

Since $\mathbf{P}_r$ is column-normalized, we can get that $\sum_{i=1}^{n}(\mathbf{P_r})_{i,l}=1$ holds for any $l \in \{1,2,\cdots,k\}$. Therefore, we have

$$\sum_{i=1}^{n}(\mathbf{P}_r\mathbf{C})_{i,j} = \sum_{l=1}^{k}\mathbf{C}_{l,j}. \tag{112}$$

So, to ensure that the sum of the elements in each column of $\mathbf{P}_r\mathbf{C}$ is equal to 1, under column-normalized $\mathbf{P}_r$, we only need

$$\sum_{l=1}^{k}\mathbf{C}_{l,j} = 1, \quad j = [1,2,\cdots,k]. \tag{113}$$

That is, $\mathbf{C}$ also needs to be column-normalized. $\qquad\square$

## E. Proof of Theorem 4

*Proof.* For the objective function, according to $F$-norm characteristic, we have $\left\|\mathbf{X}_r\mathbf{G}_r - [\mathbf{P}_r\mathbf{C}|\mathbf{C}_r]\mathbf{D}\mathbf{E}_r^\top\mathbf{G}_r\right\|_F^2 \geq 0$. Combined with the non-negative property of $\mathbf{C}$ and $\mathbf{C}_r$ as well as $\mathbf{P}_r$, we can obtain $\langle\mathbf{P}_r\mathbf{C},\mathbf{C}_r\rangle \geq 0$. Therefore, we have that the objective function is lower-bounded by $-\beta\,\mathrm{Tr}\left(\mathbf{E}^\top\sum_{r=1}^{v}b_r\mathbf{E}_r\mathbf{F}_r\right)$.

Finding the lower bound of $-\,\mathrm{Tr}\left(\mathbf{E}^\top\sum_{r=1}^{v}b_r\mathbf{E}_r\mathbf{F}_r\right)$ is equivalently to find the upper bound of $\mathrm{Tr}\left(\mathbf{E}^\top\sum_{r=1}^{v}b_r\mathbf{E}_r\mathbf{F}_r\right)$. To obtain one upper bound of $\mathrm{Tr}\left(\mathbf{E}^\top\sum_{r=1}^{v}b_r\mathbf{E}_r\mathbf{F}_r\right)$, firstly, based on the linearity property of trace operator, we can obtain

$$\mathrm{Tr}\left(\mathbf{E}^\top\sum_{r=1}^{v}b_r\mathbf{E}_r\mathbf{F}_r\right) = \sum_{r=1}^{v}b_r\,\mathrm{Tr}\left(\mathbf{E}^\top\mathbf{E}_r\mathbf{F}_r\right). \tag{114}$$

Then, utilizing the cyclic property of trace, we can get

$$\mathrm{Tr}\left(\mathbf{E}^\top\mathbf{E}_r\mathbf{F}_r\right) = \mathrm{Tr}\left(\mathbf{F}_r\mathbf{E}^\top\mathbf{E}_r\right). \tag{115}$$

The trace of a product of matrices is maximized when the matrices are aligned in a way that maximizes the sum of their diagonal elements. Combined with the fact that $\mathbf{E}$ and $\mathbf{E}_r$ are orthogonal, therefore, the maximum value of $\mathrm{Tr}\left(\mathbf{F}_r\mathbf{E}^\top\mathbf{E}_r\right)$ occurs when $\mathbf{E}_r$ is aligned with $\mathbf{E}$ and $\mathbf{F}_r$.

To demonstrate this point from theory, we firstly decompose $\mathbf{E}_r$ as $\mathbf{U}\mathbf{\Sigma}\mathbf{V}^\top$ where $\mathbf{U}$ and $\mathbf{V}$ are the left singular matrix and right singular matrix respectively, and $\mathbf{\Sigma}$ is the singular value matrix with elements greater than or equal to zero. Then we have $\mathrm{Tr}\left(\mathbf{F}_r\mathbf{E}^\top\mathbf{E}_r\right) = \mathrm{Tr}\left(\mathbf{F}_r\mathbf{E}^\top\mathbf{U}\mathbf{\Sigma}\mathbf{V}^\top\right)$. Further, via cyclic operation, we have

$$\mathrm{Tr}\left(\mathbf{F}_r\mathbf{E}^\top\mathbf{E}_r\right) = \mathrm{Tr}\left(\mathbf{V}^\top\mathbf{F}_r\mathbf{E}^\top\mathbf{U}\mathbf{\Sigma}\right). \tag{116}$$

Kindly note that $\mathbf{V}$, $\mathbf{F}_r$, $\mathbf{E}$ and $\mathbf{U}$ are all orthogonal matrices, accordingly, $\mathbf{V}^\top\mathbf{F}_r\mathbf{E}^\top\mathbf{U}$ is also an orthogonal matrix. Therefore, we have

$$\mathrm{Tr}\left(\mathbf{F}_r\mathbf{E}^\top\mathbf{E}_r\right) \leq \mathrm{Tr}\left(\mathbf{\Sigma}\right), \tag{117}$$

where the equality holds if and only if $\mathbf{V}^\top\mathbf{F}_r\mathbf{E}^\top\mathbf{U}$ equals to an identity matrix.

In conjunction with $\text{Tr}(\boldsymbol{\Sigma}) = \sum_{i=1}^{\min\{n,k\}} \sigma_i(\mathbf{E}_r)$ where $\sigma_i(\mathbf{E}_r)$ denotes the $i$-th singular value of $\mathbf{E}_r$, we have

$$\text{Tr}(\boldsymbol{\Sigma}) \leq \left[ \min\{n,k\} \sum_{i=1}^{\min\{n,k\}} \sigma_i^2(\mathbf{E}_r) \right]^{\frac{1}{2}}. \tag{118}$$

Then, given the fact that $\sigma_i^2(\mathbf{E}_r)$ is equal to $\lambda_i(\mathbf{E}_r^\top \mathbf{E}_r)$ where $\lambda_i(\mathbf{E}_r^\top \mathbf{E}_r)$ denotes the $i$-th eigenvalue of $\mathbf{E}_r^\top \mathbf{E}_r$ and that $\sum \lambda_i(\mathbf{E}_r^\top \mathbf{E}_r)$ is equal to $\text{Tr}(\mathbf{E}_r^\top \mathbf{E}_r)$, we have

$$\text{Tr}(\mathbf{E}_r^\top \mathbf{E}_r) = \sum_{i=1}^{\min\{n,k\}} \sigma_i^2(\mathbf{E}_r). \tag{119}$$

Further, considering that $\|\mathbf{E}_r\|_F^2$ is equal to $\text{Tr}(\mathbf{E}_r^\top \mathbf{E}_r)$ and that $\mathbf{E}_r$ is row-normalized and non-negative, we have that $\sum_{j=1}^k (\mathbf{E}_r)_{i,j}^2$ is less than or equal to 1 for any $i = [1, 2, \cdots, n]$. Consequently, we can obtain

$$\text{Tr}(\mathbf{E}_r^\top \mathbf{E}_r) \leq n, \tag{120}$$

where the equality holds if and only if each row of $\mathbf{E}_r$ is a one-hot vector. It can be further derived that at this point $\mathbf{E}_r$ is a (one-sided) orthogonal matrix.

In conjunction with Eqs. (117), (118), (119) and (120), consequently, we can obtain

$$\text{Tr}\left(\mathbf{F}_r \mathbf{E}^\top \mathbf{E}_r\right) \leq \sqrt{\min\{n,k\}n}. \tag{121}$$

Since the sample size $n$ is largely greater than the cluster number $k$, we have

$$\text{Tr}\left(\mathbf{F}_r \mathbf{E}^\top \mathbf{E}_r\right) \leq \sqrt{nk}. \tag{122}$$

Combining Eqs. (114), (115) and (122) yields

$$\text{Tr}\left(\mathbf{E}^\top \sum_{r=1}^v b_r \mathbf{E}_r \mathbf{F}_r\right) \leq \sqrt{nk} \sum_{r=1}^v b_r. \tag{123}$$

Afterwards, using Cauchy inequality, we can get

$$\left(\sum_{r=1}^v b_r\right)^2 \leq \left(\sum_{r=1}^v b_r^2\right) \left(\sum_{r=1}^v 1^2\right). \tag{124}$$

Accordingly, combining Eqs. (123) and (124) yields

$$\text{Tr}\left(\mathbf{E}^\top \sum_{r=1}^v b_r \mathbf{E}_r \mathbf{F}_r\right) \leq \sqrt{vnk}. \tag{125}$$

Therefore, the objective value is lower-bounded by $(-\beta\sqrt{vnk})$. That is,

$$\sum_{r=1}^v a_r^2 \left\|\mathbf{X}_r \mathbf{G}_r - [\mathbf{P}_r \mathbf{C}|\mathbf{C}_r]\mathbf{D}\mathbf{E}_r^\top \mathbf{G}_r\right\|_F^2 + \lambda\langle \mathbf{P}_r \mathbf{C}, \mathbf{C}_r\rangle - \beta \text{Tr}\left(\mathbf{E}^\top \sum_{r=1}^v b_r \mathbf{E}_r \mathbf{F}_r\right) \geq -\beta\sqrt{vnk}. \tag{126}$$

$\square$

## F. Proof of Theorem 5

*Proof.* ▶ During optimizing $\mathbf{C}$, based on Eq. (39), we can equivalently rewrite its Lagrange function as

$$
\begin{aligned}
\mathcal{L}\left(\mathbf{C}, \Psi\right) = \operatorname{Tr}\Bigg( & \sum_{r=1}^{v} a_r^2 \mathbf{P}_r^\top \mathbf{P}_r \mathbf{C} \mathbf{D}_\gamma \mathbf{E}_r^\top \mathbf{G}_r \mathbf{G}_r^\top \mathbf{E}_r \mathbf{D}_\gamma^\top \mathbf{C}^\top - 2\sum_{r=1}^{v} a_r^2 \mathbf{P}_r^\top \mathbf{X}_r \mathbf{G}_r \mathbf{G}_r^\top \mathbf{E}_r \mathbf{D}_\gamma^\top \mathbf{C}^\top + \lambda \sum_{r=1}^{v} \mathbf{P}_r^\top \mathbf{C}_r \mathbf{C}^\top + \\
& 2\sum_{r=1}^{v} a_r^2 \mathbf{P}_r^\top \mathbf{C}_r \mathbf{D}_\varphi \mathbf{E}_r^\top \mathbf{G}_r \mathbf{G}_r^\top \mathbf{E}_r \mathbf{D}_\gamma^\top \mathbf{C}^\top + \sum_{r=1}^{v} \mathbf{P}_r^\top \mathbf{1}_{d_r} \mathbf{1}_{d_r}^\top \mathbf{P}_r \mathbf{C} \Psi_r \mathbf{C}^\top - 2\sum_{r=1}^{v} \mathbf{P}_r^\top \mathbf{1}_{d_r} \mathbf{1}_k^\top \Psi_r \mathbf{C}^\top \Bigg).
\end{aligned}
\tag{127}
$$

For the matrix $\sum_{r=1}^{v} a_r^2 \mathbf{P}_r^\top \mathbf{X}_r \mathbf{G}_r \mathbf{G}_r^\top \mathbf{E}_r \mathbf{D}_\gamma^\top \mathbf{C}^\top$, combined with the property of each matrix, we can get that it is element-wisely non-negative. Meanwhile, unfolding the trace by element, we have

$$
\begin{aligned}
\operatorname{Tr}\left( \sum_{r=1}^{v} a_r^2 \mathbf{P}_r^\top \mathbf{X}_r \mathbf{G}_r \mathbf{G}_r^\top \mathbf{E}_r \mathbf{D}_\gamma^\top \mathbf{C}^\top \right) &= \sum \left( \sum_{r=1}^{v} a_r^2 \mathbf{P}_r^\top \mathbf{X}_r \mathbf{G}_r \mathbf{G}_r^\top \mathbf{E}_r \mathbf{D}_\gamma^\top \right)_{i,j} \mathbf{C}_{i,j} \\
&\geq \sum \left( \sum_{r=1}^{v} a_r^2 \mathbf{P}_r^\top \mathbf{X}_r \mathbf{G}_r \mathbf{G}_r^\top \mathbf{E}_r \mathbf{D}_\gamma^\top \right)_{i,j} \widetilde{\mathbf{C}}_{i,j} \left( 1 + \log \frac{\mathbf{C}_{i,j}}{\widetilde{\mathbf{C}}_{i,j}} \right).
\end{aligned}
\tag{128}
$$

For the term $\operatorname{Tr}(\mathbf{P}_r^\top \mathbf{1}_{d_r} \mathbf{1}_k^\top \Psi_r \mathbf{C}^\top)$, similarly, we can obtain the following inequality,

$$
\operatorname{Tr}\left( \mathbf{P}_r^\top \mathbf{1}_{d_r} \mathbf{1}_k^\top \Psi_r \mathbf{C}^\top \right) \geq \sum \left( \mathbf{P}_r^\top \mathbf{1}_{d_r} \mathbf{1}_k^\top \Psi_r \right)_{i,j} \widetilde{\mathbf{C}}_{i,j} \left( 1 + \log \frac{\mathbf{C}_{i,j}}{\widetilde{\mathbf{C}}_{i,j}} \right).
\tag{129}
$$

Then, for the term $\operatorname{Tr}\left( \sum_{r=1}^{v} \mathbf{P}_r^\top \mathbf{C}_r \mathbf{C}^\top \right)$, we can get

$$
\operatorname{Tr}\left( \sum_{r=1}^{v} \mathbf{P}_r^\top \mathbf{C}_r \mathbf{C}^\top \right) \leq \frac{1}{2} \sum \left( \sum_{r=1}^{v} \mathbf{P}_r^\top \mathbf{C}_r \right)_{i,j} \frac{\mathbf{C}_{i,j}^2 + \widetilde{\mathbf{C}}_{i,j}^2}{\widetilde{\mathbf{C}}_{i,j}}.
\tag{130}
$$

Accordingly, for the term $\operatorname{Tr}(\sum_{r=1}^{v} a_r^2 \mathbf{P}_r^\top \mathbf{C}_r \mathbf{D}_\varphi \mathbf{E}_r^\top \mathbf{G}_r \mathbf{G}_r^\top \mathbf{E}_r \mathbf{D}_\gamma^\top \mathbf{C}^\top)$, we have

$$
\operatorname{Tr}\left( \sum_{r=1}^{v} a_r^2 \mathbf{P}_r^\top \mathbf{C}_r \mathbf{D}_\varphi \mathbf{E}_r^\top \mathbf{G}_r \mathbf{G}_r^\top \mathbf{E}_r \mathbf{D}_\gamma^\top \mathbf{C}^\top \right) \leq \frac{1}{2} \sum \left( \sum_{r=1}^{v} a_r^2 \mathbf{P}_r^\top \mathbf{C}_r \mathbf{D}_\varphi \mathbf{E}_r^\top \mathbf{G}_r \mathbf{G}_r^\top \mathbf{E}_r \mathbf{D}_\gamma^\top \right)_{i,j} \frac{\mathbf{C}_{i,j}^2 + \widetilde{\mathbf{C}}_{i,j}^2}{\widetilde{\mathbf{C}}_{i,j}}.
\tag{131}
$$

Afterwards, for the matrix $\mathbf{P}_r^\top \mathbf{P}_r \mathbf{C} \mathbf{D}_\gamma \mathbf{E}_r^\top \mathbf{G}_r \mathbf{G}_r^\top \mathbf{E}_r \mathbf{D}_\gamma^\top \mathbf{C}^\top$, suppose $\mathbf{C}_{s,j} = f_{s,j} \widetilde{\mathbf{C}}_{s,j}$ for any $s$ and $j$ where $f_{s,j}$ is a constant, then we have

$$
\begin{aligned}
\operatorname{Tr}\left( \mathbf{P}_r^\top \mathbf{P}_r \mathbf{C} \mathbf{D}_\gamma \mathbf{E}_r^\top \mathbf{G}_r \mathbf{G}_r^\top \mathbf{E}_r \mathbf{D}_\gamma^\top \mathbf{C}^\top \right) &= \sum\sum \left( \mathbf{P}_r^\top \mathbf{P}_r \right)_{s,i} \mathbf{C}_{i,t} \left( \mathbf{D}_\gamma \mathbf{E}_r^\top \mathbf{G}_r \mathbf{G}_r^\top \mathbf{E}_r \mathbf{D}_\gamma^\top \right)_{t,j} \mathbf{C}_{s,j} \\
&= \sum\sum \left( \mathbf{P}_r^\top \mathbf{P}_r \right)_{s,i} \widetilde{\mathbf{C}}_{i,t} \left( \mathbf{D}_\gamma \mathbf{E}_r^\top \mathbf{G}_r \mathbf{G}_r^\top \mathbf{E}_r \mathbf{D}_\gamma^\top \right)_{t,j} \widetilde{\mathbf{C}}_{s,j} f_{i,t} f_{s,j} \\
&\leq \frac{1}{2} \sum\sum \left( \mathbf{P}_r^\top \mathbf{P}_r \right)_{s,i} \widetilde{\mathbf{C}}_{i,t} \left( \mathbf{D}_\gamma \mathbf{E}_r^\top \mathbf{G}_r \mathbf{G}_r^\top \mathbf{E}_r \mathbf{D}_\gamma^\top \right)_{t,j} \widetilde{\mathbf{C}}_{s,j} \left( f_{i,t}^2 + f_{s,j}^2 \right).
\end{aligned}
\tag{132}
$$

Subsequently, based on the commutative property and symmetry, we can get

$$
\begin{aligned}
\sum\sum \left( \mathbf{P}_r^\top \mathbf{P}_r \right)_{s,i} \widetilde{\mathbf{C}}_{i,t} \left( \mathbf{D}_\gamma \mathbf{E}_r^\top \mathbf{G}_r \mathbf{G}_r^\top \mathbf{E}_r \mathbf{D}_\gamma^\top \right)_{t,j} \widetilde{\mathbf{C}}_{s,j} f_{i,t}^2 &= \sum\sum \left( \mathbf{P}_r^\top \mathbf{P}_r \right)_{s,i} \widetilde{\mathbf{C}}_{s,j} \left( \mathbf{D}_\gamma \mathbf{E}_r^\top \mathbf{G}_r \mathbf{G}_r^\top \mathbf{E}_r \mathbf{D}_\gamma^\top \right)_{j,t} \widetilde{\mathbf{C}}_{i,t} f_{i,t}^2 \\
&= \sum \left( \mathbf{P}_r^\top \mathbf{P}_r \widetilde{\mathbf{C}} \mathbf{D}_\gamma \mathbf{E}_r^\top \mathbf{G}_r \mathbf{G}_r^\top \mathbf{E}_r \mathbf{D}_\gamma^\top \right)_{i,t} \widetilde{\mathbf{C}}_{i,t} f_{i,t}^2 \\
&= \sum \left( \mathbf{P}_r^\top \mathbf{P}_r \widetilde{\mathbf{C}} \mathbf{D}_\gamma \mathbf{E}_r^\top \mathbf{G}_r \mathbf{G}_r^\top \mathbf{E}_r \mathbf{D}_\gamma^\top \right)_{i,t} \frac{\mathbf{C}_{i,t}^2}{\widetilde{\mathbf{C}}_{i,t}}
\end{aligned}
\tag{133}
$$

and

$$\sum\sum \left(\mathbf{P}_r^\top \mathbf{P}_r\right)_{s,i} \widetilde{\mathbf{C}}_{i,t} \left(\mathbf{D}_\gamma \mathbf{E}_r^\top \mathbf{G}_r \mathbf{G}_r^\top \mathbf{E}_r \mathbf{D}_\gamma^\top\right)_{t,j} \widetilde{\mathbf{C}}_{s,j} f_{s,j}^2 = \sum \left(\mathbf{P}_r^\top \mathbf{P}_r \widetilde{\mathbf{C}} \mathbf{D}_\gamma \mathbf{E}_r^\top \mathbf{G}_r \mathbf{G}_r^\top \mathbf{E}_r \mathbf{D}_\gamma^\top\right)_{s,j} \frac{\mathbf{C}_{s,j}^2}{\widetilde{\mathbf{C}}_{s,j}}. \quad (134)$$

Consequently, combined with Eqs. (132), (133) and (134), we can obtain the following inequality,

$$\mathrm{Tr}\left(\sum_{r=1}^v a_r^2 \mathbf{P}_r^\top \mathbf{P}_r \mathbf{C} \mathbf{D}_\gamma \mathbf{E}_r^\top \mathbf{G}_r \mathbf{G}_r^\top \mathbf{E}_r \mathbf{D}_\gamma^\top \mathbf{C}^\top\right) \leq \sum_{r=1}^v a_r^2 \sum \left(\mathbf{P}_r^\top \mathbf{P}_r \widetilde{\mathbf{C}} \mathbf{D}_\gamma \mathbf{E}_r^\top \mathbf{G}_r \mathbf{G}_r^\top \mathbf{E}_r \mathbf{D}_\gamma^\top\right)_{i,j} \frac{\mathbf{C}_{i,j}^2}{\widetilde{\mathbf{C}}_{i,j}}. \quad (135)$$

For $\mathrm{Tr}\left(\mathbf{P}_r^\top \mathbf{1}_{d_r} \mathbf{1}_{d_r}^\top \mathbf{P}_r \mathbf{C} \Psi_r \mathbf{C}^\top\right)$, after element unfolding, we have

$$\begin{aligned}
\mathrm{Tr}\left(\mathbf{P}_r^\top \mathbf{1}_{d_r} \mathbf{1}_{d_r}^\top \mathbf{P}_r \mathbf{C} \Psi_r \mathbf{C}^\top\right) &= \sum\sum \left(\mathbf{P}_r^\top \mathbf{1}_{d_r} \mathbf{1}_{d_r}^\top \mathbf{P}_r\right)_{i,t} \mathbf{C}_{t,l}(\Psi_r)_{l,j}\mathbf{C}_{i,j} \\
&= \sum\sum \left(\mathbf{P}_r^\top \mathbf{1}_{d_r} \mathbf{1}_{d_r}^\top \mathbf{P}_r\right)_{i,t} \widetilde{\mathbf{C}}_{t,l}(\Psi_r)_{l,j}\widetilde{\mathbf{C}}_{i,j} f_{t,l} f_{i,j} \\
&\leq \frac{1}{2}\sum\sum \left(\mathbf{P}_r^\top \mathbf{1}_{d_r} \mathbf{1}_{d_r}^\top \mathbf{P}_r\right)_{i,t} \widetilde{\mathbf{C}}_{t,l}(\Psi_r)_{l,j}\widetilde{\mathbf{C}}_{i,j} \left(f_{i,j}^2 + f_{t,l}^2\right).
\end{aligned} \quad (136)$$

For the first term, we can get

$$\sum\sum \left(\mathbf{P}_r^\top \mathbf{1}_{d_r} \mathbf{1}_{d_r}^\top \mathbf{P}_r\right)_{i,t} \widetilde{\mathbf{C}}_{t,l}(\Psi_r)_{l,j}\widetilde{\mathbf{C}}_{i,j} f_{i,j}^2 = \sum \left(\mathbf{P}_r^\top \mathbf{1}_{d_r} \mathbf{1}_{d_r}^\top \mathbf{P}_r \widetilde{\mathbf{C}} \Psi_r\right)_{i,j} \frac{\mathbf{C}_{i,j}^2}{\widetilde{\mathbf{C}}_{i,j}}. \quad (137)$$

Further, in conjunction with the fact that $\Psi_r$ is a diagonal matrix, we can derive the following equality,

$$\begin{aligned}
\sum\sum \left(\mathbf{P}_r^\top \mathbf{1}_{d_r} \mathbf{1}_{d_r}^\top \mathbf{P}_r\right)_{i,t} \widetilde{\mathbf{C}}_{t,l}(\Psi_r)_{l,j}\widetilde{\mathbf{C}}_{i,j} f_{t,l}^2 &= \sum\sum \left(\mathbf{P}_r^\top \mathbf{1}_{d_r} \mathbf{1}_{d_r}^\top \mathbf{P}_r\right)_{t,i} \widetilde{\mathbf{C}}_{t,l}(\Psi_r)_{j,l}\widetilde{\mathbf{C}}_{i,j} f_{t,l}^2 \\
&= \sum\sum \left(\mathbf{P}_r^\top \mathbf{1}_{d_r} \mathbf{1}_{d_r}^\top \mathbf{P}_r\right)_{t,i} \widetilde{\mathbf{C}}_{i,j}(\Psi_r)_{j,l}\widetilde{\mathbf{C}}_{t,l} f_{t,l}^2 \\
&= \sum \left(\mathbf{P}_r^\top \mathbf{1}_{d_r} \mathbf{1}_{d_r}^\top \mathbf{P}_r \widetilde{\mathbf{C}} \Psi_r\right)_{t,l} \widetilde{\mathbf{C}}_{t,l} f_{t,l}^2 \\
&= \sum \left(\mathbf{P}_r^\top \mathbf{1}_{d_r} \mathbf{1}_{d_r}^\top \mathbf{P}_r \widetilde{\mathbf{C}} \Psi_r\right)_{t,l} \frac{\mathbf{C}_{t,l}^2}{\widetilde{\mathbf{C}}_{t,l}}.
\end{aligned} \quad (138)$$

Combined with Eqs. (136), (137) and (138), we can get

$$\mathrm{Tr}\left(\mathbf{P}_r^\top \mathbf{1}_{d_r} \mathbf{1}_{d_r}^\top \mathbf{P}_r \mathbf{C} \Psi_r \mathbf{C}^\top\right) \leq \sum \left(\mathbf{P}_r^\top \mathbf{1}_{d_r} \mathbf{1}_{d_r}^\top \mathbf{P}_r \widetilde{\mathbf{C}} \Psi_r\right)_{i,j} \frac{\mathbf{C}_{i,j}^2}{\widetilde{\mathbf{C}}_{i,j}}. \quad (139)$$

So, at this point, according to Eqs. (128), (129), (130), (131), (135) and (139), we can have

$$\begin{aligned}
\mathcal{F}\left(\mathbf{C}, \widetilde{\mathbf{C}}\right) =& \sum_{r=1}^v a_r^2 \sum \left(\mathbf{P}_r^\top \mathbf{P}_r \widetilde{\mathbf{C}} \mathbf{D}_\gamma \mathbf{E}_r^\top \mathbf{G}_r \mathbf{G}_r^\top \mathbf{E}_r \mathbf{D}_\gamma^\top\right)_{i,j} \frac{\mathbf{C}_{i,j}^2}{\widetilde{\mathbf{C}}_{i,j}} \\
&- 2\sum \left(\sum_{r=1}^v a_r^2 \mathbf{P}_r^\top \mathbf{X}_r \mathbf{G}_r \mathbf{G}_r^\top \mathbf{E}_r \mathbf{D}_\gamma^\top\right)_{i,j} \widetilde{\mathbf{C}}_{i,j} \left(1 + \log \frac{\mathbf{C}_{i,j}}{\widetilde{\mathbf{C}}_{i,j}}\right) \\
&+ \sum \left(\sum_{r=1}^v a_r^2 \mathbf{P}_r^\top \mathbf{C}_r \mathbf{D}_\varphi \mathbf{E}_r^\top \mathbf{G}_r \mathbf{G}_r^\top \mathbf{E}_r \mathbf{D}_\gamma^\top\right)_{i,j} \frac{\mathbf{C}_{i,j}^2 + \widetilde{\mathbf{C}}_{i,j}^2}{\widetilde{\mathbf{C}}_{i,j}} \\
&- 2\sum_{r=1}^v \sum \left(\mathbf{P}_r^\top \mathbf{1}_{d_r} \mathbf{1}_k^\top \Psi_r\right)_{i,j} \widetilde{\mathbf{C}}_{i,j} \left(1 + \log \frac{\mathbf{C}_{i,j}}{\widetilde{\mathbf{C}}_{i,j}}\right) \\
&+ \sum_{r=1}^v \sum \left(\mathbf{P}_r^\top \mathbf{1}_{d_r} \mathbf{1}_{d_r}^\top \mathbf{P}_r \widetilde{\mathbf{C}} \Psi_r\right)_{i,j} \frac{\mathbf{C}_{i,j}^2}{\widetilde{\mathbf{C}}_{i,j}} \\
&+ \frac{\lambda}{2}\sum \left(\sum_{r=1}^v \mathbf{P}_r^\top \mathbf{C}_r\right)_{i,j} \frac{\mathbf{C}_{i,j}^2 + \widetilde{\mathbf{C}}_{i,j}^2}{\widetilde{\mathbf{C}}_{i,j}}.
\end{aligned} \quad (140)$$

Accordingly, $\mathcal{F}(\mathbf{C}, \widetilde{\mathbf{C}})$ is an auxiliary function, and under the updating rule Eq. (11), the loss is monotonically decreasing. Therefore, we can get the following inequality,

$$
\begin{aligned}
\mathcal{J}\left(\mathbf{E}_r^{(h+1)}, \mathbf{C}^{(h+1)}, \mathbf{C}_r^{(h)}, \mathbf{D}^{(h)}, \mathbf{P}_r^{(h)}, \mathbf{E}^{(h)}, \mathbf{F}_r^{(h)}, a_r^{(h)}, b_r^{(h)}\right) \leq \\
\mathcal{J}\left(\mathbf{E}_r^{(h+1)}, \mathbf{C}^{(h)}, \mathbf{C}_r^{(h)}, \mathbf{D}^{(h)}, \mathbf{P}_r^{(h)}, \mathbf{E}^{(h)}, \mathbf{F}_r^{(h)}, a_r^{(h)}, b_r^{(h)}\right).
\end{aligned}
\tag{141}
$$

▶ During optimizing $\mathbf{C}_r$, after removing irrelevant items, the Lagrange function can be simplified as

$$
\begin{aligned}
\mathcal{L}(\mathbf{C}_r, \Omega_r) = \mathrm{Tr}\, \big( & a_r^2 \mathbf{C}_r \mathbf{D}_\psi \mathbf{E}_r^\top \mathbf{G}_r \mathbf{G}_r^\top \mathbf{E}_r \mathbf{D}_\psi^\top \mathbf{C}_r^\top - 2a_r^2 \mathbf{X}_r \mathbf{G}_r \mathbf{G}_r^\top \mathbf{E}_r \mathbf{D}_\psi^\top \mathbf{C}_r^\top + \\
& 2a_r^2 \mathbf{P}_r \mathbf{C} \mathbf{D}_\gamma \mathbf{E}_r^\top \mathbf{G}_r \mathbf{G}_r^\top \mathbf{E}_r \mathbf{D}_\psi^\top \mathbf{C}_r^\top + \lambda \mathbf{C}^\top \mathbf{P}_r^\top \mathbf{C}_r + \Omega_r \mathbf{C}_r^\top \mathbf{1}_{d_r} \mathbf{1}_{d_r}^\top \mathbf{C}_r - 2\Omega_r \mathbf{C}_r^\top \mathbf{1}_{d_r} \mathbf{1}_k^\top \big).
\end{aligned}
\tag{142}
$$

For the matrix $\mathbf{X}_r \mathbf{G}_r \mathbf{G}_r^\top \mathbf{E}_r \mathbf{D}_\psi^\top \mathbf{C}_r^\top$, in conjunction with the non-negative property of $\mathbf{C}_r$, we have

$$
\mathrm{Tr}\left(\mathbf{X}_r \mathbf{G}_r \mathbf{G}_r^\top \mathbf{E}_r \mathbf{D}_\psi^\top \mathbf{C}_r^\top\right) \geq \sum \left(\mathbf{X}_r \mathbf{G}_r \mathbf{G}_r^\top \mathbf{E}_r \mathbf{D}_\psi^\top\right)_{i,j} (\widetilde{\mathbf{C}}_r)_{i,j} \left(1 + \log \frac{(\mathbf{C}_r)_{i,j}}{(\widetilde{\mathbf{C}}_r)_{i,j}}\right).
\tag{143}
$$

For the term $\mathrm{Tr}\left(\Omega_r \mathbf{C}_r^\top \mathbf{1}_{d_r} \mathbf{1}_k^\top\right)$, according to the cyclic property of trace operation, we have

$$
\mathrm{Tr}\left(\Omega_r \mathbf{C}_r^\top \mathbf{1}_{d_r} \mathbf{1}_k^\top\right) = \mathrm{Tr}\left(\mathbf{1}_{d_r} \mathbf{1}_k^\top \Omega_r \mathbf{C}_r^\top\right) \geq \sum \left(\mathbf{1}_{d_r} \mathbf{1}_k^\top \Omega_r\right)_{i,j} (\widetilde{\mathbf{C}}_r)_{i,j} \left(1 + \log \frac{(\mathbf{C}_r)_{i,j}}{(\widetilde{\mathbf{C}}_r)_{i,j}}\right).
\tag{144}
$$

For the matrix $\mathbf{P}_r \mathbf{C} \mathbf{D}_\gamma \mathbf{E}_r^\top \mathbf{G}_r \mathbf{G}_r^\top \mathbf{E}_r \mathbf{D}_\psi^\top \mathbf{C}_r^\top$, we have

$$
\mathrm{Tr}\left(\mathbf{P}_r \mathbf{C} \mathbf{D}_\gamma \mathbf{E}_r^\top \mathbf{G}_r \mathbf{G}_r^\top \mathbf{E}_r \mathbf{D}_\psi^\top \mathbf{C}_r^\top\right) \leq \sum \left(\mathbf{P}_r \mathbf{C} \mathbf{D}_\gamma \mathbf{E}_r^\top \mathbf{G}_r \mathbf{G}_r^\top \mathbf{E}_r \mathbf{D}_\psi^\top\right)_{i,j} \frac{(\widetilde{\mathbf{C}}_r)_{i,j}^2 + (\mathbf{C}_r)_{i,j}^2}{2(\widetilde{\mathbf{C}}_r)_{i,j}}.
\tag{145}
$$

For $\mathrm{Tr}\left(\mathbf{C}^\top \mathbf{P}_r^\top \mathbf{C}_r\right)$, combined with the transpose property of trace operation, we have

$$
\mathrm{Tr}\left(\mathbf{C}^\top \mathbf{P}_r^\top \mathbf{C}_r\right) = \mathrm{Tr}\left(\mathbf{C}_r^\top \mathbf{P}_r \mathbf{C}\right) \leq \sum (\mathbf{P}_r \mathbf{C})_{i,j} \frac{(\mathbf{C}_r)_{i,j}^2 + (\widetilde{\mathbf{C}}_r)_{i,j}^2}{2(\widetilde{\mathbf{C}}_r)_{i,j}}.
\tag{146}
$$

For the matrix $\mathbf{C}_r \mathbf{D}_\psi \mathbf{E}_r^\top \mathbf{G}_r \mathbf{G}_r^\top \mathbf{E}_r \mathbf{D}_\psi^\top \mathbf{C}_r^\top$, suppose that for any $i$ and $j$, $(\mathbf{C}_r)_{i,j} = w_{i,j}(\widetilde{\mathbf{C}}_r)_{i,j}$ where $w_{i,j}$ is a constant, after element folding, we can get

$$
\begin{aligned}
\mathrm{Tr}\left(\mathbf{C}_r \mathbf{D}_\psi \mathbf{E}_r^\top \mathbf{G}_r \mathbf{G}_r^\top \mathbf{E}_r \mathbf{D}_\psi^\top \mathbf{C}_r^\top\right) &= \sum \sum (\widetilde{\mathbf{C}}_r)_{i,t} (\mathbf{D}_\psi \mathbf{E}_r^\top \mathbf{G}_r \mathbf{G}_r^\top \mathbf{E}_r \mathbf{D}_\psi^\top)_{t,j} (\widetilde{\mathbf{C}}_r)_{i,j} w_{i,t} w_{i,j} \\
&\leq \frac{1}{2} \sum \sum (\widetilde{\mathbf{C}}_r)_{i,t} (\mathbf{D}_\psi \mathbf{E}_r^\top \mathbf{G}_r \mathbf{G}_r^\top \mathbf{E}_r \mathbf{D}_\psi^\top)_{t,j} (\widetilde{\mathbf{C}}_r)_{i,j} (w_{i,t}^2 + w_{i,j}^2).
\end{aligned}
\tag{147}
$$

Combined with the symmetry, we can further have

$$
\begin{aligned}
\sum \sum (\widetilde{\mathbf{C}}_r)_{i,t} (\mathbf{D}_\psi \mathbf{E}_r^\top \mathbf{G}_r \mathbf{G}_r^\top \mathbf{E}_r \mathbf{D}_\psi^\top)_{t,j} (\widetilde{\mathbf{C}}_r)_{i,j} w_{i,t}^2 &= \sum \sum (\widetilde{\mathbf{C}}_r)_{i,j} (\mathbf{D}_\psi \mathbf{E}_r^\top \mathbf{G}_r \mathbf{G}_r^\top \mathbf{E}_r \mathbf{D}_\psi^\top)_{j,t} (\widetilde{\mathbf{C}}_r)_{i,t} w_{i,t}^2 \\
&= \sum (\widetilde{\mathbf{C}}_r \mathbf{D}_\psi \mathbf{E}_r^\top \mathbf{G}_r \mathbf{G}_r^\top \mathbf{E}_r \mathbf{D}_\psi^\top)_{i,t} (\mathbf{C}_r)_{i,t} w_{i,t} \\
&= \sum (\widetilde{\mathbf{C}}_r \mathbf{D}_\psi \mathbf{E}_r^\top \mathbf{G}_r \mathbf{G}_r^\top \mathbf{E}_r \mathbf{D}_\psi^\top)_{i,t} \frac{(\mathbf{C}_r)_{i,t}^2}{(\widetilde{\mathbf{C}}_r)_{i,t}}.
\end{aligned}
\tag{148}
$$

For the second term in Eq. (147), based on the element merging, we can get

$$
\begin{aligned}
\sum \sum (\widetilde{\mathbf{C}}_r)_{i,t} (\mathbf{D}_\psi \mathbf{E}_r^\top \mathbf{G}_r \mathbf{G}_r^\top \mathbf{E}_r \mathbf{D}_\psi^\top)_{t,j} (\widetilde{\mathbf{C}}_r)_{i,j} w_{i,j}^2 &= \sum (\widetilde{\mathbf{C}}_r \mathbf{D}_\psi \mathbf{E}_r^\top \mathbf{G}_r \mathbf{G}_r^\top \mathbf{E}_r \mathbf{D}_\psi^\top)_{i,j} (\widetilde{\mathbf{C}}_r)_{i,j} w_{i,j}^2 \\
&= \sum (\widetilde{\mathbf{C}}_r \mathbf{D}_\psi \mathbf{E}_r^\top \mathbf{G}_r \mathbf{G}_r^\top \mathbf{E}_r \mathbf{D}_\psi^\top)_{i,j} \frac{(\mathbf{C}_r)_{i,j}^2}{(\widetilde{\mathbf{C}}_r)_{i,j}}.
\end{aligned}
\tag{149}
$$

Combining Eqs. (147), (148) and (149) yields

$$\mathrm{Tr}\left(\mathbf{C}_r \mathbf{D}_\psi \mathbf{E}_r^\top \mathbf{G}_r \mathbf{G}_r^\top \mathbf{E}_r \mathbf{D}_\psi^\top \mathbf{C}_r^\top\right) \leq \sum \left(\widetilde{\mathbf{C}}_r \mathbf{D}_\psi \mathbf{E}_r^\top \mathbf{G}_r \mathbf{G}_r^\top \mathbf{E}_r \mathbf{D}_\psi^\top\right)_{i,j} \frac{(\mathbf{C}_r)_{i,j}^2}{(\widetilde{\mathbf{C}}_r)_{i,j}}. \tag{150}$$

For the term $\mathrm{Tr}\left(\Omega_r \mathbf{C}_r^\top \mathbf{1}_{d_r} \mathbf{1}_{d_r}^\top \mathbf{C}_r\right)$, we have

$$\begin{aligned}
\mathrm{Tr}\left(\Omega_r \mathbf{C}_r^\top \mathbf{1}_{d_r} \mathbf{1}_{d_r}^\top \mathbf{C}_r\right) &= \mathrm{Tr}\left(\mathbf{1}_{d_r} \mathbf{1}_{d_r}^\top \mathbf{C}_r \Omega_r \mathbf{C}_r^\top\right) = \sum\sum \left(\mathbf{1}_{d_r} \mathbf{1}_{d_r}^\top\right)_{i,l} (\mathbf{C}_r)_{l,t} (\Omega_r)_{t,j} (\mathbf{C}_r)_{i,j} \\
&= \sum\sum \left(\mathbf{1}_{d_r} \mathbf{1}_{d_r}^\top\right)_{i,l} (\widetilde{\mathbf{C}}_r)_{l,t} (\Omega_r)_{t,j} (\widetilde{\mathbf{C}}_r)_{i,j} w_{l,t} w_{i,j} \\
&\leq \frac{1}{2} \sum\sum \left(\mathbf{1}_{d_r} \mathbf{1}_{d_r}^\top\right)_{i,l} (\widetilde{\mathbf{C}}_r)_{l,t} (\Omega_r)_{t,j} (\widetilde{\mathbf{C}}_r)_{i,j} (w_{l,t}^2 + w_{i,j}^2).
\end{aligned} \tag{151}$$

For the term $\sum\sum \left(\mathbf{1}_{d_r} \mathbf{1}_{d_r}^\top\right)_{i,l} (\widetilde{\mathbf{C}}_r)_{l,t} (\Omega_r)_{t,j} (\widetilde{\mathbf{C}}_r)_{i,j} w_{i,j}^2$, in conjunction with the element merging property, we can obtain

$$\sum\sum \left(\mathbf{1}_{d_r} \mathbf{1}_{d_r}^\top\right)_{i,l} (\widetilde{\mathbf{C}}_r)_{l,t} (\Omega_r)_{t,j} (\widetilde{\mathbf{C}}_r)_{i,j} w_{i,j}^2 = \sum \left(\mathbf{1}_{d_r} \mathbf{1}_{d_r}^\top \widetilde{\mathbf{C}}_r \Omega_r\right)_{i,j} (\widetilde{\mathbf{C}}_r)_{i,j} w_{i,j}^2 = \sum \left(\mathbf{1}_{d_r} \mathbf{1}_{d_r}^\top \widetilde{\mathbf{C}}_r \Omega_r\right)_{i,j} \frac{(\mathbf{C}_r)_{i,j}^2}{(\widetilde{\mathbf{C}}_r)_{i,j}}. \tag{152}$$

According to the facts that $\mathbf{1}_{d_r} \mathbf{1}_{d_r}^\top$ is symmetric and that $\Omega_r$ is diagonal, we can get the following equality,

$$\sum\sum \left(\mathbf{1}_{d_r} \mathbf{1}_{d_r}^\top\right)_{i,l} (\widetilde{\mathbf{C}}_r)_{l,t} (\Omega_r)_{t,j} (\widetilde{\mathbf{C}}_r)_{i,j} w_{l,t}^2 = \sum\sum \left(\mathbf{1}_{d_r} \mathbf{1}_{d_r}^\top\right)_{l,i} (\widetilde{\mathbf{C}}_r)_{l,t} (\Omega_r)_{j,t} (\widetilde{\mathbf{C}}_r)_{i,j} w_{l,t}^2. \tag{153}$$

Further, in conjunction with the element commutative law, we have

$$\sum\sum \left(\mathbf{1}_{d_r} \mathbf{1}_{d_r}^\top\right)_{l,i} (\widetilde{\mathbf{C}}_r)_{l,t} (\Omega_r)_{j,t} (\widetilde{\mathbf{C}}_r)_{i,j} w_{l,t}^2 = \sum\sum \left(\mathbf{1}_{d_r} \mathbf{1}_{d_r}^\top\right)_{l,i} (\widetilde{\mathbf{C}}_r)_{i,j} (\Omega_r)_{j,t} (\widetilde{\mathbf{C}}_r)_{l,t} w_{l,t}^2. \tag{154}$$

Using element combination property yields

$$\begin{aligned}
\sum\sum \left(\mathbf{1}_{d_r} \mathbf{1}_{d_r}^\top\right)_{l,i} (\widetilde{\mathbf{C}}_r)_{i,j} (\Omega_r)_{j,t} (\widetilde{\mathbf{C}}_r)_{l,t} w_{l,t}^2 &= \sum\sum \left(\mathbf{1}_{d_r} \mathbf{1}_{d_r}^\top \widetilde{\mathbf{C}}_r \Omega_r\right)_{l,t} (\widetilde{\mathbf{C}}_r)_{l,t} w_{l,t}^2 \\
&= \sum\sum \left(\mathbf{1}_{d_r} \mathbf{1}_{d_r}^\top \widetilde{\mathbf{C}}_r \Omega_r\right)_{l,t} (\mathbf{C}_r)_{l,t} w_{l,t} \\
&= \sum\sum \left(\mathbf{1}_{d_r} \mathbf{1}_{d_r}^\top \widetilde{\mathbf{C}}_r \Omega_r\right)_{l,t} \frac{(\mathbf{C}_r)_{l,t}^2}{(\widetilde{\mathbf{C}}_r)_{l,t}}.
\end{aligned} \tag{155}$$

Based on Eqs. (151), (152), (153), (154) and (155), therefore, we have

$$\mathrm{Tr}\left(\Omega_r \mathbf{C}_r^\top \mathbf{1}_{d_r} \mathbf{1}_{d_r}^\top \mathbf{C}_r\right) \leq \sum \left(\mathbf{1}_{d_r} \mathbf{1}_{d_r}^\top \widetilde{\mathbf{C}}_r \Omega_r\right)_{i,j} \frac{(\mathbf{C}_r)_{i,j}^2}{(\widetilde{\mathbf{C}}_r)_{i,j}}. \tag{156}$$

Combining Eqs. (143), (144), (145), (146), (150) and (156), we can get

$$\begin{aligned}
\mathcal{F}(\mathbf{C}_r, \widetilde{\mathbf{C}}_r) &= a_r^2 \sum \left(\widetilde{\mathbf{C}}_r \mathbf{D}_\psi \mathbf{E}_r^\top \mathbf{G}_r \mathbf{G}_r^\top \mathbf{E}_r \mathbf{D}_\psi^\top\right)_{i,j} \frac{(\mathbf{C}_r)_{i,j}^2}{(\widetilde{\mathbf{C}}_r)_{i,j}} - 2a_r^2 \sum \left(\mathbf{X}_r \mathbf{G}_r \mathbf{G}_r^\top \mathbf{E}_r \mathbf{D}_\psi^\top\right)_{i,j} (\widetilde{\mathbf{C}}_r)_{i,j} \left(1 + \log \frac{(\mathbf{C}_r)_{i,j}}{(\widetilde{\mathbf{C}}_r)_{i,j}}\right) \\
&\quad + a_r^2 \sum \left(\mathbf{P}_r \mathbf{C} \mathbf{D}_\gamma \mathbf{E}_r^\top \mathbf{G}_r \mathbf{G}_r^\top \mathbf{E}_r \mathbf{D}_\psi^\top\right)_{i,j} \frac{(\widetilde{\mathbf{C}}_r)_{i,j}^2 + (\mathbf{C}_r)_{i,j}^2}{(\widetilde{\mathbf{C}}_r)_{i,j}} + \frac{\lambda}{2} \sum (\mathbf{P}_r \mathbf{C})_{i,j} \frac{(\mathbf{C}_r)_{i,j}^2 + (\widetilde{\mathbf{C}}_r)_{i,j}^2}{(\widetilde{\mathbf{C}}_r)_{i,j}} \\
&\quad + \left(\mathbf{1}_{d_r} \mathbf{1}_{d_r}^\top \widetilde{\mathbf{C}}_r \Omega_r\right)_{i,j} \frac{(\mathbf{C}_r)_{i,j}^2}{(\widetilde{\mathbf{C}}_r)_{i,j}} - 2 \sum \left(\mathbf{1}_{d_r} \mathbf{1}_k^\top \Omega_r\right)_{i,j} (\widetilde{\mathbf{C}}_r)_{i,j} \left(1 + \log \frac{(\mathbf{C}_r)_{i,j}}{(\widetilde{\mathbf{C}}_r)_{i,j}}\right).
\end{aligned} \tag{157}$$

$\mathcal{F}(\mathbf{C}_r, \widetilde{\mathbf{C}}_r)$ is an auxiliary function and with the updating rule Eq. (13), the loss value decreases monotonically. Consequently, we can obtain the following inequality,

$$
\mathcal{J}\left(\mathbf{E}_r^{(h+1)}, \mathbf{C}^{(h+1)}, \mathbf{C}_r^{(h+1)}, \mathbf{D}^{(h)}, \mathbf{P}_r^{(h)}, \mathbf{E}^{(h)}, \mathbf{F}_r^{(h)}, a_r^{(h)}, b_r^{(h)}\right) \le
$$
$$
\mathcal{J}\left(\mathbf{E}_r^{(h+1)}, \mathbf{C}^{(h+1)}, \mathbf{C}_r^{(h)}, \mathbf{D}^{(h)}, \mathbf{P}_r^{(h)}, \mathbf{E}^{(h)}, \mathbf{F}_r^{(h)}, a_r^{(h)}, b_r^{(h)}\right).
$$
(158)

▶ During optimizing $\mathbf{D}$, owing to it being non-negative, we can get

$$
\mathrm{Tr}\left(\sum_{r=1}^{v} a_r^2 [\mathbf{P}_r\mathbf{C}|\mathbf{C}_r]^\top \mathbf{X}_r \mathbf{G}_r \mathbf{G}_r^\top \mathbf{E}_r \mathbf{D}^\top\right) = \sum\left(\sum_{r=1}^{v} a_r^2 [\mathbf{P}_r\mathbf{C}|\mathbf{C}_r]^\top \mathbf{X}_r \mathbf{G}_r \mathbf{G}_r^\top \mathbf{E}_r\right)_{i,j} \mathbf{D}_{i,j}
$$
$$
\ge \sum\left(\sum_{r=1}^{v} a_r^2 [\mathbf{P}_r\mathbf{C}|\mathbf{C}_r]^\top \mathbf{X}_r \mathbf{G}_r \mathbf{G}_r^\top \mathbf{E}_r\right)_{i,j} \widetilde{\mathbf{D}}_{i,j}\left(1 + \log\frac{\mathbf{D}_{i,j}}{\widetilde{\mathbf{D}}_{i,j}}\right).
$$
(159)

Suppose that for any $s$ and $t$, it holds for $\mathbf{D}_{s,t} = d_{s,t}\widetilde{\mathbf{D}}_{s,t}$ under constant $d_{s,t}$. Then, utilizing element-wise expanding, we have

$$
\mathrm{Tr}\left([\mathbf{P}_r\mathbf{C}|\mathbf{C}_r]^\top[\mathbf{P}_r\mathbf{C}|\mathbf{C}_r]\mathbf{D}\mathbf{E}_r^\top \mathbf{G}_r \mathbf{G}_r^\top \mathbf{E}_r \mathbf{D}^\top\right) =
$$
$$
\sum\sum \left([\mathbf{P}_r\mathbf{C}|\mathbf{C}_r]^\top[\mathbf{P}_r\mathbf{C}|\mathbf{C}_r]\right)_{i,s} \mathbf{D}_{s,t}\left(\mathbf{E}_r^\top \mathbf{G}_r \mathbf{G}_r^\top \mathbf{E}_r\right)_{t,j} \mathbf{D}_{i,j} =
$$
$$
\sum\sum \left([\mathbf{P}_r\mathbf{C}|\mathbf{C}_r]^\top[\mathbf{P}_r\mathbf{C}|\mathbf{C}_r]\right)_{i,s} \widetilde{\mathbf{D}}_{s,t}\left(\mathbf{E}_r^\top \mathbf{G}_r \mathbf{G}_r^\top \mathbf{E}_r\right)_{t,j} \widetilde{\mathbf{D}}_{i,j} d_{s,t} d_{i,j} \le
$$
$$
\frac{1}{2}\sum\sum \left([\mathbf{P}_r\mathbf{C}|\mathbf{C}_r]^\top[\mathbf{P}_r\mathbf{C}|\mathbf{C}_r]\right)_{i,s} \widetilde{\mathbf{D}}_{s,t}\left(\mathbf{E}_r^\top \mathbf{G}_r \mathbf{G}_r^\top \mathbf{E}_r\right)_{t,j} \widetilde{\mathbf{D}}_{i,j}(d_{i,j}^2 + d_{s,t}^2).
$$
(160)

For the first term, we have

$$
\sum\sum \left([\mathbf{P}_r\mathbf{C}|\mathbf{C}_r]^\top[\mathbf{P}_r\mathbf{C}|\mathbf{C}_r]\right)_{i,s} \widetilde{\mathbf{D}}_{s,t}\left(\mathbf{E}_r^\top \mathbf{G}_r \mathbf{G}_r^\top \mathbf{E}_r\right)_{t,j} \widetilde{\mathbf{D}}_{i,j} d_{i,j}^2 =
$$
$$
\sum \left([\mathbf{P}_r\mathbf{C}|\mathbf{C}_r]^\top[\mathbf{P}_r\mathbf{C}|\mathbf{C}_r]\widetilde{\mathbf{D}}\mathbf{E}_r^\top \mathbf{G}_r \mathbf{G}_r^\top \mathbf{E}_r\right)_{i,j} \widetilde{\mathbf{D}}_{i,j} d_{i,j}^2 =
$$
$$
\sum \left([\mathbf{P}_r\mathbf{C}|\mathbf{C}_r]^\top[\mathbf{P}_r\mathbf{C}|\mathbf{C}_r]\widetilde{\mathbf{D}}\mathbf{E}_r^\top \mathbf{G}_r \mathbf{G}_r^\top \mathbf{E}_r\right)_{i,j} \frac{\mathbf{D}_{i,j}^2}{\widetilde{\mathbf{D}}_{i,j}}.
$$
(161)

Further, combining the symmetry, we have

$$
\sum\sum \left([\mathbf{P}_r\mathbf{C}|\mathbf{C}_r]^\top[\mathbf{P}_r\mathbf{C}|\mathbf{C}_r]\right)_{i,s} \widetilde{\mathbf{D}}_{s,t}\left(\mathbf{E}_r^\top \mathbf{G}_r \mathbf{G}_r^\top \mathbf{E}_r\right)_{t,j} \widetilde{\mathbf{D}}_{i,j} d_{i,j}^2 =
$$
$$
\sum\sum \left([\mathbf{P}_r\mathbf{C}|\mathbf{C}_r]^\top[\mathbf{P}_r\mathbf{C}|\mathbf{C}_r]\right)_{s,i} \widetilde{\mathbf{D}}_{i,j}\left(\mathbf{E}_r^\top \mathbf{G}_r \mathbf{G}_r^\top \mathbf{E}_r\right)_{j,t} \widetilde{\mathbf{D}}_{s,t} d_{s,t}^2 =
$$
$$
\sum \left([\mathbf{P}_r\mathbf{C}|\mathbf{C}_r]^\top[\mathbf{P}_r\mathbf{C}|\mathbf{C}_r]\widetilde{\mathbf{D}}\mathbf{E}_r^\top \mathbf{G}_r \mathbf{G}_r^\top \mathbf{E}_r\right)_{i,j} \frac{\mathbf{D}_{i,j}^2}{\widetilde{\mathbf{D}}_{i,j}}.
$$
(162)

Combining Eqs. (160), (161) and (162) yields

$$
\mathrm{Tr}\left([\mathbf{P}_r\mathbf{C}|\mathbf{C}_r]^\top[\mathbf{P}_r\mathbf{C}|\mathbf{C}_r]\mathbf{D}\mathbf{E}_r^\top \mathbf{G}_r \mathbf{G}_r^\top \mathbf{E}_r\mathbf{D}^\top\right) \le \sum \left([\mathbf{P}_r\mathbf{C}|\mathbf{C}_r]^\top[\mathbf{P}_r\mathbf{C}|\mathbf{C}_r]\widetilde{\mathbf{D}}\mathbf{E}_r^\top \mathbf{G}_r \mathbf{G}_r^\top \mathbf{E}_r\right)_{i,j} \frac{\mathbf{D}_{i,j}^2}{\widetilde{\mathbf{D}}_{i,j}}.
$$
(163)

Therefore, we have

$$
\mathrm{Tr}\left(\sum_{r=1}^{v} a_r^2 [\mathbf{P}_r\mathbf{C}|\mathbf{C}_r]^\top[\mathbf{P}_r\mathbf{C}|\mathbf{C}_r]\mathbf{D}\mathbf{E}_r^\top \mathbf{G}_r \mathbf{G}_r^\top \mathbf{E}_r\mathbf{D}^\top\right) \le \sum_{r=1}^{v} a_r^2 \sum \left([\mathbf{P}_r\mathbf{C}|\mathbf{C}_r]^\top[\mathbf{P}_r\mathbf{C}|\mathbf{C}_r]\widetilde{\mathbf{D}}\mathbf{E}_r^\top \mathbf{G}_r \mathbf{G}_r^\top \mathbf{E}_r\right)_{i,j} \frac{\mathbf{D}_{i,j}^2}{\widetilde{\mathbf{D}}_{i,j}}.
$$
(164)

Based on Eqs. (159) and (164), we can get

$$
\begin{aligned}
\mathcal{F}(\mathbf{D}, \widetilde{\mathbf{D}}) = \sum_{r=1}^{v} a_r^2 \sum \left( [\mathbf{P}_r \mathbf{C} | \mathbf{C}_r]^\top [\mathbf{P}_r \mathbf{C} | \mathbf{C}_r] \widetilde{\mathbf{D}} \mathbf{E}_r^\top \mathbf{G}_r \mathbf{G}_r^\top \mathbf{E}_r \right)_{i,j} \frac{\mathbf{D}_{i,j}^2}{\widetilde{\mathbf{D}}_{i,j}} - \\
2 \sum_{r=1}^{v} a_r^2 \sum \left( [\mathbf{P}_r \mathbf{C} | \mathbf{C}_r]^\top \mathbf{X}_r \mathbf{G}_r \mathbf{G}_r^\top \mathbf{E}_r \right)_{i,j} \widetilde{\mathbf{D}}_{i,j} \left( 1 + \log \frac{\mathbf{D}_{i,j}}{\widetilde{\mathbf{D}}_{i,j}} \right).
\end{aligned}
\tag{165}
$$

So, $\mathcal{F}(\mathbf{C}, \widetilde{\mathbf{C}})$ is an auxiliary function. Under the updating rule Eq. (15), the loss is monotonically decreasing. As a result, we have the following inequality,

$$
\begin{aligned}
\mathcal{J} \left( \mathbf{E}_r^{(h+1)}, \mathbf{C}^{(h+1)}, \mathbf{C}_r^{(h+1)}, \mathbf{D}^{(h+1)}, \mathbf{P}_r^{(h)}, \mathbf{E}^{(h)}, \mathbf{F}_r^{(h)}, a_r^{(h)}, b_r^{(h)} \right) \leq \\
\mathcal{J} \left( \mathbf{E}_r^{(h+1)}, \mathbf{C}^{(h+1)}, \mathbf{C}_r^{(h+1)}, \mathbf{D}^{(h)}, \mathbf{P}_r^{(h)}, \mathbf{E}^{(h)}, \mathbf{F}_r^{(h)}, a_r^{(h)}, b_r^{(h)} \right).
\end{aligned}
\tag{166}
$$

▶ During optimizing $\mathbf{P}_r$, the Lagrange function can be simplified as

$$
\begin{aligned}
\mathcal{L}(\mathbf{P}_r, \Gamma_r) = \mathrm{Tr} \big( a_r^2 \mathbf{P}_r \mathbf{C} \mathbf{D}_\gamma \mathbf{E}_r^\top \mathbf{G}_r \mathbf{G}_r^\top \mathbf{E}_r \mathbf{D}_\gamma^\top \mathbf{C}^\top \mathbf{P}_r^\top + 2 a_r^2 \mathbf{C}_r \mathbf{D}_\varphi \mathbf{E}_r^\top \mathbf{G}_r \mathbf{G}_r^\top \mathbf{E}_r \mathbf{D}_\gamma^\top \mathbf{C}^\top \mathbf{P}_r^\top + \lambda \mathbf{P}_r^\top \mathbf{C}_r \mathbf{C}^\top \\
- 2 a_r^2 \mathbf{X}_r \mathbf{G}_r \mathbf{G}_r^\top \mathbf{E}_r \mathbf{D}_\gamma^\top \mathbf{C}^\top \mathbf{P}_r^\top + \Gamma_r \mathbf{C}^\top \mathbf{P}_r^\top \mathbf{1}_{d_r} \mathbf{1}_{d_r}^\top \mathbf{P}_r \mathbf{C} - 2 \Gamma_r \mathbf{C}^\top \mathbf{P}_r^\top \mathbf{1}_{d_r} \mathbf{1}_k^\top \big).
\end{aligned}
\tag{167}
$$

Then, about the matrix $\mathbf{X}_r \mathbf{G}_r \mathbf{G}_r^\top \mathbf{E}_r \mathbf{D}_\gamma^\top \mathbf{C}^\top \mathbf{P}_r^\top$, in conjunction with the non-negativity of elements, we can have the following inequality,

$$
\mathrm{Tr} \left( \mathbf{X}_r \mathbf{G}_r \mathbf{G}_r^\top \mathbf{E}_r \mathbf{D}_\gamma^\top \mathbf{C}^\top \mathbf{P}_r^\top \right) \geq \sum \left( \mathbf{X}_r \mathbf{G}_r \mathbf{G}_r^\top \mathbf{E}_r \mathbf{D}_\gamma^\top \mathbf{C}^\top \right)_{i,j} \left( \widetilde{\mathbf{P}}_r \right)_{i,j} \left( 1 + \log \frac{(\mathbf{P}_r)_{i,j}}{(\widetilde{\mathbf{P}}_r)_{i,j}} \right).
\tag{168}
$$

For the term $\mathrm{Tr} \left( \mathbf{P}_r \mathbf{C} \mathbf{D}_\gamma \mathbf{E}_r^\top \mathbf{G}_r \mathbf{G}_r^\top \mathbf{E}_r \mathbf{D}_\gamma^\top \mathbf{C}^\top \mathbf{P}_r^\top \right)$, after unfolding element by element, we have

$$
\begin{aligned}
\mathrm{Tr} \left( \mathbf{P}_r \mathbf{C} \mathbf{D}_\gamma \mathbf{E}_r^\top \mathbf{G}_r \mathbf{G}_r^\top \mathbf{E}_r \mathbf{D}_\gamma^\top \mathbf{C}^\top \mathbf{P}_r^\top \right) = \sum \sum (\widetilde{\mathbf{P}}_r)_{t,l} (\mathbf{C} \mathbf{D}_\gamma \mathbf{E}_r^\top \mathbf{G}_r \mathbf{G}_r^\top \mathbf{E}_r \mathbf{D}_\gamma^\top \mathbf{C}^\top)_{l,s} (\widetilde{\mathbf{P}}_r)_{t,s} \, q_{t,l} q_{t,s} \\
\leq \sum \sum (\widetilde{\mathbf{P}}_r \mathbf{C} \mathbf{D}_\gamma \mathbf{E}_r^\top \mathbf{G}_r \mathbf{G}_r^\top \mathbf{E}_r \mathbf{D}_\gamma^\top \mathbf{C}^\top)_{t,s} \frac{(\mathbf{P}_r)_{t,s}^2}{2(\widetilde{\mathbf{P}}_r)_{t,s}} + \\
\sum \sum (\widetilde{\mathbf{P}}_r)_{t,l} (\mathbf{C} \mathbf{D}_\gamma \mathbf{E}_r^\top \mathbf{G}_r \mathbf{G}_r^\top \mathbf{E}_r \mathbf{D}_\gamma^\top \mathbf{C}^\top)_{l,s} (\widetilde{\mathbf{P}}_r)_{t,s} \frac{q_{t,l}^2}{2},
\end{aligned}
\tag{169}
$$

where $\widetilde{\mathbf{P}}_r$ is acquired based on the assumption that $(\mathbf{P}_r)_{t,l}$ is $q_{t,l}$ times of $(\widetilde{\mathbf{P}}_r)_{t,l}$ for any $t$ and $l$. $q_{t,l}$ is a constant.

Further, for $(\widetilde{\mathbf{P}}_r)_{t,l} (\mathbf{C} \mathbf{D}_\gamma \mathbf{E}_r^\top \mathbf{G}_r \mathbf{G}_r^\top \mathbf{E}_r \mathbf{D}_\gamma^\top \mathbf{C}^\top)_{l,s} (\widetilde{\mathbf{P}}_r)_{t,s} \frac{q_{t,l}^2}{2}$, by element exchange and combining the symmetry, we can get

$$
\begin{aligned}
\sum \sum (\widetilde{\mathbf{P}}_r)_{t,l} (\mathbf{C} \mathbf{D}_\gamma \mathbf{E}_r^\top \mathbf{G}_r \mathbf{G}_r^\top \mathbf{E}_r \mathbf{D}_\gamma^\top \mathbf{C}^\top)_{l,s} (\widetilde{\mathbf{P}}_r)_{t,s} \frac{q_{t,l}^2}{2} = \sum (\widetilde{\mathbf{P}}_r \mathbf{C} \mathbf{D}_\gamma \mathbf{E}_r^\top \mathbf{G}_r \mathbf{G}_r^\top \mathbf{E}_r \mathbf{D}_\gamma^\top \mathbf{C}^\top)_{t,l} (\widetilde{\mathbf{P}}_r)_{t,l} \frac{q_{t,l}^2}{2} \\
= \sum (\widetilde{\mathbf{P}}_r \mathbf{C} \mathbf{D}_\gamma \mathbf{E}_r^\top \mathbf{G}_r \mathbf{G}_r^\top \mathbf{E}_r \mathbf{D}_\gamma^\top \mathbf{C}^\top)_{t,l} \frac{(\mathbf{P}_r)_{t,l}^2}{2(\widetilde{\mathbf{P}}_r)_{t,l}}.
\end{aligned}
\tag{170}
$$

Therefore, for $\mathbf{P}_r \mathbf{C} \mathbf{D}_\gamma \mathbf{E}_r^\top \mathbf{G}_r \mathbf{G}_r^\top \mathbf{E}_r \mathbf{D}_\gamma^\top \mathbf{C}^\top \mathbf{P}_r^\top$, we can obtain the following inequality,

$$
\mathrm{Tr} \left( \mathbf{P}_r \mathbf{C} \mathbf{D}_\gamma \mathbf{E}_r^\top \mathbf{G}_r \mathbf{G}_r^\top \mathbf{E}_r \mathbf{D}_\gamma^\top \mathbf{C}^\top \mathbf{P}_r^\top \right) \leq \sum \left( \widetilde{\mathbf{P}}_r \mathbf{C} \mathbf{D}_\gamma \mathbf{E}_r^\top \mathbf{G}_r \mathbf{G}_r^\top \mathbf{E}_r \mathbf{D}_\gamma^\top \mathbf{C}^\top \right)_{i,j} \frac{(\mathbf{P}_r)_{i,j}^2}{(\widetilde{\mathbf{P}}_r)_{i,j}}.
\tag{171}
$$

For the term $\mathbf{C}_r \mathbf{D}_\varphi \mathbf{E}_r^\top \mathbf{G}_r \mathbf{G}_r^\top \mathbf{E}_r \mathbf{D}_\gamma^\top \mathbf{C}^\top \mathbf{P}_r^\top$, combining Cauchy inequality, we can get

$$
\mathrm{Tr} \left( \mathbf{C}_r \mathbf{D}_\varphi \mathbf{E}_r^\top \mathbf{G}_r \mathbf{G}_r^\top \mathbf{E}_r \mathbf{D}_\gamma^\top \mathbf{C}^\top \mathbf{P}_r^\top \right) \leq \frac{1}{2} \sum \left( \mathbf{C}_r \mathbf{D}_\varphi \mathbf{E}_r^\top \mathbf{G}_r \mathbf{G}_r^\top \mathbf{E}_r \mathbf{D}_\gamma^\top \mathbf{C}^\top \right)_{i,j} \frac{(\mathbf{P}_r)_{i,j}^2 + (\widetilde{\mathbf{P}}_r)_{i,j}^2}{(\widetilde{\mathbf{P}}_r)_{i,j}}.
\tag{172}
$$

For the term $\mathbf{P}_r^\top \mathbf{C}_r \mathbf{C}^\top$, likewise, we have

$$\mathrm{Tr}\left(\mathbf{P}_r^\top \mathbf{C}_r \mathbf{C}^\top\right) \le \frac{1}{2}\sum \left(\mathbf{C}_r \mathbf{C}^\top\right)_{i,j} \frac{(\mathbf{P}_r)_{i,j}^2 + (\widetilde{\mathbf{P}}_r)_{i,j}^2}{(\widetilde{\mathbf{P}}_r)_{i,j}}. \tag{173}$$

Subsequently, for the term $\Gamma_r \mathbf{C}^\top \mathbf{P}_r^\top \mathbf{1}_{d_r} \mathbf{1}_{d_r}^\top \mathbf{P}_r \mathbf{C}$, utilizing cyclic property and element-to-element expanding, we can obtain

$$
\begin{aligned}
\mathrm{Tr}\left(\Gamma_r \mathbf{C}^\top \mathbf{P}_r^\top \mathbf{1}_{d_r} \mathbf{1}_{d_r}^\top \mathbf{P}_r \mathbf{C}\right) &= \mathrm{Tr}\left(\mathbf{1}_{d_r} \mathbf{1}_{d_r}^\top \mathbf{P}_r \mathbf{C}\Gamma_r \mathbf{C}^\top \mathbf{P}_r^\top\right) \\
&\le \sum \left(\mathbf{1}_{d_r} \mathbf{1}_{d_r}^\top \widetilde{\mathbf{P}}_r \mathbf{C}\Gamma_r \mathbf{C}^\top\right)_{t,j} \frac{(\mathbf{P}_r)_{t,j}^2}{2(\widetilde{\mathbf{P}}_r)_{t,j}} + \\
&\qquad \sum\sum \left(\mathbf{1}_{d_r} \mathbf{1}_{d_r}^\top\right)_{t,l} (\widetilde{\mathbf{P}}_r)_{l,s} \left(\mathbf{C}\Gamma_r \mathbf{C}^\top\right)_{s,j} (\widetilde{\mathbf{P}}_r)_{t,j} \frac{q_{l,s}^2}{2}.
\end{aligned} \tag{174}
$$

For the second item, combined the symmetry, we can derive the following equality,

$$
\begin{aligned}
\sum\sum \left(\mathbf{1}_{d_r} \mathbf{1}_{d_r}^\top\right)_{t,l} (\widetilde{\mathbf{P}}_r)_{l,s} \left(\mathbf{C}\Gamma_r \mathbf{C}^\top\right)_{s,j} (\widetilde{\mathbf{P}}_r)_{t,j}\, q_{l,s}^2 &= \sum\sum \left(\mathbf{1}_{d_r} \mathbf{1}_{d_r}^\top\right)_{l,t} (\widetilde{\mathbf{P}}_r)_{t,j} \left(\mathbf{C}\Gamma_r \mathbf{C}^\top\right)_{j,s} (\widetilde{\mathbf{P}}_r)_{l,s}\, q_{l,s}^2 \\
&= \sum \left(\mathbf{1}_{d_r} \mathbf{1}_{d_r}^\top \widetilde{\mathbf{P}}_r \mathbf{C}\Gamma_r \mathbf{C}^\top\right)_{l,s} \frac{(\mathbf{P}_r)_{l,s}^2}{(\widetilde{\mathbf{P}}_r)_{l,s}}.
\end{aligned} \tag{175}
$$

Therefore, we have

$$\mathrm{Tr}\left(\Gamma_r \mathbf{C}^\top \mathbf{P}_r^\top \mathbf{1}_{d_r} \mathbf{1}_{d_r}^\top \mathbf{P}_r \mathbf{C}\right) \le \sum \left(\mathbf{1}_{d_r} \mathbf{1}_{d_r}^\top \widetilde{\mathbf{P}}_r \mathbf{C}\Gamma_r \mathbf{C}^\top\right)_{i,j} \frac{(\mathbf{P}_r)_{i,j}^2}{(\widetilde{\mathbf{P}}_r)_{i,j}}. \tag{176}$$

Likewise, for $\Gamma_r \mathbf{C}^\top \mathbf{P}_r^\top \mathbf{1}_{d_r} \mathbf{1}_k^\top$, we have

$$\mathrm{Tr}\left(\Gamma_r \mathbf{C}^\top \mathbf{P}_r^\top \mathbf{1}_{d_r} \mathbf{1}_k^\top\right) = \mathrm{Tr}\left(\mathbf{1}_{d_r} \mathbf{1}_k^\top \Gamma_r \mathbf{C}^\top \mathbf{P}_r^\top\right) \ge \sum \left(\mathbf{1}_{d_r} \mathbf{1}_k^\top \Gamma_r \mathbf{C}^\top\right)_{i,j} \left(\widetilde{\mathbf{P}}_r\right)_{i,j} \left(1 + \log \frac{(\mathbf{P}_r)_{i,j}}{(\widetilde{\mathbf{P}}_r)_{i,j}}\right). \tag{177}$$

Hence, combining Eqs. (168), (171), (172), (173), (176) and (177), we can have

$$
\begin{aligned}
\mathcal{F}(\mathbf{P}_r, \widetilde{\mathbf{P}}_r) =\, & a_r^2 \sum \left(\widetilde{\mathbf{P}}_r \mathbf{C}\mathbf{D}_\gamma \mathbf{E}_r^\top \mathbf{G}_r \mathbf{G}_r^\top \mathbf{E}_r \mathbf{D}_\gamma^\top \mathbf{C}^\top\right)_{i,j} \frac{(\mathbf{P}_r)_{i,j}^2}{(\widetilde{\mathbf{P}}_r)_{i,j}} + \frac{\lambda}{2}\sum \left(\mathbf{C}_r \mathbf{C}^\top\right)_{i,j} \frac{(\mathbf{P}_r)_{i,j}^2 + (\widetilde{\mathbf{P}}_r)_{i,j}^2}{(\widetilde{\mathbf{P}}_r)_{i,j}} + \\
& a_r^2 \sum \left(\mathbf{C}\mathbf{D}_\varphi \mathbf{E}_r^\top \mathbf{G}_r \mathbf{G}_r^\top \mathbf{E}_r \mathbf{D}_\gamma^\top \mathbf{C}^\top\right)_{i,j} \frac{(\mathbf{P}_r)_{i,j}^2 + (\widetilde{\mathbf{P}}_r)_{i,j}^2}{(\widetilde{\mathbf{P}}_r)_{i,j}} + \sum \left(\mathbf{1}_{d_r} \mathbf{1}_{d_r}^\top \widetilde{\mathbf{P}}_r \mathbf{C}\Gamma_r \mathbf{C}^\top\right)_{i,j} \frac{(\mathbf{P}_r)_{i,j}^2}{(\widetilde{\mathbf{P}}_r)_{i,j}} \\
& - 2a_r^2 \sum \left(\mathbf{X}_r \mathbf{G}_r \mathbf{G}_r^\top \mathbf{E}_r \mathbf{D}_\gamma^\top \mathbf{C}^\top\right)_{i,j} \left(\widetilde{\mathbf{P}}_r\right)_{i,j} \left(1 + \log \frac{(\mathbf{P}_r)_{i,j}}{(\widetilde{\mathbf{P}}_r)_{i,j}}\right) \\
& - 2 \sum \left(\mathbf{1}_{d_r} \mathbf{1}_k^\top \Gamma_r \mathbf{C}^\top\right)_{i,j} \left(\widetilde{\mathbf{P}}_r\right)_{i,j} \left(1 + \log \frac{(\mathbf{P}_r)_{i,j}}{(\widetilde{\mathbf{P}}_r)_{i,j}}\right).
\end{aligned} \tag{178}
$$

Consequently, $\mathcal{F}(\mathbf{P}_r, \widetilde{\mathbf{P}}_r)$ is an auxiliary function, and under the updating rule Eq. (17), the loss function is monotonically decreasing. Accordingly, we have the following inequality,

$$
\begin{aligned}
\mathcal{J}\left(\mathbf{E}_r^{(h+1)}, \mathbf{C}^{(h+1)}, \mathbf{C}_r^{(h+1)}, \mathbf{D}^{(h+1)}, \mathbf{P}_r^{(h+1)}, \mathbf{E}^{(h)}, \mathbf{F}_r^{(h)}, a_r^{(h)}, b_r^{(h)}\right) &\le \\
\mathcal{J}\left(\mathbf{E}_r^{(h+1)}, \mathbf{C}^{(h+1)}, \mathbf{C}_r^{(h+1)}, \mathbf{D}^{(h+1)}, \mathbf{P}_r^{(h)}, \mathbf{E}^{(h)}, \mathbf{F}_r^{(h)}, a_r^{(h)}, b_r^{(h)}\right)&.
\end{aligned} \tag{179}
$$

▶ During optimizing $\mathbf{E}$, since its optimal solution can be acquired via singular value decomposition, we can get that after each iteration, there always has

$$
\begin{aligned}
\mathcal{J}\left(\mathbf{E}_r^{(h+1)}, \mathbf{C}^{(h+1)}, \mathbf{C}_r^{(h+1)}, \mathbf{D}^{(h+1)}, \mathbf{P}_r^{(h+1)}, \mathbf{E}^{(h+1)}, \mathbf{F}_r^{(h)}, a_r^{(h)}, b_r^{(h)}\right) \leq \\
\mathcal{J}\left(\mathbf{E}_r^{(h+1)}, \mathbf{C}^{(h+1)}, \mathbf{C}_r^{(h+1)}, \mathbf{D}^{(h+1)}, \mathbf{P}_r^{(h+1)}, \mathbf{E}^{(h)}, \mathbf{F}_r^{(h)}, a_r^{(h)}, b_r^{(h)}\right).
\end{aligned}
\tag{180}
$$

▶ During optimizing $\mathbf{F}_r$, its optimal solution is obtained by Eq. (76). Therefore, the objective value is decreasing when optimizing $\mathbf{F}_r$. Accordingly, we have the following inequality,

$$
\begin{aligned}
\mathcal{J}\left(\mathbf{E}_r^{(h+1)}, \mathbf{C}^{(h+1)}, \mathbf{C}_r^{(h+1)}, \mathbf{D}^{(h+1)}, \mathbf{P}_r^{(h+1)}, \mathbf{E}^{(h+1)}, \mathbf{F}_r^{(h+1)}, a_r^{(h)}, b_r^{(h)}\right) \leq \\
\mathcal{J}\left(\mathbf{E}_r^{(h+1)}, \mathbf{C}^{(h+1)}, \mathbf{C}_r^{(h+1)}, \mathbf{D}^{(h+1)}, \mathbf{P}_r^{(h+1)}, \mathbf{E}^{(h+1)}, \mathbf{F}_r^{(h)}, a_r^{(h)}, b_r^{(h)}\right).
\end{aligned}
\tag{181}
$$

▶ During optimizing $a_r$, according to Eq. (82), its solution can be directly obtained. Therefore, we have

$$
\begin{aligned}
\mathcal{J}\left(\mathbf{E}_r^{(h+1)}, \mathbf{C}^{(h+1)}, \mathbf{C}_r^{(h+1)}, \mathbf{D}^{(h+1)}, \mathbf{P}_r^{(h+1)}, \mathbf{E}^{(h+1)}, \mathbf{F}_r^{(h+1)}, a_r^{(h+1)}, b_r^{(h)}\right) \leq \\
\mathcal{J}\left(\mathbf{E}_r^{(h+1)}, \mathbf{C}^{(h+1)}, \mathbf{C}_r^{(h+1)}, \mathbf{D}^{(h+1)}, \mathbf{P}_r^{(h+1)}, \mathbf{E}^{(h+1)}, \mathbf{F}_r^{(h+1)}, a_r^{(h)}, b_r^{(h)}\right).
\end{aligned}
\tag{182}
$$

▶ During optimizing $b_r$, its optimal solution is acquired by Eq. (86). Consequently, we have

$$
\begin{aligned}
\mathcal{J}\left(\mathbf{E}_r^{(h+1)}, \mathbf{C}^{(h+1)}, \mathbf{C}_r^{(h+1)}, \mathbf{D}^{(h+1)}, \mathbf{P}_r^{(h+1)}, \mathbf{E}^{(h+1)}, \mathbf{F}_r^{(h+1)}, a_r^{(h+1)}, b_r^{(h+1)}\right) \leq \\
\mathcal{J}\left(\mathbf{E}_r^{(h+1)}, \mathbf{C}^{(h+1)}, \mathbf{C}_r^{(h+1)}, \mathbf{D}^{(h+1)}, \mathbf{P}_r^{(h+1)}, \mathbf{E}^{(h+1)}, \mathbf{F}_r^{(h+1)}, a_r^{(h+1)}, b_r^{(h)}\right).
\end{aligned}
\tag{183}
$$

Combining Eqs. (109), (141), (158), (166), (179), (180), (181), (182) and (183), we can obtain

$$
\begin{aligned}
\mathcal{J}\left(\mathbf{E}_r^{(h+1)}, \mathbf{C}^{(h+1)}, \mathbf{C}_r^{(h+1)}, \mathbf{D}^{(h+1)}, \mathbf{P}_r^{(h+1)}, \mathbf{E}^{(h+1)}, \mathbf{F}_r^{(h+1)}, a_r^{(h+1)}, b_r^{(h+1)}\right) \leq \\
\mathcal{J}\left(\mathbf{E}_r^{(h)}, \mathbf{C}^{(h)}, \mathbf{C}_r^{(h)}, \mathbf{D}^{(h)}, \mathbf{P}_r^{(h)}, \mathbf{E}^{(h)}, \mathbf{F}_r^{(h)}, a_r^{(h)}, b_r^{(h)}\right),
\end{aligned}
\tag{184}
$$

which indicates that the objective value is monotonically decreasing with each iteration of the algorithm.

Combining the monotonic descent characteristic (i.e., Eq. (184) ) and the lower bound characteristic (i.e., Eq. (126)), we can conclude that the proposed method is convergent. $\qquad\square$

## G. Proof of Theorem 6

### G.1. Computational Complexity

*Proof.* The computing cost of presented algorithm primarily comes from solving $\mathbf{E}_r$, $\mathbf{C}$, $\mathbf{C}_r$, $\mathbf{D}$, $\mathbf{P}_r$, $\mathbf{E}$, $\mathbf{F}_r$, $a_r$ and $b_r$.

During optimizing $\mathbf{E}_r$, note that $\mathbf{G}_r\mathbf{G}_r^\top$ is in $\mathbb{R}^{n\times n}$, directly computing the term $\mathbf{G}_r\mathbf{G}_r^\top\mathbf{X}_r^\top[\mathbf{P}_r\mathbf{C}|\mathbf{C}_r]\mathbf{D}$ will require at least $\mathcal{O}(n^2)$ cost. To decrease the computational cost, we observe that $\mathbf{G}_r\mathbf{G}_r^\top$ is a diagonal matrix. Specially, its diagonal elements consist of $\sum_{j=1}^{n_v}(\mathbf{G}_r)_{1,j}, \sum_{j=1}^{n_v}(\mathbf{G}_r)_{2,j}, \cdots, \sum_{j=1}^{n_v}(\mathbf{G}_r)_{n,j}$. We can equivalently transform the term $\mathbf{G}_r\mathbf{G}_r^\top\mathbf{X}_r^\top[\mathbf{P}_r\mathbf{C}|\mathbf{C}_r]\mathbf{D}$ as $(\mathbf{X}_r \odot \mathbf{O}_r)^\top[\mathbf{P}_r\mathbf{C}|\mathbf{C}_r]\mathbf{D}$ where $\mathbf{O}_r \in \mathbb{R}^{d_r\times n}$ is equal to $\mathbf{1}_{d_r}\left(\sum_{j=1}^{n_v}(\mathbf{G}_r)_{1,j}, \sum_{j=1}^{n_v}(\mathbf{G}_r)_{2,j}, \cdots, \sum_{j=1}^{n_v}(\mathbf{G}_r)_{n,j}\right)$ and $\odot$ denotes the element-wise multiplication operation. Based on this, computing $\mathbf{G}_r\mathbf{G}_r^\top\mathbf{X}_r^\top[\mathbf{P}_r\mathbf{C}|\mathbf{C}_r]\mathbf{D}$ will take $\mathcal{O}(d_rn + 2nd_rk + 2nk^2)$ cost. In the similar way, we can obtain that computing $\mathbf{G}_r\mathbf{G}_r^\top\mathbf{E}_r\mathbf{D}^\top[\mathbf{P}_r\mathbf{C}|\mathbf{C}_r]^\top[\mathbf{P}_r\mathbf{C}|\mathbf{C}_r]\mathbf{D}$ will need $\mathcal{O}(nk + 2nk^2 + 2nkd_r)$ cost. In addition, computing $\mathbf{E}\mathbf{F}_r^\top$ will need $\mathcal{O}(nk^2)$. Therefore, optimizing each $\mathbf{E}_r$ totally requires $\mathcal{O}\left(nd_rk + nk^2\right)$ cost.

During optimizing $\mathbf{C}$, adopting the element-wise multiplication technique employed on $\mathbf{E}_r$, we can obtain that computing $\mathbf{X}_r\mathbf{G}_r\mathbf{G}_r^\top\mathbf{E}_r\mathbf{D}_\gamma^\top$ and $[\mathbf{P}_r\mathbf{C}|\mathbf{C}_r]\mathbf{D}\mathbf{E}_r^\top\mathbf{G}_r\mathbf{G}_r^\top\mathbf{E}_r\mathbf{D}_\gamma^\top$ will require $\mathcal{O}(d_rn + d_rnk + d_rk^2)$ and $\mathcal{O}(2d_rk^2 + d_rkn + d_rk^2)$

respectively. Accordingly, computing $\sum_{r=1}^{v} a_r^2 \mathbf{X}_r \mathbf{G}_r \mathbf{G}_r^\top \mathbf{E}_r \mathbf{D}_\gamma^\top$ and $\sum_{r=1}^{v} a_r^2 [\mathbf{P}_r \mathbf{C} | \mathbf{C}_r] \mathbf{D} \mathbf{E}_r^\top \mathbf{G}_r \mathbf{G}_r^\top \mathbf{E}_r \mathbf{D}_\gamma^\top$ will require $\mathcal{O}(dkn + dk^2)$ and $\mathcal{O}(dk^2 + dkn)$ respectively where $d$ is the dimension sum of all view data. Therefore, optimizing the matrix $\mathbf{C}$ totally requires $\mathcal{O}\left(dk^2 + dkn\right)$ computing cost.

During optimizing $\mathbf{C}_r$, computing the term $\mathbf{X}_r \mathbf{G}_r \mathbf{G}_r^\top \mathbf{E}_r \mathbf{D}_\psi^\top$ requires $\mathcal{O}(d_r n + d_r nk + d_r k^2)$ computational cost. Computing $[\mathbf{P}_r \mathbf{C} | \mathbf{C}_r] \mathbf{D} \mathbf{E}_r^\top \mathbf{G}_r \mathbf{G}_r^\top \mathbf{E}_r \mathbf{D}_\psi^\top$ requires $\mathcal{O}(nk + 2d_r k^2 + d_r nk + d_r k^2)$. Therefore, optimizing each $\mathbf{C}_r$ totally requires $\mathcal{O}\left(d_r nk + d_r k^2\right)$ cost.

During optimizing $\mathbf{D}$, computing the terms $[\mathbf{P}_r \mathbf{C} | \mathbf{C}_r]^\top \mathbf{X}_r \mathbf{G}_r \mathbf{G}_r^\top \mathbf{E}_r$ and $[\mathbf{P}_r \mathbf{C} | \mathbf{C}_r]^\top [\mathbf{P}_r \mathbf{C} | \mathbf{C}_r] \mathbf{D} \mathbf{E}_r^\top \mathbf{G}_r \mathbf{G}_r^\top \mathbf{E}_r$ take $\mathcal{O}(d_r n + 2kd_r n + 2k^2 n)$ and $\mathcal{O}(nk + 4k^2 d_r + 4k^3 + 2nk^2)$ respectively. Consequently, computing $\sum_{r=1}^{v} a_r^2 [\mathbf{P}_r \mathbf{C} | \mathbf{C}_r]^\top \mathbf{X}_r \mathbf{G}_r \mathbf{G}_r^\top \mathbf{E}_r$ and $\sum_{r=1}^{v} a_r^2 [\mathbf{P}_r \mathbf{C} | \mathbf{C}_r]^\top [\mathbf{P}_r \mathbf{C} | \mathbf{C}_r] \mathbf{D} \mathbf{E}_r^\top \mathbf{G}_r \mathbf{G}_r^\top \mathbf{E}_r$ takes $\mathcal{O}(kdn + vk^2 n)$ and $\mathcal{O}(k^2 d + vk^3 + vnk^2)$ respectively. Therefore, optimizing $\mathbf{D}$ totally requires $\mathcal{O}\left(kdn + vk^2 n + k^2 d + vk^3 + vnk^2\right)$.

During optimizing $\mathbf{P}_r$, constructing $\mathbf{X}_r \mathbf{G}_r \mathbf{G}_r^\top \mathbf{E}_r \mathbf{D}_\gamma^\top \mathbf{C}^\top$ and $\mathbf{C}_r \mathbf{C}^\top$ require $\mathcal{O}(d_r n + d_r nk + d_r k^2)$ and $\mathcal{O}(d_r k^2)$ computing costs respectively. Constructing $[\mathbf{P}_r \mathbf{C} | \mathbf{C}_r] \mathbf{D} \mathbf{E}_r^\top \mathbf{G}_r \mathbf{G}_r^\top \mathbf{E}_r \mathbf{D}_\gamma^\top \mathbf{C}^\top$ requires $\mathcal{O}(nk + 2d_r k^2 + d_r nk)$. Therefore, optimizing each $\mathbf{P}_r$ requires $\mathcal{O}(d_r nk + d_r k^2)$ cost.

During optimizing $\mathbf{E}$, it involves conducting singular value decomposition on $\sum_{r=1}^{v} b_r \mathbf{E}_r \mathbf{F}_r$. Constructing the term $\sum_{r=1}^{v} b_r \mathbf{E}_r \mathbf{F}_r$ and performing singular value decomposition on it require $\mathcal{O}(vnk^2)$ and $\mathcal{O}(nk^2)$ respectively. Performing $\mathbf{U}$ multiplying $\mathbf{V}^\top$ to generate the optimal solution takes $\mathcal{O}(nk^2)$ Therefore, optimizing $\mathbf{E}$ totally requires $\mathcal{O}\left(vnk^2\right)$ cost.

During optimizing $\mathbf{F}_r$, it involves conducting singular value decomposition on $\mathbf{E}^\top \mathbf{E}_r$, which requires $\mathcal{O}(k^3)$ computing cost. Additionally, constructing the term $\mathbf{E}^\top \mathbf{E}_r$ requires $\mathcal{O}(nk^2)$ cost. Therefore, optimizing each $\mathbf{F}_r$ totally requires $\mathcal{O}\left(nk^2 + k^3\right)$ cost.

During optimizing $a_r$, it involves calculating the term $\left\| \mathbf{X}_r \mathbf{G}_r - [\mathbf{P}_r \mathbf{C} | \mathbf{C}_r] \mathbf{D} \mathbf{E}_r^\top \mathbf{G}_r \right\|_F^2$. Note that $\mathbf{X}_r$ and $\mathbf{G}_r$ are in $\mathbb{R}^{d_v \times n}$ and $\mathbb{R}^{n \times n_v}$ respectively. Directly computing $\left\| \mathbf{X}_r \mathbf{G}_r - [\mathbf{P}_r \mathbf{C} | \mathbf{C}_r] \mathbf{D} \mathbf{E}_r^\top \mathbf{G}_r \right\|_F^2$ will require at least $\mathcal{O}(nn_v)$ computational cost. Especially, when the incomplete percentage is relatively small, i.e., $n_v$ is larger, the cost is almost close to $\mathcal{O}(n^2)$. To reduce the cost, we observe that the value of $\left\| \mathbf{X}_r \mathbf{G}_r - [\mathbf{P}_r \mathbf{C} | \mathbf{C}_r] \mathbf{D} \mathbf{E}_r^\top \mathbf{G}_r \right\|_F^2$ is the same as that of $\left\| \mathbf{X}_r \mathbf{G}_r \mathbf{G}_r^\top - [\mathbf{P}_r \mathbf{C} | \mathbf{C}_r] \mathbf{D} \mathbf{E}_r^\top \mathbf{G}_r \mathbf{G}_r^\top \right\|_F^2$. In virtue of the element-wise multiplication technique adopted in analyzing $\mathbf{E}_r$, we can obtain that computing the term $\left\| \mathbf{X}_r \mathbf{G}_r \mathbf{G}_r^\top - [\mathbf{P}_r \mathbf{C} | \mathbf{C}_r] \mathbf{D} \mathbf{E}_r^\top \mathbf{G}_r \mathbf{G}_r^\top \right\|_F^2$ requires $\mathcal{O}(d_r n + kn + 2d_r k^2 + d_r kn)$ cost. Accordingly, computing the term $\sum_{r=1}^{v} \frac{1}{\| \mathbf{X}_r \mathbf{G}_r - [\mathbf{P}_r \mathbf{C} | \mathbf{C}_r] \mathbf{D} \mathbf{E}_r^\top \mathbf{G}_r \|_F^2}$ requires $\mathcal{O}(dn + vkn + 2dk^2 + dkn)$ cost. Therefore, optimizing all $a_r$ totally requires $\mathcal{O}(vkn + dk^2 + dkn)$ computing cost.

During optimizing $b_r$, computing the term $\text{Tr}\left(\mathbf{E}^\top \mathbf{E}_r \mathbf{F}_r\right)$ will require $\mathcal{O}(nk^2 + k^3)$ cost. Accordingly, computing $\sum_{r=1}^{v} \text{Tr}\left(\mathbf{E}^\top \mathbf{E}_r \mathbf{F}_r\right)$ will need $\mathcal{O}(vnk^2 + vk^3)$. Therefore, optimizing all $b_r$ totally requires $\mathcal{O}\left(vnk^2 + vk^3\right)$ computational cost.

Based on the above analysis, we can obtain that optimizing all $\mathbf{E}_r$, $\mathbf{C}$, $\mathbf{C}_r$, $\mathbf{D}$, $\mathbf{P}_r$, $\mathbf{E}$, $\mathbf{F}_r$, $a_r$ and $b_r$ will require $\mathcal{O}\left(ndk + vnk^2\right)$, $\mathcal{O}\left(dk^2 + dkn\right)$, $\mathcal{O}\left(dnk + dk^2\right)$, $\mathcal{O}\left(kdn + vk^2 n + k^2 d + vk^3 + vnk^2\right)$, $\mathcal{O}(dnk + dk^2)$, $\mathcal{O}\left(vnk^2\right)$, $\mathcal{O}\left(vnk^2 + vk^3\right)$, $\mathcal{O}(vkn + dk^2 + dkn)$ and $\mathcal{O}\left(vnk^2 + vk^3\right)$ respectively. Therefore, the total computing cost is $\mathcal{O}(ndk + vnk^2 + dk^2 + vk^3)$. Considering that the view number $v$ and the cluster number $k$ are generally greatly smaller than the sample number $n$ and that the dimension sum $d$ is independent of $n$, we can obtain that the total computational complexity is $\mathcal{O}(n)$. That is, the computing cost is linear with respect to the sample size $n$. $\qquad \square$

## G.2. Space complexity

*Proof.* During optimizing $\mathbf{E}_r$, generating the term $\mathbf{G}_r \mathbf{G}_r^\top \mathbf{X}_r^\top [\mathbf{P}_r \mathbf{C} | \mathbf{C}_r] \mathbf{D}$ needs $\mathcal{O}(d_r n + 2d_r k + 2k^2 + 2nk)$ memory cost. Similarly, generating $\mathbf{G}_r \mathbf{G}_r^\top \mathbf{E}_r \mathbf{D}^\top [\mathbf{P}_r \mathbf{C} | \mathbf{C}_r]^\top [\mathbf{P}_r \mathbf{C} | \mathbf{C}_r] \mathbf{D}$ needs $\mathcal{O}(nk + 2k^2 + 2kd_r + 2k^2 d_r + nkd_r)$ cost. For the term $\mathbf{E} \mathbf{F}_r^\top$, it needs $\mathcal{O}(nk + k^2 + nk^2)$. Therefore, during optimizing the variable $\mathbf{E}_r$, it requires $\mathcal{O}(k^2 d_r + nkd_r + nk^2)$ memory cost.

During optimizing $\mathbf{C}$, generating the term $\mathbf{X}_r \mathbf{G}_r \mathbf{G}_r^\top \mathbf{E}_r \mathbf{D}_\gamma^\top$ needs $\mathcal{O}(d_r n + nk + k^2 + d_r k)$ memory cost. Generating the term $[\mathbf{P}_r \mathbf{C} | \mathbf{C}_r] \mathbf{D} \mathbf{E}_r^\top \mathbf{G}_r \mathbf{G}_r^\top \mathbf{E}_r \mathbf{D}_\gamma^\top$ needs $\mathcal{O}(nk + 2d_r k + 2k^2 + d_r n)$ cost. Consequently, generating $\sum_{r=1}^{v} a_r^2 \mathbf{X}_r \mathbf{G}_r \mathbf{G}_r^\top \mathbf{E}_r \mathbf{D}_\gamma^\top$ and $\sum_{r=1}^{v} a_r^2 [\mathbf{P}_r \mathbf{C} | \mathbf{C}_r] \mathbf{D} \mathbf{E}_r^\top \mathbf{G}_r \mathbf{G}_r^\top \mathbf{E}_r \mathbf{D}_\gamma^\top$ will need $\mathcal{O}(dn + vnk + vk^2 + dk)$ and $\mathcal{O}(vnk + 2dk + 2vk^2 + dn)$ respectively. Therefore, during optimizing $\mathbf{C}$, it requires $\mathcal{O}(dn + vnk + vk^2 + dk)$ memory cost.

During optimizing $\mathbf{C}_r$, generating the terms $\mathbf{X}_r\mathbf{G}_r\mathbf{G}_r^\top\mathbf{E}_r\mathbf{D}_\psi^\top$ and $[\mathbf{P}_r\mathbf{C}|\mathbf{C}_r]\mathbf{D}\mathbf{E}_r^\top\mathbf{G}_r\mathbf{G}_r^\top\mathbf{E}_r\mathbf{D}_\psi^\top$ needs $\mathcal{O}(d_r n + nk + k^2 + d_r k)$ and $\mathcal{O}(2d_r k + 2k^2 + nk + d_r n)$ memory costs respectively. Therefore, during optimizing $\mathbf{C}_r$, it requires $\mathcal{O}(d_r n + nk + k^2 + d_r k)$ memory cost.

During optimizing $\mathbf{D}$, generating the term $[\mathbf{P}_r\mathbf{C}|\mathbf{C}_r]^\top\mathbf{X}_r\mathbf{G}_r\mathbf{G}_r^\top\mathbf{E}_r$ needs $\mathcal{O}(d_r n + 2d_r k + 2nk + 2k^2)$ memory cost. Generating $[\mathbf{P}_r\mathbf{C}|\mathbf{C}_r]^\top[\mathbf{P}_r\mathbf{C}|\mathbf{C}_r]\mathbf{D}\mathbf{E}_r^\top\mathbf{G}_r\mathbf{G}_r^\top\mathbf{E}_r$ needs $\mathcal{O}(2d_r k + 4k^2 + 2nk)$. Accordingly, generating $\sum_{r=1}^v[\mathbf{P}_r\mathbf{C}|\mathbf{C}_r]^\top\mathbf{X}_r\mathbf{G}_r\mathbf{G}_r^\top\mathbf{E}_r$ and $\sum_{r=1}^v[\mathbf{P}_r\mathbf{C}|\mathbf{C}_r]^\top[\mathbf{P}_r\mathbf{C}|\mathbf{C}_r]\mathbf{D}\mathbf{E}_r^\top\mathbf{G}_r\mathbf{G}_r^\top\mathbf{E}_r$ will need $\mathcal{O}(dn + 2dk + 2vnk + 2vk^2)$ and $\mathcal{O}(2dk + 4k^2v + 2vnk)$ respectively. Therefore, during optimizing $\mathbf{D}$, it requires $\mathcal{O}(dn + dk + vnk + vk^2)$ memory cost.

During optimizing $\mathbf{P}_r$, generating the terms $\mathbf{X}_r\mathbf{G}_r\mathbf{G}_r^\top\mathbf{E}_r\mathbf{D}_\gamma^\top\mathbf{C}^\top$ and $\mathbf{C}_r\mathbf{C}^\top$ needs $\mathcal{O}(d_r n + nk + k^2 + d_r k)$ and $\mathcal{O}(d_r k + k^2)$ memory costs respectively. Generating the term $[\mathbf{P}_r\mathbf{C}|\mathbf{C}_r]\mathbf{D}\mathbf{E}_r^\top\mathbf{G}_r\mathbf{G}_r^\top\mathbf{E}_r\mathbf{D}_\gamma^\top\mathbf{C}^\top$ needs $\mathcal{O}(d_r k + 2k^2 + nk + d_r n)$. Therefore, during optimizing $\mathbf{P}_r$, it requires $\mathcal{O}(d_r n + nk + k^2 + d_r k)$ memory cost.

During optimizing $\mathbf{E}$, generating $\mathbf{E}_r\mathbf{F}_r$ needs $\mathcal{O}(nk + k^2)$ memory cost. Accordingly, constructing $\sum_{r=1}^v\mathbf{E}_r\mathbf{F}_r$ will need $\mathcal{O}(vnk + vk^2)$. Singular value decomposition and singular matrix multiplication will need $\mathcal{O}(nk + k^2)$. Therefore, during optimizing $\mathbf{E}$, it requires $\mathcal{O}(vnk + vk^2)$ memory cost.

During optimizing $\mathbf{F}_r$, generating $\mathbf{E}^\top\mathbf{E}_r$ needs $\mathcal{O}(nk + k^2)$ cost. Performing singular value decomposition needs $\mathcal{O}(k^2)$. Therefore, during optimizing $\mathbf{F}_r$, it requires $\mathcal{O}(nk + k^2)$ memory cost.

During optimizing $a_r$, instead of calculating the term $\left\|\mathbf{X}_r\mathbf{G}_r - [\mathbf{P}_r\mathbf{C}|\mathbf{C}_r]\mathbf{D}\mathbf{E}_r^\top\mathbf{G}_r\right\|_F^2$, we equivalently calculate the term $\left\|\mathbf{X}_r\mathbf{G}_r\mathbf{G}_r^\top - [\mathbf{P}_r\mathbf{C}|\mathbf{C}_r]\mathbf{D}\mathbf{E}_r^\top\mathbf{G}_r\mathbf{G}_r^\top\right\|_F^2$, which requires $\mathcal{O}(d_r n + kn + 2d_r k)$ memory cost. Accordingly, generating $\sum_{r=1}^v\left\|\mathbf{X}_r\mathbf{G}_r\mathbf{G}_r^\top - [\mathbf{P}_r\mathbf{C}|\mathbf{C}_r]\mathbf{D}\mathbf{E}_r^\top\mathbf{G}_r\mathbf{G}_r^\top\right\|_F^2$ requires $\mathcal{O}(dn + vkn + 2dk)$ cost. Therefore, optimizing all $a_r$ requires $\mathcal{O}(dn + vkn + dk)$ memory cost.

During optimizing $b_r$, constructing the term $\mathbf{E}^\top\mathbf{E}_r\mathbf{F}_r$ needs $\mathcal{O}(nk + k^2)$ memory cost. Accordingly, constructing $\sum_{r=1}^v\left(\text{Tr}\left(\mathbf{E}^\top\mathbf{E}_r\mathbf{F}_r\right)\right)^2$ will need $\mathcal{O}(vnk + vk^2)$. Therefore, optimizing all $b_r$ requires $\mathcal{O}(vnk + vk^2)$ memory cost.

Consequently, we have that during optimizing all $\mathbf{E}_r$, $\mathbf{C}$, $\mathbf{C}_r$, $\mathbf{D}$, $\mathbf{P}_r$, $\mathbf{E}$, $\mathbf{F}_r$, $a_r$ and $b_r$, it will need $\mathcal{O}(k^2 d + nkd + vnk^2)$, $\mathcal{O}(dn + vnk + vk^2 + dk)$, $\mathcal{O}(dn + vnk + vk^2 + dk)$, $\mathcal{O}(dn + dk + vnk + vk^2)$, $\mathcal{O}(dn + vnk + vk^2 + dk)$, $\mathcal{O}(vnk + vk^2)$, $\mathcal{O}(vnk + vk^2)$, $\mathcal{O}(dn + vkn + dk)$ and $\mathcal{O}(vnk + vk^2)$ memory costs respectively. Accordingly, the total memory cost is $\mathcal{O}(k^2 d + nkd + vnk^2)$. Owing to the data dimension sum $d$ being independent of the sample size $n$ and the view number $v$ and the cluster number being far less than $n$, we have that the total space complexity is $\mathcal{O}(n)$. That is, our memory cost is linear with respect to the sample size $n$. $\quad\square$

# H. Experimental Background

## H.1. Data Preprocessing

In experiments, each (observed) view data is normalized to a probability form. Specially, we first map the data to have a mean of 0 and a variance of 1 so as to prevent certain dimensions from encountering excessively large scales, and then employ the softmax function to transform negative elements into positive ones. Subsequently, we conduct normalization operation and transform each view data into a probability distribution. Afterwards, all comparison algorithms are evaluated on these preprocessed datasets.

## H.2. Parameter Setting

Owing to the elements in $\mathbf{X}_r$, $\mathbf{G}_r$, $\mathbf{P}_r$, $\mathbf{C}$, $\mathbf{C}_r$, $\mathbf{D}$, $\mathbf{E}_r$, $\mathbf{a}$ and $\mathbf{b}$ being between 0 and 1, we can obtain that the terms $a_r^2\left\|\mathbf{X}_r\mathbf{G}_r - [\mathbf{P}_r\mathbf{C}|\mathbf{C}_r]\mathbf{D}\mathbf{E}_r^\top\mathbf{G}_r\right\|_F^2$ and $\langle\mathbf{P}_r\mathbf{C},\mathbf{C}_r\rangle$ ( i.e., $\text{Tr}(\mathbf{C}^\top\mathbf{P}_r^\top\mathbf{C}_r)$ ) are less than 1. Additionally, combined with the orthogonality of $\mathbf{E}$ and $\mathbf{E}_r$, we have that the value of $\text{Tr}\left(\mathbf{E}^\top\sum_{r=1}^v b_r\mathbf{E}_r\mathbf{F}_r\right)$ is generally much larger than that of $a_r^2\left\|\mathbf{X}_r\mathbf{G}_r - [\mathbf{P}_r\mathbf{C}|\mathbf{C}_r]\mathbf{D}\mathbf{E}_r^\top\mathbf{G}_r\right\|_F^2$ and that of $\text{Tr}\left(\mathbf{C}^\top\mathbf{P}_r^\top\mathbf{C}_r\right)$. Therefore, we fine-tune $\lambda$ in a wide range while fine-tuning $\beta$ in a relatively narrow range. Further, considering that $\text{Tr}\left(\mathbf{E}^\top\sum_{r=1}^v b_r\mathbf{E}_r\mathbf{F}_r\right)$ is upper-bounded by $\sqrt{vnk}$ and that the sample size $n$ is largely greater than the view number $v$ and the cluster number $k$, we search $\lambda$ and $\beta$ in $[10^{-2}, 10^{-1}, 10^0, 10^1, 10^2]$ and $[10^{-2}, 10^{-1}, 10^0, 10^1, 10^2]\cdot\frac{1}{\sqrt{vnk}}$ respectively.

### H.3. Implementation Details

Since $\mathbf{P}_r\mathbf{C}$ denotes the feature cluster, we normalize it column by column. During optimizing $\mathbf{P}_r$ and $\mathbf{C}$, we do column normalization on them respectively. Similarly, $\mathbf{C}_r$ denotes the feature cluster, and we also normalize it column by column. Additionally, $\mathbf{E}_r$ denotes the sample cluster, and thus we do row normalization on it. $\mathbf{D}$ denotes the association between feature cluster and sample cluster, and we do distribution normalization on it.

To evaluate the clustering outcomes from various perspectives, we adopt three commonly-utilized metrics, purity(PUR), accuracy(ACC), F-score(FSC).

## I. Symbol Description

Table 7 summarizes the symbols used in this manuscript and corresponding meaning as well as the size.

*Table 7.* Symbol Description

| Notation | Description | Size |
|---|---|---|
| $\mathbf{X}_r$ | Data matrix on $r$-th view | $d_r \times n$ |
| $\mathbf{P}_r$ | Guidance matrix on $r$-th view | $d_r \times k$ |
| $\mathbf{C}$ | Perspective-shared matrix | $k \times k$ |
| $\mathbf{C}_r$ | Perspective-specific matrix | $d_r \times k$ |
| $\mathbf{D}$ | Association matrix between feature clusters and sample clusters | $2k \times k$ |
| $\mathbf{E}_r$ | Feature clusters on $r$-th view | $n \times k$ |
| $\mathbf{G}_r$ | Index matrix on $r$-th view | $n \times n_r$ |
| $\mathbf{E}$ | Common feature clusters | $n \times k$ |
| $\mathbf{F}_r$ | Space rotation matrix on $r$-th view | $k \times k$ |
| $\mathbf{a}$ | View weight vector | $v \times 1$ |
| $\mathbf{b}$ | Sample cluster weight vector | $v \times 1$ |
| $\mathbf{D}_\gamma$ | $\mathbf{D}_{1:k,:}$ | $k \times k$ |
| $\mathbf{D}_\varphi$ | $\mathbf{D}_{k+1:2k,:}$ | $k \times k$ |
| $\Phi_r$ | Lagrange multiplier matrix | $n_r \times n_r$ |
| $\Psi_r$ | Lagrange multiplier matrix | $k \times k$ |
| $\Omega_r$ | Lagrange multiplier matrix | $k \times k$ |
| $\Gamma_r$ | Lagrange multiplier matrix | $k \times k$ |
| $\mathbf{H}_r$ | Temporary matrix | $d_r \times n_r$ |
| $\mathbf{J}_r$ | Temporary matrix | $k \times k$ |
| $\mathbf{L}$ | Temporary matrix | $k \times k$ |
| $\mathbf{M}$ | Temporary matrix | $k \times k$ |
| $\widehat{\mathbf{A}}_r$ | Temporary matrix | $d_r \times k$ |
| $\widehat{\mathbf{B}}_r$ | Temporary matrix | $n \times n$ |
| $\widehat{\mathbf{C}}_r$ | Temporary matrix | $n \times k$ |
| $\widehat{\mathbf{K}}_r$ | Temporary matrix | $n \times k$ |
| $\widehat{\mathbf{L}}_r$ | Temporary matrix | $d_r \times 2k$ |
| $d_r$ | Data dimension on $r$-th view | - |
| $d$ | Data dimension sum | - |
| $v$ | The number of views | - |
| $n$ | The number of samples | - |
| $k$ | The number of clusters | - |
| $n_r$ | The number of samples observed on view $r$ | - |
| $[\cdot|\cdot]$ | Matrix concatenation | - |
| $(\cdot)_{i,j}$ | Element in the $i$-th row and $j$-th column of matrix. | - |

## J. Hyper-parameter Sensitivity

In the paper, our model involves two hyper-parameters, $\lambda$ and $\beta$. To investigate the model sensitivity about hyper-parameters, we plot the performance changes with respect to different hyper-parameter values, as suggested in Fig. 3 where for the sake of simplicity, we in figure omit the coefficient $1/\sqrt{nvk}$. One can observe that under given $\beta$, the performance is relatively stable with respect to $\lambda$ while under given $\lambda$, the performance fluctuates within a tolerable range. Therefore, we can state that the proposed model to a certain extent is parameter-robust.

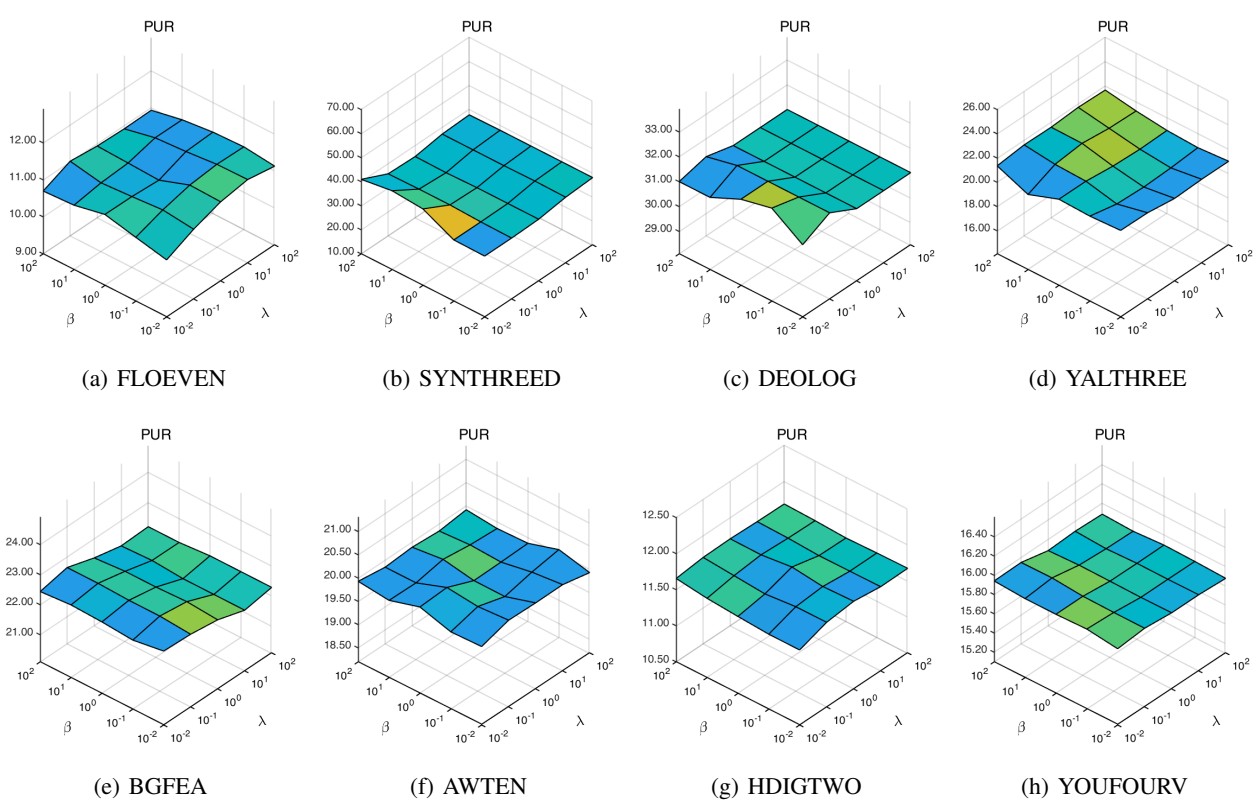

*Figure 3.* Performance Changes under Different Hyper-parameter Values

## K. Experimental Convergence

We have proven the convergence theoretically. Here, to confirm the convergence, we draw the objective value changes, as presented in Fig. 4. One can observe that along with iterations, our loss value is gradually decreasing, which demonstrates that our objective function is experimentally convergent.

## L. More Ablations

### L.1. The Ablation of Common Sample Clusters

In the manuscript, we utilize a common sample cluster matrix to formulate full spectral embedding. This strategy not only facilitates view information communication during the procedure of learning but avoids the fusion between view-specific sample clusters. To validate the effectiveness of this common sample cluster strategy, we organize relevant ablation. The comparison results are reported in Table 8 where OCSC and WCSC represent the clustering results without/with common sample clusters respectively. It can be observed that WCSC outperforms OCSC, illustrating that our common sample cluster strategy is functional.

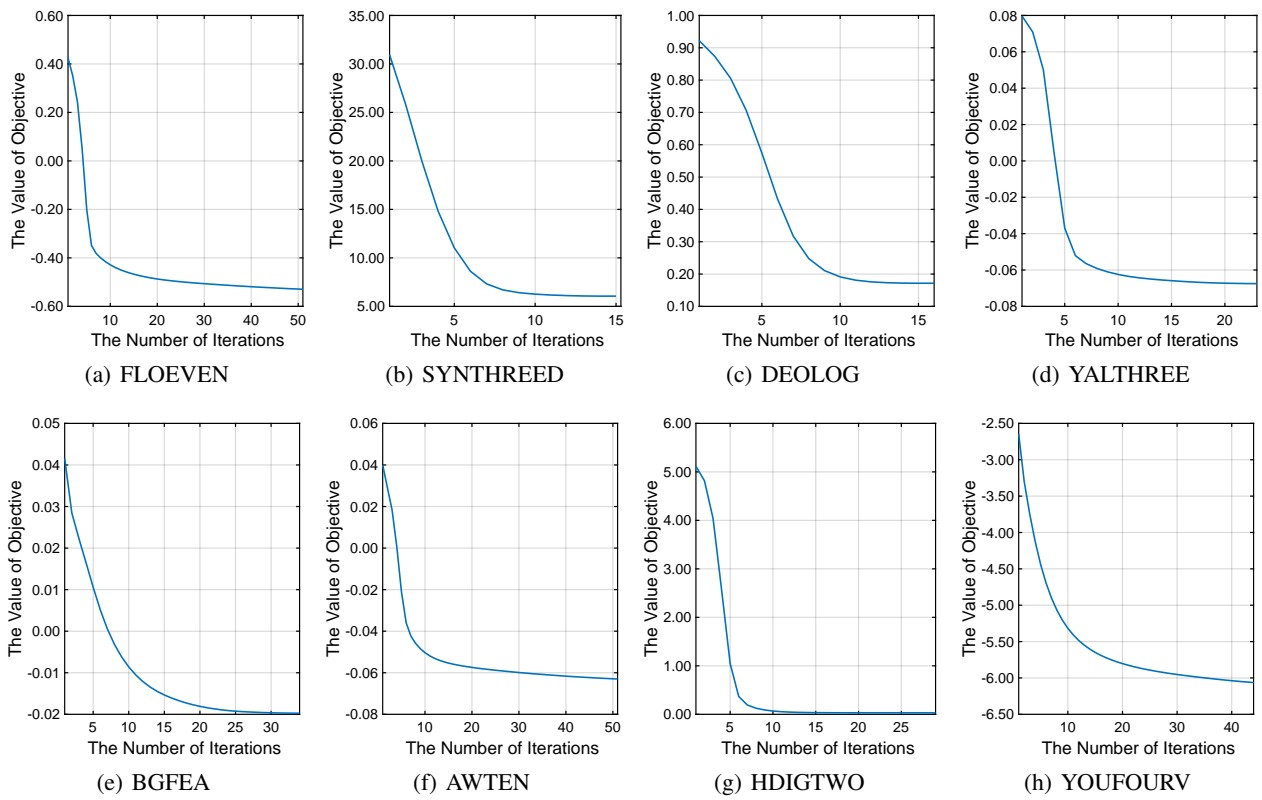

*Figure 4.* Loss Changes along with Iterations

*Table 8.* Ablation for Common Sample Clusters

| DATASET | | FLOEVEN | | | | | | | | | SYNTHREED | | | | | | | | |
|---|---|---|---|---|---|---|---|---|---|---|---|---|---|---|---|---|---|---|
| A/B | | 0.2 | | | 0.5 | | | 0.8 | | | 0.2 | | | 0.5 | | | 0.8 | | |
| | PUR | ACC | FSC | PUR | ACC | FSC | PUR | ACC | FSC | PUR | ACC | FSC | PUR | ACC | FSC | PUR | ACC | FSC |
| OCSC | 10.23 | 10.14 | 4.93 | 10.33 | 9.57 | 5.01 | 10.14 | 9.52 | 4.38 | 40.05 | 38.97 | 33.98 | 40.73 | 38.56 | 33.69 | 36.73 | 36.40 | 30.79 |
| WCSC | **11.17** | **10.80** | **5.89** | **11.25** | **10.53** | **5.88** | **11.54** | **10.74** | **5.83** | **42.50** | **42.12** | **35.21** | **42.83** | **42.83** | **35.24** | **41.04** | **41.04** | **34.59** |
| | | DEOLOG | | | | | | | | | YALTHREE | | | | | | | | |
| OCSC | 29.06 | 20.82 | 16.62 | 30.29 | 21.87 | 17.79 | 28.75 | 22.85 | 18.51 | 22.19 | 20.98 | 6.75 | 19.31 | 19.42 | 5.59 | 18.59 | 18.97 | 5.19 |
| WCSC | **31.84** | **22.32** | **18.45** | **31.84** | **23.74** | **18.41** | **31.28** | **24.25** | **19.56** | **24.94** | **23.55** | **7.89** | **21.73** | **20.61** | **6.30** | **21.55** | **20.70** | **6.70** |
| | | BGFEA | | | | | | | | | AWTEN | | | | | | | | |
| OCSC | 18.76 | 19.24 | 19.32 | 20.44 | 21.88 | 19.70 | 20.45 | 20.21 | 19.46 | 18.54 | 11.21 | 9.91 | 18.53 | 11.02 | 9.60 | 18.53 | 10.10 | 9.33 |
| WCSC | **22.08** | **22.08** | **20.08** | **22.64** | **22.40** | **20.16** | **22.48** | **22.28** | **20.20** | **20.74** | **12.24** | **11.01** | **20.17** | **12.01** | **10.97** | **20.18** | **12.06** | **10.94** |
| | | HDIGTWO | | | | | | | | | YOUFOURV | | | | | | | | |
| OCSC | 9.57 | 9.65 | 9.79 | 8.79 | 10.73 | 9.01 | 10.45 | 9.97 | 9.36 | 13.53 | 8.39 | 8.86 | 14.26 | 9.36 | 8.61 | 13.22 | 9.12 | 8.41 |
| WCSC | **11.73** | **11.51** | **10.03** | **11.75** | **11.60** | **10.11** | **11.79** | **11.70** | **10.22** | **15.99** | **10.91** | **10.48** | **15.95** | **10.85** | **10.47** | **15.95** | **10.80** | **10.47** |

## L.2. The Ablation of Space Rotation

We adopt a space rotation strategy to reorder sample clusters on each view so as to decrease the risk of misregistration. To validate its effectiveness, we conduct ablation experiments, and the results are summarized in Table 9. OSR and WSR represent the clustering results without/with our space rotation strategy respectively. As seen, after rotating, the clustering performance is improved. This gives evidence that our rotation strategy is beneficial for performance enhancement.

*Table 9.* Ablation for Space Rotation

| DATASET | FLOEVEN | | | | | | | | | SYNTHREED | | | | | | | | |
|---|---|---|---|---|---|---|---|---|---|---|---|---|---|---|---|---|---|---|
| A/B | 0.2 | | | 0.5 | | | 0.8 | | | 0.2 | | | 0.5 | | | 0.8 | | |
| | PUR | ACC | FSC | PUR | ACC | FSC | PUR | ACC | FSC | PUR | ACC | FSC | PUR | ACC | FSC | PUR | ACC | FSC |
| OSR | 10.57 | 10.13 | 5.18 | 10.35 | 10.13 | 5.13 | 9.98 | 9.39 | 5.07 | 38.17 | 38.17 | 33.68 | 37.67 | 36.33 | 33.55 | 38.50 | 37.00 | 33.65 |
| WSR | **11.17** | **10.80** | **5.89** | **11.25** | **10.53** | **5.88** | **11.54** | **10.74** | **5.83** | **42.50** | **42.12** | **35.21** | **42.83** | **42.83** | **35.24** | **41.04** | **41.04** | **34.59** |
| | DEOLOG | | | | | | | | | YALTHREE | | | | | | | | |
| OSR | 30.62 | 21.51 | 17.96 | 30.34 | 21.89 | 18.16 | 30.23 | 22.01 | 18.07 | 22.26 | 21.05 | 5.79 | 20.49 | 18.97 | 5.59 | 20.72 | 19.51 | 6.39 |
| WSR | **31.84** | **22.32** | **18.45** | **31.84** | **23.74** | **18.41** | **31.28** | **24.25** | **19.56** | **24.94** | **23.55** | **7.89** | **21.73** | **20.61** | **6.30** | **21.55** | **20.70** | **6.70** |
| | BGFEA | | | | | | | | | AWTEN | | | | | | | | |
| OSR | 21.46 | 21.26 | 19.68 | 21.74 | 21.66 | 19.88 | 21.70 | 21.42 | 19.80 | 19.05 | 11.09 | 10.02 | 19.05 | 11.03 | 9.97 | 19.05 | 11.19 | 10.16 |
| WSR | **22.08** | **22.08** | **20.08** | **22.64** | **22.40** | **20.16** | **22.48** | **22.28** | **20.20** | **20.74** | **12.24** | **11.01** | **20.17** | **12.01** | **10.97** | **20.18** | **12.06** | **10.94** |
| | HDIGTWO | | | | | | | | | YOUFOURV | | | | | | | | |
| OSR | 10.27 | 10.19 | 9.22 | 10.34 | 10.17 | 9.42 | 10.60 | 10.27 | 9.42 | 14.27 | 9.01 | 8.75 | 14.21 | 9.27 | 9.27 | 14.25 | 9.24 | 9.27 |
| WSR | **11.73** | **11.51** | **10.03** | **11.75** | **11.60** | **10.11** | **11.79** | **11.70** | **10.22** | **15.99** | **10.91** | **10.48** | **15.95** | **10.85** | **10.47** | **15.95** | **10.80** | **10.47** |

## L.3. The Ablation of View Weight

Orthogonal to previous techniques regarding views equally, in this manuscript we associate a variable for each view to automatically balance the view contributions. To verify its effectiveness, we ablate the view weighting. The comparison results are summarized in Table 10 where OVW and WVW represent the results without/with view weight respectively. Apparently, WVW does better than OVW, illustrating that the view weighting is effective.

*Table 10.* Ablation for View Weight

| DATASET | FLOEVEN | | | | | | | | | SYNTHREED | | | | | | | | |
|---|---|---|---|---|---|---|---|---|---|---|---|---|---|---|---|---|---|---|
| A/B | 0.2 | | | 0.5 | | | 0.8 | | | 0.2 | | | 0.5 | | | 0.8 | | |
| | PUR | ACC | FSC | PUR | ACC | FSC | PUR | ACC | FSC | PUR | ACC | FSC | PUR | ACC | FSC | PUR | ACC | FSC |
| OVW | 10.06 | 9.61 | 4.78 | 9.92 | 9.41 | 4.69 | 10.43 | 9.44 | 4.75 | 40.95 | 40.95 | 33.82 | 41.52 | 41.14 | 33.89 | 39.21 | 39.97 | 33.19 |
| WVW | **11.17** | **10.80** | **5.89** | **11.25** | **10.53** | **5.88** | **11.54** | **10.74** | **5.83** | **42.50** | **42.12** | **35.21** | **42.83** | **42.83** | **35.24** | **41.04** | **41.04** | **34.59** |
| | DEOLOG | | | | | | | | | YALTHREE | | | | | | | | |
| OVW | 30.12 | 20.84 | 17.32 | 30.39 | 21.74 | 17.62 | **31.81** | 22.99 | 17.92 | 22.06 | 20.61 | **7.92** | 19.89 | 19.13 | 5.25 | 19.67 | 18.74 | 6.00 |
| WVW | **31.84** | **22.32** | **18.45** | **31.84** | **23.74** | **18.41** | 31.28 | **24.25** | **19.56** | **24.94** | **23.55** | 7.89 | **21.73** | **20.61** | **6.30** | **21.55** | **20.70** | **6.70** |
| | BGFEA | | | | | | | | | AWTEN | | | | | | | | |
| OVW | 20.34 | 20.34 | 18.25 | 20.27 | 20.54 | 18.37 | 20.58 | 20.34 | 18.36 | 19.29 | 11.24 | 10.09 | 19.28 | 10.85 | 10.04 | 19.23 | 11.05 | 10.04 |
| WVW | **22.08** | **22.08** | **20.08** | **22.64** | **22.40** | **20.16** | **22.48** | **22.28** | **20.20** | **20.74** | **12.24** | **11.01** | **20.17** | **12.01** | **10.97** | **20.18** | **12.06** | **10.94** |
| | HDIGTWO | | | | | | | | | YOUFOURV | | | | | | | | |
| OVW | 10.78 | 10.54 | 9.13 | 10.86 | 10.77 | 9.22 | 10.88 | 10.78 | 9.24 | 13.65 | 9.56 | 9.16 | 14.17 | 9.50 | 9.14 | 14.03 | 9.45 | 9.15 |
| WVW | **11.73** | **11.51** | **10.03** | **11.75** | **11.60** | **10.11** | **11.79** | **11.70** | **10.22** | **15.99** | **10.91** | **10.48** | **15.95** | **10.85** | **10.47** | **15.95** | **10.80** | **10.47** |

## L.4. The Ablation of Sample Cluster Weight

We also associate weights for the sample clusters on all views to automatically balance them. The ablation results are presented in Table 11 where OSCW and WSCW represent the clustering results without/with the sample cluster weight respectively. As seen, the sample cluster weights indeed facilitate the clustering performance improvement.

# M. Comparison Results under Other Missing Ratios

In order to further exhibit the strengths, we conduct experiments under more missing ratios. Specially, we organize the comparison under missing ratio being 0.3, 0.4, 0.6 and 0.7 respectively. Table 12, 13, 14 and 15 summarize relevant clustering results. According to these tables, one can observe that under these missing ratios, our proposed model still can provide competitive clustering results.

*Table 11.* Ablation for Sample Cluster Weight

| DATASET | FLOEVEN | | | | | | | | | SYNTHREED | | | | | | | | |
|---|---|---|---|---|---|---|---|---|---|---|---|---|---|---|---|---|---|---|
| A/B | 0.2 | | | 0.5 | | | 0.8 | | | 0.2 | | | 0.5 | | | 0.8 | | |
| | PUR | ACC | FSC | PUR | ACC | FSC | PUR | ACC | FSC | PUR | ACC | FSC | PUR | ACC | FSC | PUR | ACC | FSC |
| OSCW | 10.68 | 10.25 | 5.52 | 10.64 | 9.94 | 5.52 | 10.85 | 10.17 | 5.49 | 40.92 | 40.57 | 33.68 | 40.97 | 41.13 | 33.63 | 39.72 | 39.79 | 33.16 |
| WSCW | **11.17** | **10.80** | **5.89** | **11.25** | **10.53** | **5.88** | **11.54** | **10.74** | **5.83** | **42.50** | **42.12** | **35.21** | **42.83** | **42.83** | **35.24** | **41.04** | **41.04** | **34.59** |
| | DEOLOG | | | | | | | | | YALTHREE | | | | | | | | |
| OSCW | 30.44 | 21.01 | 17.57 | 30.71 | 22.89 | 17.74 | 30.44 | 22.34 | 18.03 | 23.13 | 22.27 | 6.89 | 20.55 | 19.58 | 5.17 | 21.16 | 20.25 | 5.53 |
| WSCW | **31.84** | **22.32** | **18.45** | **31.84** | **23.74** | **18.41** | **31.28** | **24.25** | **19.56** | **24.94** | **23.55** | **7.89** | **21.73** | **20.61** | **6.30** | **21.55** | **20.70** | **6.70** |
| | BGFEA | | | | | | | | | AWTEN | | | | | | | | |
| OSCW | 20.98 | 20.94 | 18.88 | 21.38 | 21.72 | 19.43 | 21.18 | 20.98 | 19.01 | 19.52 | 11.42 | 10.39 | 19.52 | 11.25 | 10.37 | 19.52 | 11.25 | 10.35 |
| WSCW | **22.08** | **22.08** | **20.08** | **22.64** | **22.40** | **20.16** | **22.48** | **22.28** | **20.20** | **20.74** | **12.24** | **11.01** | **20.17** | **12.01** | **10.97** | **20.18** | **12.06** | **10.94** |
| | HDIGTWO | | | | | | | | | YOUFOURV | | | | | | | | |
| OSCW | 11.03 | 10.93 | 9.44 | 11.13 | 10.98 | 9.51 | 11.19 | 11.11 | 9.54 | 15.07 | 9.96 | 9.55 | 14.94 | 9.86 | 9.53 | 14.91 | 9.89 | 9.54 |
| WSCW | **11.73** | **11.51** | **10.03** | **11.75** | **11.60** | **10.11** | **11.79** | **11.70** | **10.22** | **15.99** | **10.91** | **10.48** | **15.95** | **10.85** | **10.47** | **15.95** | **10.80** | **10.47** |

*Table 12.* Clustering Comparison under Missing Ratio Being 0.3

| DATASET | FLOEVEN | | | SYNTHREED | | | DEOLOG | | | YALTHREE | | |
|---|---|---|---|---|---|---|---|---|---|---|---|---|
| METRIC | PUR | ACC | FSC | PUR | ACC | FSC | PUR | ACC | FSC | PUR | ACC | FSC |
| LRTL | 10.58 | 10.67 | 8.18 | 38.36 | 39.36 | 32.75 | 27.42 | 20.55 | 16.42 | 20.09 | 20.76 | 7.11 |
| TCIMC | 11.31 | 10.21 | 8.26 | 38.74 | 37.72 | 31.43 | 31.74 | 18.43 | 15.48 | 20.63 | 17.65 | 8.31 |
| AGCIM | 9.41 | 8.46 | 8.36 | 33.67 | 33.67 | 28.33 | 31.82 | 21.26 | 17.67 | 21.88 | 18.45 | 7.33 |
| LSIMV | 9.06 | 10.13 | 5.32 | 36.43 | 38.83 | 33.27 | 31.74 | 17.88 | 15.88 | 18.47 | 15.73 | 7.23 |
| GIMC | 8.13 | 8.86 | 5.83 | 36.53 | 34.84 | 32.96 | 31.27 | 18.27 | 17.38 | 18.58 | 16.32 | 6.23 |
| IMVCI | 9.92 | 9.94 | 5.81 | 40.71 | 41.71 | 33.22 | 31.96 | 20.96 | 18.35 | 21.97 | 18.39 | 6.70 |
| PIMVC | 10.83 | 9.29 | 6.32 | 40.53 | 39.86 | 31.37 | 30.77 | 23.21 | 17.99 | 21.39 | 18.00 | 7.61 |
| HCCGL | 7.14 | 6.83 | 5.72 | 38.00 | 34.67 | 34.97 | 31.58 | 20.46 | 18.11 | 22.45 | 21.45 | 8.29 |
| USETL | 10.20 | 9.56 | 7.35 | 39.33 | 39.33 | 29.95 | 29.73 | 19.05 | 18.12 | 19.94 | 20.67 | 8.49 |
| LBIMV | 10.66 | 10.46 | 5.01 | 41.07 | 41.74 | 27.90 | 30.98 | 20.98 | 17.59 | 21.18 | 18.97 | 5.22 |
| UIMC | 10.88 | 10.44 | 6.24 | 36.17 | 39.56 | 33.27 | 33.02 | 19.43 | 15.82 | 20.82 | 18.61 | 5.36 |
| OURS | 11.63 | 10.90 | 5.96 | 42.40 | 42.40 | 34.92 | 32.12 | 22.91 | 18.48 | 22.91 | 21.61 | 6.64 |
| | BGFEA | | | AWTEN | | | HDIGTWO | | | YOUFOURV | | |
| LRTL | 17.57 | 20.16 | 19.02 | 16.79 | 10.18 | 7.88 | 10.25 | 10.63 | 9.07 | - | - | - |
| TCIMC | 18.37 | 20.83 | 17.72 | 16.84 | 11.33 | 10.84 | 10.73 | 10.24 | 8.63 | - | - | - |
| AGCIM | 19.43 | 20.16 | 15.89 | 18.33 | 10.37 | 10.57 | 11.67 | 10.77 | 8.72 | - | - | - |
| LSIMV | 17.27 | 19.36 | 19.62 | 20.77 | 11.12 | 10.14 | 11.57 | 9.12 | 8.23 | - | - | - |
| GIMC | 19.79 | 19.13 | 18.66 | 19.78 | 9.32 | 9.89 | 10.94 | 8.11 | 7.73 | 10.84 | 10.27 | 8.42 |
| IMVCI | 18.78 | 19.79 | 15.74 | 20.53 | 8.47 | 10.79 | 11.27 | 11.12 | 8.74 | - | - | - |
| PIMVC | 20.64 | 20.51 | 17.23 | 20.61 | 9.87 | 9.35 | 10.49 | 8.11 | 8.78 | 15.24 | 10.36 | 8.75 |
| HCCGL | 22.57 | 19.28 | 18.95 | 20.11 | 9.87 | 8.23 | 10.57 | 10.97 | 8.27 | - | - | - |
| USETL | 20.36 | 19.11 | 16.84 | 20.18 | 7.84 | 9.11 | 10.32 | 8.12 | 8.34 | 10.53 | 9.32 | 8.92 |
| LBIMV | 19.12 | 21.94 | 18.38 | 20.33 | 10.41 | 8.71 | 8.36 | 11.72 | 8.85 | 15.88 | 9.98 | 9.27 |
| UIMC | 19.87 | 18.27 | 18.46 | 18.69 | 11.37 | 9.36 | 9.26 | 10.43 | 9.17 | - | - | - |
| OURS | 22.16 | 21.88 | 20.06 | 20.85 | 12.14 | 11.02 | 11.81 | 11.57 | 10.07 | 16.02 | 10.82 | 10.48 |

# N. Stability and Reliability

In order to demonstrate the stability and reliability of devised model, we record the standard deviations and present them in Fig. 6 where D-1 ∼ D-8 are the alternative name of datasets in Table 1 correspondingly. According to this figure, we can observe that even under various missing ratios' scenarios, the value of standard deviation is fairly small relative to the mean value. This provides evidence that our model is stable and the produced results are reliable.

*Table 13.* Clustering Comparison under Missing Ratio Being 0.4

| DATASET | FLOEVEN | | | SYNTHREED | | | DEOLOG | | | YALTHREE | | |
|---|---|---|---|---|---|---|---|---|---|---|---|---|
| METRIC | PUR | ACC | FSC | PUR | ACC | FSC | PUR | ACC | FSC | PUR | ACC | FSC |
| LRTL | 11.07 | 9.42 | 7.57 | 39.56 | 41.78 | 33.92 | 29.33 | 19.49 | 17.40 | 19.24 | 22.82 | 5.71 |
| TCIMC | 10.43 | 9.73 | 7.87 | 37.74 | 36.68 | 30.63 | 31.86 | 18.47 | 16.43 | 21.43 | 18.32 | 5.87 |
| AGCIM | 9.04 | 8.01 | 6.73 | 33.83 | 33.83 | 29.18 | 31.36 | 22.30 | 17.86 | 22.03 | 18.03 | 8.29 |
| LSIMV | 7.97 | 9.23 | 4.33 | 36.23 | 37.79 | 33.73 | 31.76 | 17.26 | 14.46 | 17.69 | 15.62 | 8.23 |
| GIMC | 7.84 | 7.73 | 6.74 | 36.74 | 32.57 | 33.26 | 31.26 | 18.32 | 16.47 | 19.47 | 16.57 | 6.73 |
| IMVCI | 10.27 | 9.62 | 5.65 | 41.68 | 39.68 | 34.72 | 30.56 | 21.56 | 18.38 | 19.52 | 20.64 | 7.31 |
| PIMVC | 10.71 | 9.38 | 5.12 | 42.17 | 37.17 | 32.87 | 30.88 | 21.77 | 18.22 | 17.79 | 14.79 | 6.11 |
| HCCGL | 7.06 | 6.91 | 5.36 | 40.18 | 37.33 | 32.41 | 30.53 | 20.97 | 17.31 | 19.61 | 20.12 | 7.37 |
| USETL | 9.57 | 8.97 | 6.88 | 38.04 | 37.48 | 29.01 | 29.63 | 19.51 | 17.89 | 20.85 | 19.85 | 7.04 |
| LBIMV | 10.96 | 9.04 | 5.47 | 40.51 | 40.53 | 30.24 | 30.18 | 21.18 | 18.51 | 21.33 | 19.32 | 5.52 |
| UIMC | 11.18 | 9.83 | 6.06 | 35.67 | 40.67 | 33.16 | 31.43 | 20.26 | 16.53 | 20.64 | 18.82 | 6.67 |
| OURS | 11.25 | 10.47 | 5.86 | 41.83 | 41.67 | 34.85 | 32.12 | 22.63 | 18.47 | 22.21 | 21.33 | 6.23 |
| | BGFEA | | | AWTEN | | | HDIGTWO | | | YOUFOURV | | |
| LRTL | 19.76 | 19.76 | 19.33 | 17.74 | 10.11 | 8.67 | 12.53 | 9.72 | 9.21 | - | - | - |
| TCIMC | 19.21 | 21.47 | 17.65 | 15.73 | 10.76 | 9.79 | 11.53 | 9.83 | 8.73 | - | - | - |
| AGCIM | 20.48 | 20.40 | 15.78 | 18.00 | 11.42 | 10.32 | 10.86 | 11.01 | 8.46 | - | - | - |
| LSIMV | 19.34 | 18.57 | 18.88 | 19.28 | 11.09 | 9.72 | 11.89 | 7.65 | 7.89 | - | - | - |
| GIMC | 17.87 | 18.86 | 20.23 | 20.97 | 10.25 | 10.35 | 11.73 | 8.93 | 8.17 | 11.72 | 9.32 | 8.74 |
| IMVCI | 22.03 | 19.68 | 15.33 | 20.56 | 9.20 | 10.05 | 11.83 | 10.98 | 9.03 | - | - | - |
| PIMVC | 22.98 | 20.66 | 16.72 | 20.37 | 10.02 | 9.08 | 10.37 | 9.23 | 9.23 | 15.88 | 10.47 | 9.42 |
| HCCGL | 20.52 | 18.36 | 18.57 | 20.17 | 10.21 | 8.14 | 11.53 | 10.26 | 9.74 | - | - | - |
| USETL | 20.61 | 18.12 | 17.21 | 20.13 | 7.39 | 8.97 | 9.43 | 9.21 | 7.87 | 11.32 | 9.74 | 8.77 |
| LBIMV | 19.16 | 21.43 | 17.74 | 20.09 | 9.45 | 8.83 | 8.70 | 10.44 | 8.90 | 15.42 | 10.32 | 9.24 |
| UIMC | 22.37 | 19.21 | 18.21 | 20.14 | 11.63 | 9.47 | 9.96 | 10.78 | 8.63 | - | - | - |
| OURS | 22.28 | 22.16 | 20.11 | 20.58 | 12.06 | 10.98 | 11.94 | 11.61 | 10.09 | 16.00 | 10.87 | 10.48 |

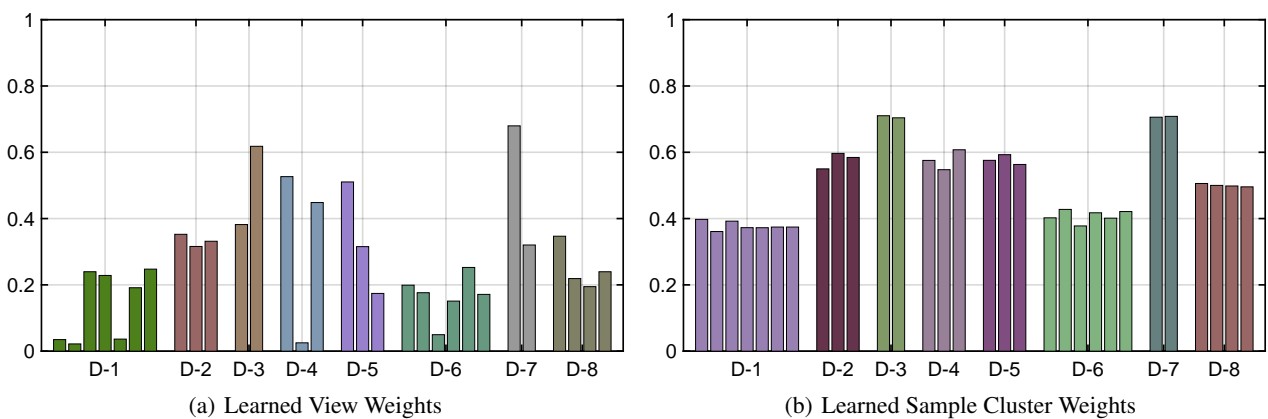

(a) Learned View Weights  (b) Learned Sample Cluster Weights

*Figure 5.* Visualization of View Weights and Sample Cluster Weights

## O. Weight Visualization

We associate views and sample clusters with learnable weight vectors respectively to adaptively measure their contributions. To validate that it indeed learns different weights for views and sample clusters respectively, we visualize these weights, as illustrated in Fig. 5. D-1 ~ D-8 are the alternative name of datasets in Table 1. It is easy to see that on each dataset, the view weights learned are different and the sample cluster weights learned are also different, which indicates that our optimization procedure is functional. Additionally, in conjunction with Table 10 and 11, we have that the weight strategy brings performance increasement. Therefore, we can state that the devised weight strategy works well with incomplete multi-view clustering.

*Table 14.* Clustering Comparison under Missing Ratio Being 0.6

| DATASET | FLOEVEN | | | SYNTHREED | | | DEOLOG | | | YALTHREE | | |
|---|---|---|---|---|---|---|---|---|---|---|---|---|
| METRIC | PUR | ACC | FSC | PUR | ACC | FSC | PUR | ACC | FSC | PUR | ACC | FSC |
| LRTL | 11.05 | 10.47 | 7.34 | 37.83 | 40.83 | 30.68 | 31.42 | 20.70 | 16.26 | 18.06 | 19.82 | 5.98 |
| TCIMC | 10.56 | 10.38 | 6.68 | 36.89 | 35.43 | 30.55 | 31.27 | 20.78 | 16.47 | 20.36 | 18.36 | 8.10 |
| AGCIM | 8.97 | 8.24 | 6.95 | 33.67 | 33.67 | 29.41 | 32.91 | 23.07 | 17.55 | 19.98 | 17.03 | 6.22 |
| LSIMV | 8.24 | 8.58 | 7.82 | 35.86 | 35.86 | 34.55 | 32.58 | 17.56 | 15.37 | 17.49 | 15.74 | 6.37 |
| GIMC | 8.37 | 8.47 | 7.32 | 34.62 | 32.97 | 35.37 | 31.94 | 17.67 | 16.58 | 18.85 | 17.46 | 6.73 |
| IMVCI | 11.16 | 9.68 | 5.84 | 41.03 | 37.18 | 30.47 | 31.49 | 23.49 | 17.87 | 20.21 | 19.33 | 7.23 |
| PIMVC | 11.13 | 9.28 | 8.33 | 36.14 | 35.18 | 31.47 | 30.18 | 21.07 | 18.31 | 16.70 | 14.27 | 8.01 |
| HCCGL | 7.28 | 6.99 | 5.36 | 38.17 | 38.17 | 31.66 | 30.04 | 21.76 | 17.91 | 18.88 | 21.52 | 6.82 |
| USETL | 9.59 | 8.92 | 7.81 | 37.61 | 37.61 | 28.64 | 27.42 | 21.28 | 17.33 | 20.71 | 19.06 | 6.72 |
| LBIMV | 9.81 | 10.93 | 5.57 | 40.50 | 41.58 | 29.46 | 30.77 | 22.77 | 17.57 | 20.12 | 19.09 | 6.25 |
| UIMC | 10.81 | 9.29 | 5.93 | 34.67 | 38.33 | 33.09 | 29.94 | 19.89 | 16.53 | 20.64 | 18.42 | 6.86 |
| OURS | 11.54 | 10.74 | 5.98 | 42.50 | 41.50 | 35.08 | 31.28 | 24.02 | 18.53 | 21.33 | 20.36 | 5.87 |
| | BGFEA | | | AWTEN | | | HDIGTWO | | | YOUFOURV | | |
| LRTL | 19.37 | 21.97 | 21.22 | 17.89 | 10.23 | 8.33 | 10.98 | 10.27 | 9.12 | - | - | - |
| TCIMC | 18.67 | 20.42 | 18.92 | 16.83 | 10.31 | 10.32 | 10.63 | 10.42 | 8.45 | - | - | - |
| AGCIM | 20.32 | 20.20 | 16.48 | 19.84 | 9.53 | 10.11 | 11.36 | 9.87 | 9.46 | - | - | - |
| LSIMV | 19.35 | 17.92 | 17.97 | 16.47 | 10.64 | 10.33 | 11.32 | 8.58 | 8.27 | - | - | - |
| GIMC | 19.13 | 17.85 | 18.16 | 19.32 | 9.26 | 9.38 | 10.63 | 8.47 | 7.78 | 10.74 | 9.52 | 8.53 |
| IMVCI | 22.68 | 22.68 | 16.83 | 20.58 | 8.92 | 9.69 | 10.97 | 12.43 | 9.88 | - | - | - |
| PIMVC | 20.60 | 20.54 | 16.79 | 20.51 | 10.28 | 9.67 | 10.24 | 8.73 | 8.66 | 15.48 | 9.32 | 9.62 |
| HCCGL | 20.56 | 18.32 | 18.77 | 20.23 | 10.83 | 8.44 | 9.43 | 11.24 | 7.43 | - | - | - |
| USETL | 19.06 | 18.01 | 17.33 | 20.21 | 8.59 | 10.23 | 8.37 | 8.89 | 8.43 | 9.87 | 9.11 | 9.27 |
| LBIMV | 21.96 | 19.76 | 15.35 | 20.14 | 9.05 | 7.51 | 9.37 | 9.64 | 9.00 | 14.32 | 9.48 | 9.22 |
| UIMC | 21.37 | 18.27 | 18.45 | 19.67 | 10.16 | 8.96 | 10.27 | 9.75 | 8.49 | - | - | - |
| OURS | 22.53 | 22.24 | 20.16 | 20.79 | 12.07 | 10.96 | 11.80 | 11.69 | 10.15 | 15.93 | 10.81 | 10.47 |

*Table 15.* Clustering Comparison under Missing Ratio Being 0.7

| DATASET | FLOEVEN | | | SYNTHREED | | | DEOLOG | | | YALTHREE | | |
|---|---|---|---|---|---|---|---|---|---|---|---|---|
| METRIC | PUR | ACC | FSC | PUR | ACC | FSC | PUR | ACC | FSC | PUR | ACC | FSC |
| LRTL | 11.03 | 10.42 | 7.40 | 35.43 | 40.43 | 32.43 | 31.06 | 21.25 | 17.24 | 19.12 | 20.48 | 6.14 |
| TCIMC | 11.07 | 10.43 | 7.12 | 37.38 | 36.62 | 30.32 | 30.88 | 18.56 | 15.76 | 20.17 | 18.12 | 6.75 |
| AGCIM | 8.90 | 8.31 | 6.96 | 34.17 | 34.00 | 28.80 | 30.11 | 23.26 | 18.12 | 20.73 | 17.12 | 6.73 |
| LSIMV | 8.12 | 9.23 | 7.32 | 34.67 | 34.72 | 32.36 | 31.17 | 17.47 | 15.35 | 17.74 | 15.63 | 7.22 |
| GIMC | 7.73 | 8.23 | 6.43 | 34.85 | 31.87 | 33.43 | 32.87 | 18.32 | 16.78 | 19.74 | 15.78 | 7.21 |
| IMVCI | 9.70 | 9.95 | 5.66 | 39.58 | 38.74 | 31.59 | 30.24 | 23.24 | 17.38 | 20.16 | 20.79 | 8.62 |
| PIMVC | 11.27 | 8.36 | 7.48 | 34.33 | 32.50 | 31.39 | 30.19 | 21.08 | 18.13 | 16.52 | 13.97 | 7.11 |
| HCCGL | 7.13 | 6.78 | 5.76 | 36.00 | 36.00 | 32.97 | 29.57 | 22.17 | 18.38 | 20.67 | 20.27 | 7.14 |
| USETL | 9.41 | 8.85 | 7.33 | 36.51 | 36.51 | 28.13 | 30.41 | 20.38 | 18.88 | 19.88 | 20.06 | 7.89 |
| LBIMV | 9.12 | 10.06 | 5.81 | 39.17 | 36.17 | 27.11 | 30.57 | 20.30 | 17.68 | 21.33 | 18.52 | 5.35 |
| UIMC | 10.74 | 10.37 | 5.98 | 42.67 | 40.67 | 30.92 | 31.18 | 22.47 | 16.73 | 20.61 | 17.20 | 6.32 |
| OURS | 11.81 | 11.01 | 5.90 | 41.83 | 41.50 | 35.00 | 31.28 | 24.02 | 18.68 | 20.91 | 19.82 | 5.83 |
| | BGFEA | | | AWTEN | | | HDIGTWO | | | YOUFOURV | | |
| LRTL | 18.68 | 20.84 | 18.25 | 17.72 | 10.81 | 8.29 | 10.26 | 11.03 | 9.23 | - | - | - |
| TCIMC | 18.73 | 19.83 | 18.27 | 15.68 | 9.57 | 9.78 | 10.26 | 10.37 | 7.84 | - | - | - |
| AGCIM | 20.32 | 20.28 | 14.45 | 14.11 | 9.84 | 11.08 | 9.87 | 10.41 | 8.50 | - | - | - |
| LSIMV | 19.16 | 17.88 | 18.26 | 17.86 | 11.29 | 9.47 | 10.47 | 8.37 | 7.94 | - | - | - |
| GIMC | 19.72 | 19.53 | 18.13 | 18.37 | 8.97 | 9.63 | 11.33 | 8.86 | 7.43 | 11.28 | 9.32 | 8.43 |
| IMVCI | 18.73 | 20.73 | 17.47 | 20.05 | 8.11 | 10.22 | 11.25 | 11.32 | 8.28 | - | - | - |
| PIMVC | 21.74 | 20.74 | 16.38 | 20.36 | 10.11 | 9.25 | 9.39 | 9.51 | 8.47 | 16.36 | 10.89 | 8.94 |
| HCCGL | 21.35 | 18.36 | 18.27 | 20.21 | 10.14 | 7.87 | 9.15 | 9.32 | 7.83 | - | - | - |
| USETL | 19.24 | 19.09 | 15.83 | 20.14 | 8.74 | 9.46 | 8.64 | 8.47 | 8.47 | 9.49 | 8.32 | 9.56 |
| LBIMV | 20.84 | 20.47 | 18.36 | 20.07 | 8.34 | 8.61 | 8.11 | 10.49 | 9.03 | 15.23 | 9.74 | 8.72 |
| UIMC | 20.72 | 17.75 | 18.33 | 18.47 | 11.32 | 9.36 | 9.67 | 9.06 | 8.01 | - | - | - |
| OURS | 22.60 | 22.40 | 20.19 | 20.32 | 11.92 | 10.95 | 11.91 | 11.80 | 10.17 | 15.98 | 10.88 | 10.48 |

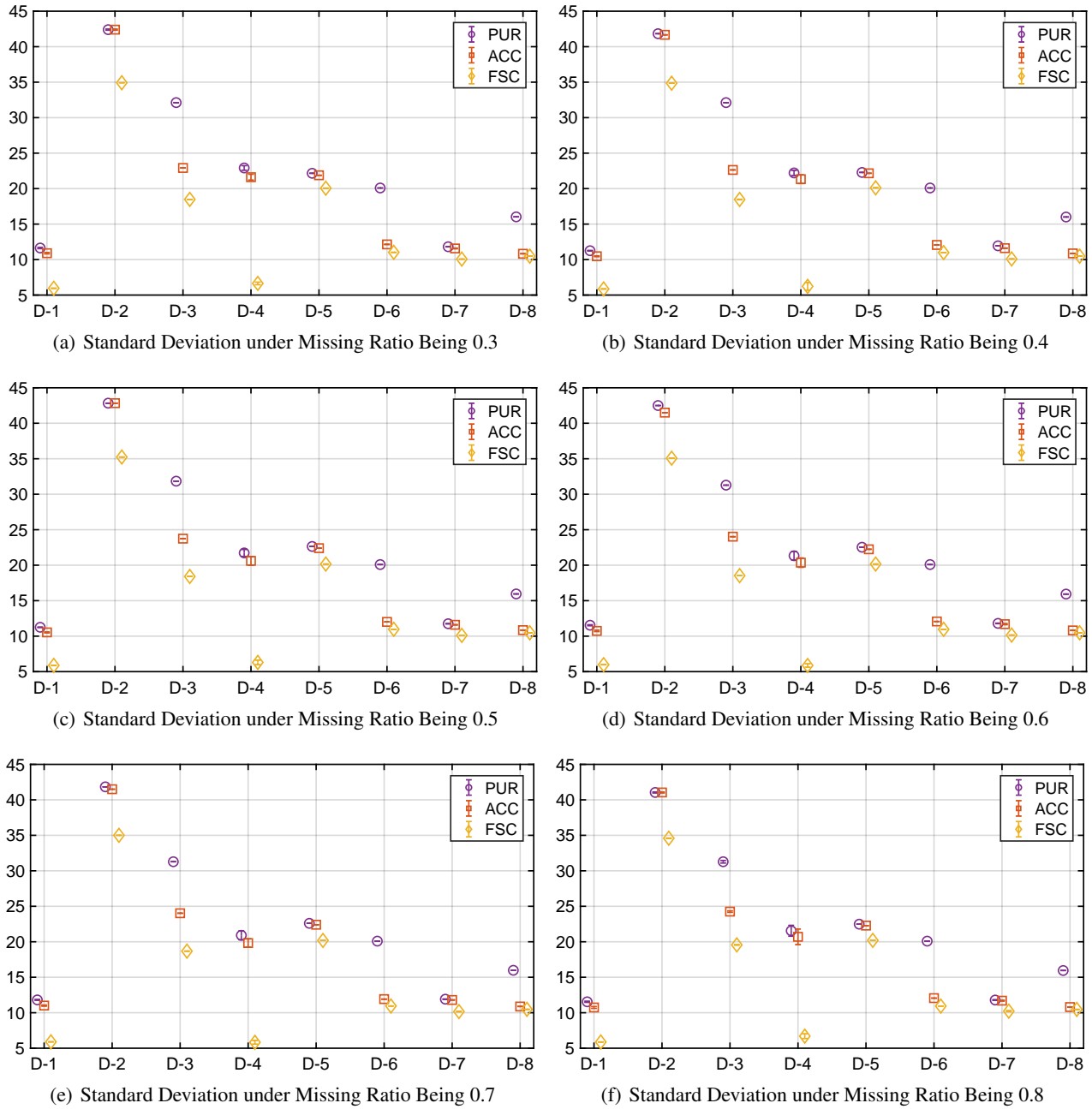

(a) Standard Deviation under Missing Ratio Being 0.3

(b) Standard Deviation under Missing Ratio Being 0.4

(c) Standard Deviation under Missing Ratio Being 0.5

(d) Standard Deviation under Missing Ratio Being 0.6

(e) Standard Deviation under Missing Ratio Being 0.7

(f) Standard Deviation under Missing Ratio Being 0.8

*Figure 6.* Errorbar of Standard Deviation and Mean Value

