# OpenReview forum: "Bifurcate then Alienate: Incomplete Multi-view Clustering via Coupled Distribution Learning with Linear Overhead"
_ICML.cc/2025/Conference — ICML 2025 poster_

### Official Review · Reviewer_hkPK · 2025-03-11

**Overall Recommendation:** 4

**Summary:**

In the paper, the authors propose a dual-determinant incomplete multi-view clustering algorithm BACDL. They partition feature clusters through a bifurcation scheme, and alienate bifurcations to differentiate determinants. With coupled distribution learning, it alleviates the dimension inconsistency by introducing view guidance.  Then, they bridge all views based on the principle between marginal distribution and conditional distribution, and reorder all incomplete sample clusters in potential space to construct the clustering embedding.  Finally, compared experiments with different missing ratios are organized to reveal the merits of BACDL.

**Claims And Evidence:**

The provided evidence supports the claims.

**Essential References Not Discussed:**

None

**Experimental Designs Or Analyses:**

Yes. The designs confirm the effectiveness of proposed BACDL.

**Methods And Evaluation Criteria:**

The method is applicable to large-scale clustering scenarios.

**Other Comments Or Suggestions:**

None

**Other Strengths And Weaknesses:**

Strengths,

-  The designed dual-determinant learning paradigm and coupled distribution learning paradigm for IMC issues are interesting, and owns reference value for further study.

-  The solution is technically sound, making linear overhead and ensuring convergence.

-  The introduction of sample cluster transformation for misregistration is novel. Experiments are evaluated from multiple perspectives.


 Weaknesses

 The discussions regarding Remark 1 appears overly concise. It is recommended to provide more details.

-  Regarding TCIMC and LSIMV (without feature cluster bifurcating), they receive preferable results on DEOLOG.  From table 2, there is a remarkable 2.19 PUR interval. These is no interpretation for this phenomenon.

-  The application of orthogonal rotation in the potential representation space to reorganize the sample clusters could deteriorate the distribution attribute. This implies that the cluster label quality is highly dependent on the rotation.

**Questions For Authors:**

-   About the number of feature clusters and the number of sample clusters, how to set their values? Based on eq (1), one can have that these numbers can be arbitrary and only the association needs to satisfy the dimensionality constraints.

-  The view guidance plays a role in avoiding the dimension difference, and is related to the common representation matrix to extract perspective-shared features. If assigning it to the samples (This also can learn shared features since all sample dimensions are consistent after projecting.  Moreover, the dual-determinant representations are concurrently learned in the same dimension space), will that facilitate increases in the results?

-  Instead of the unified association, if using the framework in eq (1) to construct the sample clusters and then remapping them, it seems that the loss 1 is not necessary. In this case, is the performance still competitive? Could you please provide some empirical evidence for this baseline?

**Relation To Broader Scientific Literature:**

The ability to be applicable to large-scale scenarios highlights its practicality.

**Theoretical Claims:**

Yes. No issues were found.

---

> ### Author Rebuttal · Authors · 2025-04-01
>
> **Q1:**  More details regarding Remark 1.
>
> **A1:** Thanks. Due to $\mathbf{X} _r\in\mathbb{R}^{d_r\times n}$ and $\mathbf{G} _r\in\mathbb{R}^{n\times n_r}$, computing $q=\|\mathbf{X} _r\mathbf{G} _r-\widehat{\mathbf{A}} _r\mathbf{E} _r^{\top}\mathbf{G} _r\| _F^2$ will take  at least $\mathcal{O}(n n_r)$ overhead. Note that $\mathbf{G}_r$ consists of 0 and 1, and there is only one element that is 1 in each column. $\mathbf{G} _r\mathbf{G} _r^{\top}$ is diagonal with 0 and 1. $\mathbf{X} _r\mathbf{G} _r\mathbf{G} _r^{\top}$ and $\mathbf{E} _r^{\top}\mathbf{G} _r\mathbf{G} _r^{\top}$ equal to $\mathbf{X} _r\odot\mathbf{O} _r$ and $\mathbf{E} _r^{\top}\odot\mathbf{Q} _r$. $\mathbf{O} _r$ is $\mathbf{1} _{d _r}\cdot[\sum _{j=1}^{n _v}(\mathbf{G} _r) _{1,j},\cdots,\sum _{j=1}^{n _v}(\mathbf{G} _r) _{n,j}]$ and $\mathbf{Q} _r$ is $\mathbf{1} _{k}\cdot[\sum _{j=1}^{n _v}(\mathbf{G} _r) _{1,j},\cdots,\sum _{j=1}^{n _v}(\mathbf{G} _r) _{n,j}]$. So, we have $q$ equals to $\|\mathbf{X} _r\odot\mathbf{O} _r-\widehat{\mathbf{A}} _r\mathbf{E} _r^{\top}\odot\mathbf{Q} _r\| _F^2$. Computing the latter needs $\mathcal{O}(d_rn)$, which is $\mathcal{O}(n)$.
>
> **Q2:** Interpretation regarding TCIMC and LSIMV.
>
> **A2:** TCIMC uses the tensor Schatten p-norm to explore complementary and spatial structure. Despite enhancing view interaction, it induces intensive time overhead due to the tensor operation, and is unsuitable for large-scale tasks. LSIMV constructs sparse and structured representations. It adopts a norm based sparse constraint to generate low-dimensional features, and uses local embedding to do aggregation. This needs a relatively larger memory cost due to the almost full-sized graph.
>
> **Q3:** Does the orthogonal rotation affect the cluster label quality?
>
> **A3:** This rotation reorganizes the sample clusters to relieve misregistration. Although altering the distribution, kindly note that the final common sample clusters are also required to be orthogonal, which keeps pace with the rotation. See the following comparisons. NRN: not involve rotation; ART: contains rotation.
>
> |FLOEVEN||||||||||
> |:---:|:---:|:---:|:---:|:---:|:---:|:---:|:---:|:---:|:---:|
> ||0.2|||0.5|||0.8|||
> ||PUR|ACC|FSC|PUR|ACC|FSC|PUR|ACC|FSC|
> |NRN|10.57|10.13|5.18|10.35|10.13|5.13|9.98|9.39|5.07|
> |ART|**11.17**|**10.80**|**5.89**|**11.25**|**10.53**|**5.88**|**11.54**|**10.74**|**5.83**|
> |**SYNTHREED**||||||||||
> |NRN|38.17|38.17|33.68|37.67|36.33|33.55|38.50|37.00|33.65|
> |ART|**42.50**|**42.12**|**35.21**|**42.83**|**42.83**|**35.24**|**41.04**|**41.04**|**34.59**|
>
> **Q4:** How to set the numbers of feature clusters and sample clusters?
>
> **A4:** The model based on eq (1) mainly shows that this paradigm can factorize multi-view data as feature clusters and sample clusters as well as association to mine latent patterns. For generality, the numbers of feature clusters and sample clusters can be any value (smaller than the feature dimension and sample size). In practice, we set their values just as $k$. The benefits are multifold. Dataset will be divided into $k$ groups, whether from the view of samples or features. The misregistration will be relieved owing to the small sample cluster number. Also, this makes the association small-sized, saving resources.
>
> **Q5:** Does assigning view guidance to samples (VGS) facilitate the results?
>
> **A5:** This a solution to avoid dimension difference. The following experiments reveal the performance. DAM: devised assigning mechanism.
>
> |FLOEVEN||||||||||
> |:---:|:---:|:---:|:---:|:---:|:---:|:---:|:---:|:---:|:---:|
> |VGS|10.83|10.32|5.47|10.77|9.73|5.58|10.63|10.07|5.37|
> |DAM|**11.17**|**10.80**|**5.89**|**11.25**|**10.53**|**5.88**|**11.54**|**10.74**|**5.83**|
> |**SYNTHREED**||||||||||
> |VGS|41.53|40.93|33.97|41.74|42.17|34.82|39.92|39.83|33.64|
> |DAM|**42.50**|**42.12**|**35.21**|**42.83**|**42.83**|**35.24**|**41.04**|**41.04**|**34.59**|
>
> The reasons for superior results of VGS are that this operation filters out noisy and data redundancy. For inferior results, possible reasons are that the operation causes information loss in diversity.
>
> **Q6:** Performance when using the association in eq (1).
>
> **A6:** This will build an association on each view for feature clusters and sample clusters separately. SAS: separated association scheme in eq (1); DUA: devised unified association.
>
> |FLOEVEN||||||||||
> |:---:|:---:|:---:|:---:|:---:|:---:|:---:|:---:|:---:|:---:|
> |SAS|10.95|10.50|5.35|10.54|10.24|5.76|11.43|10.60|**6.01**|
> |DUA|**11.17**|**10.80**|**5.89**|**11.25**|**10.53**|**5.88**|**11.54**|**10.74**|5.83|
> |**SYNTHREED**||||||||||
> |SAS|40.51|40.57|**35.95**|41.48|41.37|34.46|40.76|**41.76**|33.18|
> |DUA|**42.50**|**42.12**|35.21|**42.83**|**42.83**|**35.24**|**41.04**|41.04|**34.59**|
>
> Under eq (1), the view communication could be inadequate due to the separability. Also, this could fail to propagate feature cluster information across views into sample clusters since the association is established individually.

---

> > ### Comment · Reviewer_hkPK · 2025-04-07
> >
> > The author's rebuttal has addressed my concerns, so I have decided to raise my score to accept.

---

> > > ### Author Response · Authors · 2025-04-07
> > >
> > > Dear Reviewer hkPK,
> > >
> > > Thank you for your encouraging words! We will further enhance the manuscript in line with your expert suggestions.
> > >
> > > Best Wishes,
> > >
> > > The authors

---

### Official Review · Reviewer_KBMg · 2025-03-11

**Overall Recommendation:** 4

**Summary:**

This work aims to alleviate the issue of single-determinant paradigm in incomplete multi-view clustering. It introduces distribution learning and associates each type of determinants to bifurcated feature clusters. Through mutual exclusion learning and view guidance learning, it eliminates the dimension inconsistency and enlarges the determinant distinction. All views are interconnected together based on the distribution principle. After rotating and remapping, incomplete sample clusters are formed into full clustering embedding. A nine-step updating rule with overall linear overhead and theoretical convergence efficiently minimizes the objective function.

**Claims And Evidence:**

Utilizing dual perspective determinants to encode cluster representations is under-studied in incomplete multi-view clustering. I agree with that. Table 4 and 5 illustrate its effectiveness.

**Essential References Not Discussed:**

No.

**Experimental Designs Or Analyses:**

Yes. I checked experimental designs. They are valid.

**Methods And Evaluation Criteria:**

The authors conduct comparing experiments under several missing ratios to illustrate the effectiveness in tackling IMC issues

**Other Comments Or Suggestions:**

See the weakness above.

**Other Strengths And Weaknesses:**

Strengths:
1)  The algorithm analysis is comprehensive, covering complexity, convergence, sensitivity, etc.
2)  Experiments are solid, involving diverse missing ratios and data scales. Ablation is also thorough.
3)  The flow chart of the proposed model is intuitive, and the function of each part is clear.

Weaknesses:
1)   During formulating Eq.(3),  the descriptions about the sample cluster distribution needs further enhancement.
2)   The space rotation operation utilized could alter the value of learned sample clusters. This influence requires more illustrations.
3)   The initialization lacks clarity, for instance, the cluster association.
4)   A systematic analysis of each term's contribution to the objective loss would significantly strengthen the motivation.

**Questions For Authors:**

1. As mentioned, the final clustering results are derived by performing spectral clustering on the common sample clusters. The sample clusters from each view contribute to the formation of the common sample clusters.  A question arises: why do the common sample clusters not adhere to the similar constraints as the view sample clusters?

2. The number of samples observed on each view $n_r$ is embedded into the original data matrix, how is it merged during updating the view coefficient? If not merged, is the computing overhead impacted by it? In practice, whether does it affect the running speed?

3. When generating the clustering embedding by accumulating sample clusters on all views vertically, how does the algorithm performance behave? possible reasons?

**Relation To Broader Scientific Literature:**

The key contributions in this paper are the dual perspective determinant cluster coding and the solution with overall linear overhead, which may serve as a groundwork for subsequent studies.

**Theoretical Claims:**

Yes. They are correct.

---

> ### Author Rebuttal · Authors · 2025-04-01
>
> **Q1:** More descriptions on the sample cluster distribution in Eq.(3).
>
> **A1:** Thanks. Each row of the sample cluster $\mathbf{E}_r$ represents a probability distribution. So, for each row of $\mathbf{E}_r$, its sum needs to be 1. For handling the incompleteness, we introduce the index matrix $\mathbf{G}_r$ which consists of 0 and 1. We can formulate the observed sample clusters as $\mathbf{G} _r^{\top}\mathbf{E} _r$. Hence, we have  the incomplete sample clusters need to satisfy $\mathbf{G} _r^{\top}\mathbf{E} _r\mathbf{1} _k =\mathbf{1} _{n _r}$.
>
> **Q2:** More illustrations on the space rotation influence.
>
> **A2:** It mainly serves as reordering sample clusters to make them as consistent as possible. Note that the common sample clusters are subject to the similar constraints as space operation, which can help tackle the negative elements that rotation operation brings. Please see the following performance comparisons. DM: direct mapping; RO: rotation operation.
>
> |FLOEVEN||||||||||
> |:---:|:---:|:---:|:---:|:---:|:---:|:---:|:---:|:---:|:---:|
> ||0.2|||0.5|||0.8|||
> ||PUR|ACC|FSC|PUR|ACC|FSC|PUR|ACC|FSC|
> |DM|10.57|10.13|5.18|10.35|10.13|5.13|9.98|9.39|5.07|
> |RO|**11.17**|**10.80**|**5.89**|**11.25**|**10.53**|**5.88**|**11.54**|**10.74**|**5.83**|
> |**SYNTHREED**||||||||||
> |DM|38.17|38.17|33.68|37.67|36.33|33.55|38.50|37.00|33.65|
> |RO|**42.50**|**42.12**|**35.21**|**42.83**|**42.83**|**35.24**|**41.04**|**41.04**|**34.59**|
> |**DEOLOG**||||||||||
> |DM|30.62|21.51|17.96|30.34|21.89|18.16|30.23|22.01|18.07|
> |RO|**31.84**|**22.32**|**18.45**|**31.84**|**23.74**|**18.41**|**31.28**|**24.25**|**19.56**|
>
> One can observe that after space rotation, the performance receives improvement.
>
> **Q3:** The initialization lacks clarity.
>
> **A3:** Thanks! We create a random matrix with element from 0 to 1, and then do column-normalization. We use it to initialize $\mathbf{P}_r$, $\mathbf{C}_r$ and $\mathbf{C}$. For $\mathbf{F}_r$ and $\mathbf{E}$, we use random orthogonal matrix to initialize them. We use the row-normalized random matrix with element from 0 to 1 to initialize $\mathbf{E}_r$, and the random matrix with element from 0 to 1 to initialize $\mathbf{D}$. For $a_r$ and $b_r$, we initialize them with $1/v$ and $1/\sqrt{v}$.
>
> **Q4:** Analysis of each term's contribution.
>
> **A4:** The first term is deemed as a error reconstruction, and extracts feature clusters and sample clusters as well as their association. It also adaptively balances the important of each view. The second term separates feature clusters from sample clusters to highlight their discrimination by point-to-point alienation. The third term mitigates the misregistration and constructs complete sample clusters.
>
> **Q5:** Why do the common sample clusters not adhere to the similar constraints as the view sample clusters?
>
> **A5:** We introduce orthogonal transformation to reformulate the view sample clusters, which inevitably brings negative elements. Note that the essence of view sample clusters is a probability distribution, and accordingly all elements in them are non-negative. After transformation to produce common sample clusters, there will have negative elements. So, we make the common sample clusters adhere to similar constraints as the orthogonal transformation.
>
> **Q6:** How is $n_r$ merged during updating the view coefficient? Affect the running speed?
>
> **A6:**  Updating the view coefficient needs to compute $\left\|\mathbf{X} _r\mathbf{G}_r\right\| _F^2$. Note that for each column of $\mathbf{G}_r$, there is only one 1 and other elements are 0. We have $\left\|\mathbf{X} _r\mathbf{G} _r\right\| _F^2$ is equal to $\left\|\mathbf{X} _r\mathbf{G} _r\mathbf{G} _r^{\top}\right\| _F^2$. The diagonal elements of $\mathbf{G} _r\mathbf{G} _r^{\top}$ are 1 or 0 and other elements are 0. Consequently, we have $\mathbf{X} _r\mathbf{G} _r\mathbf{G} _r^{\top}$ aims to select some columns of $\mathbf{X}_r$. So, we have that $\left\|\mathbf{X} _r\mathbf{G} _r\right\| _F^2$ is equal to $\left\|\mathbf{X} _r\odot\mathbf{O} _r\right\| _F^2$. $\mathbf{O}_r$ is $\mathbf{1} _{d _r}\cdot[\sum _{j=1}^{n_v}(\mathbf{G} _r) _{1,j},\cdots,\sum _{j=1}^{n _v}(\mathbf{G} _r) _{n,j} ]$. Before merging, the computing overhead is $\mathcal{O}(n n_r)$. After merging, it is $\mathcal{O}(n)$ and not affected by $n_r$.
>
> **Q7:** When accumulating sample clusters vertically, how does the algorithm behave? Reasons?
>
> **A7:** For the performance comparison, please refer to A7 in Reviewer xp7C. We attribute this phenomenon to three points. Accumulating sample clusters on all views vertically could result in  inadequate communication between sample clusters across different views, which is not conductive to formulating rich representations. This also could not automatically measure the importance of different sample clusters.  Meanwhile, the misregistration will disturb the cluster structure, and consequently degrades the formulated representations.

---

> > ### Comment · Reviewer_KBMg · 2025-04-04
> >
> > My comments have been partially addressed in the rebuttal. I raise my rating to accept..

---

> > > ### Author Response · Authors · 2025-04-04
> > >
> > > Dear Reviewer KBMg,
> > >
> > > Many thanks for recognizing our contributions! We will further polish the manuscript according to your profound and professional guidance in the further.
> > >
> > >
> > > Best Wishes,
> > >
> > > The authors

---

### Official Review · Reviewer_xp7C · 2025-03-13

**Overall Recommendation:** 4

**Summary:**

A BACDL algorithm with dual-determinant learning is specially designed for incomplete multi-view clustering (IMC) in this paper.  It bifurcates feature clusters and further alienates them through mutual exclusion learning to strengthen the discrimination. It alleviates the dimension inconsistency, and bridges all views by unifying the association between feature clusters and sample clusters. The full clustering embedding is formulated by weighted space rotation and remapping. Theoretical analysis demonstrates its linear overhead and convergence. Experimental results under multiple missing ratios validate its effectiveness.

**Claims And Evidence:**

This paper is organized in a clear manner and provides some insights for incomplete multi-view clustering. I understand the motivation of dual-determinant scheme. The feature cluster bifurcating highlights the novelty.

**Essential References Not Discussed:**

None

**Experimental Designs Or Analyses:**

The experimental designs are reasonable.

**Methods And Evaluation Criteria:**

This paper compares the results with multiple methods in three metrics.

**Other Comments Or Suggestions:**

Refer to the above comments

**Other Strengths And Weaknesses:**

The strengths:

1. The motivation is clearly illustrated and the overall organization is logically-structured.
2. The idea that bifurcates feature clusters and alienates via mutual exclusion learning is novel to a certain degree.
3. The authors conduct extensive experiments and also make moderate discussions about the results.

The weaknesses:

1. Introducing view guidance may induce additional computing complexity, which could impair the efficiency goal.
2. The reasons why introducing common sample clusters and why setting them to be orthogonal are not illustrated in depth.

**Questions For Authors:**

1.  Does the missing ratio affect the complexity of BACDL? If yes, how? As illustrated in the updating rule, the index matrix is associated with the data matrix.
2. Why it is necessary to guarantee the matrix C to be column-normalized?
3. How does the element-wise multiplication operation decrease the computational overhead?
4.  Why learning the sample clusters on each view respectively? Why not directly learning common sample clusters? Is this beneficial for the performance improvement?
5. Rather than formulating the full clustering embedding by mapping the sample clusters of each view, when stacking them, there is no common plane, how does the model perform?

.

**Relation To Broader Scientific Literature:**

The feature cluster bifurcating and alienating could provide further reference for incomplete multi-view clustering.

**Theoretical Claims:**

It is ok in convergence and complexity.

---

> ### Author Rebuttal · Authors · 2025-04-01
>
> **Q1:** View guidance may impair the efficiency goal.
>
> **A1:** Thanks. The computing cost of view guidance is linear, and thus hardly affects the efficiency goal. It requires constructing $\mathbf{X} _r\mathbf{G} _r\mathbf{G} _r^{\top}\mathbf{E} _r\mathbf{D} _{\gamma}^{\top}\mathbf{C}^{\top}$, $\mathbf{C} _r\mathbf{C}^{\top}$ and $[\mathbf{P} _r\mathbf{C}|\mathbf{C} _r]\mathbf{D}\mathbf{E} _r^{\top}\mathbf{G} _r\mathbf{G} _r^{\top}\mathbf{E} _r\mathbf{D} _{\gamma}^{\top}\mathbf{C}^{\top}$, which takes $\mathcal{O}(d_rn+d_rnk+d_rk^2)$, $\mathcal{O}(d_rk^2)$, $\mathcal{O}( nk + 2d_rk^2 + d_rnk)$. So, it totally takes $\mathcal{O}(d_rnk+d_rk^2)$. This is $\mathcal{O}(n)$ and consistent with the efficiency goal.
>
> **Q2:** Why introducing common orthogonal sample clusters?
>
> **A2:** They gather all view sample clusters to formulate full embedding. Due to the missing instances, the sample clusters on each view are incomplete. The orthogonality plays a role in enhancing the separability of learned common sample clusters to better group samples.
>
> **Q3:** Does the missing ratio affect the complexity of BACDL?
>
> **A3:** It does not affect the complexity. The index matrix containing missing ratio owns the properties, i.e., there is only one element that is 1 while the other elements are 0 in each column. We have $\mathbf{G} _r\mathbf{G} _r^{\top}$ is diagonal with elements either 0 or 1. So, $\mathbf{X} _r\mathbf{G} _r\mathbf{G} _r^{\top}$ equals to $\mathbf{X} _r\odot\mathbf{O} _r$ where $\mathbf{O} _r=\mathbf{1} _{d _r}\left[\sum _{j=1}^{n _v}(\mathbf{G} _r) _{1,j}, \cdots,\sum _{j=1}^{n _v}(\mathbf{G} _r) _{n,j}\right]$. This takes $\mathcal{O}(d_rn)$ cost and is irrelevant to the missing ratio.
>
> **Q4:** Why guaranteeing C column-normalized?
>
> **A4:** Each column of feature clusters denotes a probability distribution on all feature dimensions. Each column sum needs to add up to 1. After bifurcating, each part also should conform to this point. On the basis of column-normalized $\mathbf{P}_r$, we derive that the premise for $\mathbf{P} _r\mathbf{C}$ being column-normalized is that $\mathbf{C}$ only needs to be column-normalized.
>
> **Q5:** How does the element-wise operation reduce the computational overhead?
>
> **A5:** This mainly benefits from the equivalent element transformation. For $\mathbf{E}_r$, directly calculating $\left\|\mathbf{E} _r^{\top}\mathbf{G} _r\right\| _F^2$ will take $\mathcal{O}(knn_r)$. As the missing ratio decreases, $n_r$ is gradually increasing. The overhead is almost close to  $\mathcal{O}(n^2)$.
>
> Note that $\left\|\mathbf{E} _r^{\top}\mathbf{G} _r\right\| _F^2$ is equal to $\left\|\mathbf{E} _r^{\top}\mathbf{G} _r\mathbf{G} _r^{\top}\right\| _F^2$. Then, by the element-wise operation, $\left\|\mathbf{E} _r^{\top}\mathbf{G} _r\mathbf{G} _r^{\top}\right\| _F^2$ is $\left\|\mathbf{E} _r^{\top}\odot\mathbf{Q} _r\right\| _F^2$,
> which takes $\mathcal{O}(nk)$. $\mathbf{Q}_r$ is $\mathbf{1} _{k}\left[\sum _{j=1}^{n _v}(\mathbf{G} _r) _{1,j}, \cdots, \sum _{j=1}^{n _v}(\mathbf{G} _r) _{n,j}\right]$.
>
> **Q6:** Why learning the sample clusters on each view respectively (SEV)? Why not directly learning common ones?
>
> **A6:** The former may facilitate capturing the view characteristics. Learning common sample clusters (CSC) is a feasible scheme, and yet could weaken the view data diversity.
>
> |FLOEVEN||||||||||
> |:---:|:---:|:---:|:---:|:---:|:---:|:---:|:---:|:---:|:---:|
> ||0.2|||0.5|||0.8|||
> ||PUR|ACC|FSC|PUR|ACC|FSC|PUR|ACC|FSC|
> |CSC|10.47|10.64|4.78|10.52|9.32|**5.92**|10.25|9.36|4.57|
> |SEV|**11.17**|**10.80**|**5.89**|**11.25**|**10.53**|5.88|**11.54**|**10.74**|**5.83**|
> |**SYNTHREED**||||||||||
> |CSC|40.72|39.89|34.73|40.37|39.87|33.83|38.87|37.84|31.59|
> |SEV|**42.50**|**42.12**|**35.21**|**42.83**|**42.83**|**35.24**|**41.04**|**41.04**|**34.59**|
> |**DEOLOG**||||||||||
> |CSC|30.68|20.57|17.58|**31.93**|19.82|17.63|28.93|22.47|18.97|
> |SEV|**31.84**|**22.32**|**18.45**|31.84|**23.74**|**18.41**|**31.28**|**24.25**|**19.56**|
>
> **Q7:** When stacking sample clusters (STA), the performance?
>
> **A7:** Please see the following table. MSC: mapping sample clusters.
>
> |FLOEVEN||||||||||
> |:---:|:---:|:---:|:---:|:---:|:---:|:---:|:---:|:---:|:---:|
> ||0.2|||0.5|||0.8|||
> ||PUR|ACC|FSC|PUR|ACC|FSC|PUR|ACC|FSC|
> |STA|10.23|10.14|4.93|10.33|9.57|5.01|10.14|9.52|4.38|
> |MSC|**11.17**|**10.80**|**5.89**|**11.25**|**10.53**|**5.88**|**11.54**|**10.74**|**5.83**|
> |**SYNTHREED**||||||||||
> |STA|40.05|38.97|33.98|40.73|38.56|33.69|36.73|36.40|30.79|
> |MSC|**42.50**|**42.12**|**35.21**|**42.83**|**42.83**|**35.24**|**41.04**|**41.04**|**34.59**|
> |**DEOLOG**||||||||||
> |STA|29.06|20.82|16.62|30.29|21.87|17.79|28.75|22.85|18.51|
> |MSC|**31.84**|**22.32**|**18.45**|**31.84**|**23.74**|**18.41**|**31.28**|**24.25**|**19.56**|
>
> Directly stacking sample clusters could lead to insufficient interaction among views. Moreover, this does not adaptively balance their contributions. Misregistration also will weaken the quality of sample clusters.

---

### Official Review · Reviewer_7KrE · 2025-03-14

**Overall Recommendation:** 3

**Summary:**

This paper introduces a new incomplete multi-view clustering (IMC) algorithm named BACDL. It simultaneously explores both perspective-shared and perspective-specific determinants through coupled distribution learning, with linear overhead. The approach bifurcates feature clusters and enhances discrimination via alienation and mutual exclusion learning. Extensive experiments validate the effectiveness of BACDL across multiple large-scale datasets and benchmarks, with results outperforming several state-of-the-art methods in terms of accuracy and efficiency.

**Claims And Evidence:**

The claims in the submission are not fully supported by clear evidence. Specifically, Theorem 5 lacks clarity regarding whether convergence refers to the objective function value or iterations. Additionally, the term "global optimal" in Theorems 1 and 2 is used imprecisely, without specifying if it refers to the global minimum of the original problem or the proxy problem. This paper also fails to clarify whether the function is strictly convex, which is crucial for ensuring a global minimum. These issues weaken the theoretical rigor of the claims.

**Essential References Not Discussed:**

A thorough review of the existing works that are based on NMF should be included, such as:

[1] Wen, J., Zhang, Z., Zhang, Z., Zhu, L., Fei, L., Zhang, B., & Xu, Y. (2021, May). Unified tensor framework for incomplete multi-view clustering and missing-view inferring. In Proceedings of the AAAI conference on artificial intelligence (Vol. 35, No. 11, pp. 10273-10281).

[2] Wen, J., Xu, G., Tang, Z., Wang, W., Fei, L., & Xu, Y. (2023). Graph regularized and feature aware matrix factorization for robust incomplete multi-view clustering. IEEE Transactions on Circuits and Systems for Video Technology, 34(5), 3728-3741.

**Experimental Designs Or Analyses:**

The experiments are very thorough. However, in the analysis of the results, the paper should provide a more quantitative evaluation of the performance gains. Additionally, hypothesis testing should be included to confirm the statistical significance of the observed performance improvements.

**Methods And Evaluation Criteria:**

This paper validates the proposed method using multiple large-scale datasets, which is appropriate for assessing its effectiveness and efficiency. The experimental design is well-structured, and a variety of widely used evaluation metrics are employed, providing sufficient evidence to support the claims. The experiments are comprehensive and thoroughly validate the proposed approach.

**Other Comments Or Suggestions:**

1.In Algorithm 1, it should be (g^h-g^{h+1}) rather than (g^{h+1}-g^h).

2.Figure 2: Include units for runtime to provide clearer context.

3.Ensure that all definitions (e.g., Definition 1) are properly referenced, specifying the exact source from which they are drawn. Additionally, this paper should cite the relevant literature for the Majorization-Minimization (MM) framework used in the optimization step, as this method is central to the proposed approach.

**Other Strengths And Weaknesses:**

Strengths:

1.Well-written and organized. This paper is clear, logically structured, and easy to follow.

2.Novel Approach. Introduces BACDL, addressing the limitations of current methods by capturing both perspective-shared and perspective-specific determinants, improving clustering performance.

3.Comprehensive Literature Review. Relevant works are reviewed in depth, providing a strong foundation for the proposed method.

4.Thorough Theoretical and Experimental Analysis. This paper includes detailed theoretical proofs, optimization steps, and a well-structured experimental setup. Performance is validated using large datasets and state-of-the-art baselines.

5.Efficiency: The algorithm exhibits linear overhead in terms of time and space, demonstrating scalability to large-scale datasets.

Weaknesses:

1.Convergence and Approximation Guarantees. Theorem 1 proves the validity of the proxy problem, demonstrating that the proposed proxy problem is a valid approximation that can represent the original problem for optimization. However, Theorem 2 proves that solving this proxy problem guarantees a decrease in the objective function value, but it does not provide an analysis of convergence at the iteration level. In other words, while the objective value decreases, there is no clear proof that the algorithm will converge to a fixed point in a finite number of iterations. For approximation methods, in addition to convergence in terms of the objective value, it is important to prove an approximation guarantee, which would clarify the relationship between the solution to the approximate problem (i.e., the proxy problem) and the solution to the original problem. Many approximation methods provide such guarantees, and incorporating this would enhance the theoretical rigor of this work.

2.Ambiguities in Theoretical Details. (1) In Theorem 5, it is unclear which level of convergence is being discussed. The theorem mentions the convergence of the algorithm, but it is not specified whether this refers to convergence in the objective function value or convergence in terms of iterations. It is crucial to clarify this to prevent potential misinterpretation, as readers might assume iteration convergence is guaranteed when only objective value convergence is proven. (2) Similarly, the use of the term “global optimal” in Theorem 1 and Theorem 2 needs to be more precise. This paper uses the term "global solution" without clearly specifying whether it refers to the global minimum of the original problem or the approximate solution to the proxy problem. It would be more precise to refer to the "global minimum" of the proxy problem, especially in the context of convexity. Additionally, since a semi-definite Hessian matrix ensures that the function is convex, it only guarantees a global minimum if the function is strictly convex (i.e., the Hessian matrix is positive definite). If the function is not strictly convex, multiple global solutions might exist. The paper should clarify these points for better theoretical rigor.

3.Lack of Justification for Method Choices. Although non-negative matrix factorization (NMF) is used in the proposed method, this paper does not provide a clear justification for why NMF is preferred over other potential techniques. There are various other matrix factorization methods available, and it would be useful to explain why NMF is particularly suited for this incomplete multi-view clustering (IMC) problem. Additionally, the choice of perspective-shared and perspective-specific determinants is made from the feature clusters' perspective, but the paper does not provide any reasoning as to why this approach is chosen rather than considering the sample clusters' perspective. Moreover, while distribution learning and mutual exclusion learning are critical components of the algorithm, their roles and necessity are not clearly explained.

4.Performance Gains and Statistical Validation. Although this paper demonstrates significant performance gains of the proposed BACDL algorithm over several state-of-the-art methods, these gains are not sufficiently quantified statistically. For example, the performance comparison could benefit from hypothesis testing to determine whether the differences observed in performance metrics (e.g., clustering accuracy, purity) are statistically significant. This would provide stronger evidence that the observed improvements are not merely due to random variation.

5.Lack Thorough Review on NMF-based Approaches. Although the related work overview is comprehensive, the proposed method is based on Non-negative Matrix Factorization (NMF). Therefore, a thorough review of the existing works that are based on NMF should be included, such as:
[1] Wen, J., Zhang, Z., Zhang, Z., Zhu, L., Fei, L., Zhang, B., & Xu, Y. (2021, May). Unified tensor framework for incomplete multi-view clustering and missing-view inferring. In Proceedings of the AAAI conference on artificial intelligence (Vol. 35, No. 11, pp. 10273-10281).
[2] Wen, J., Xu, G., Tang, Z., Wang, W., Fei, L., & Xu, Y. (2023). Graph regularized and feature aware matrix factorization for robust incomplete multi-view clustering. IEEE Transactions on Circuits and Systems for Video Technology, 34(5), 3728-3741.

6.Writing and Structural Issues. (1) In the introduction, this paper mentions several limitations of the proposed method but fails to address the consequences of these limitations. (2) Section Transitions: There is a lack of smooth transitions between some sections, which affects the readability of the paper. For example, the introduction of Definition 1 (lines 180-188) feels abrupt and lacks context—it's unclear why this definition is introduced at this point. Similarly, Theorem 3 is presented without explaining its relevance or application, leaving readers to wonder what it contributes to the overall algorithm. (3) Example Clarifications: In some sections, such as the one introducing the index matrix G_r, it would be helpful to provide a concrete example to better illustrate its structure and functionality.

7.Complexity Analysis. The complexity analysis section provides a general overview of the computational costs, but it fails to account for the number of iterations required for convergence. Given that the algorithm involves several steps of optimization, the number of iterations could significantly affect the time complexity, especially for large-scale datasets. This paper does not clarify whether more iterations are needed when handling larger datasets or whether the algorithm's performance scales well with the number of iterations.

8.Reproducibility Issues. While the paper provides extensive experimental results, it does not offer any code or datasets, which makes it difficult for other researchers to verify the results or build upon this work. Providing access to the implementation would greatly enhance the transparency and reproducibility of the research.

**Questions For Authors:**

1.How do the authors ensure that the algorithm converges in a finite number of iterations? Can the authors provide proof of convergence in terms of the number of iterations?

2.Can the authors provide an approximation guarantee for the solution to the approximate problem used in the optimization? How does this compare to the exact solution?

3.This paper uses the term "global optimal" in the theoretical proofs. Is this referring to a global minimum of the objective function, or is it specific to the approximation method?

4.Why are perspective-shared and perspective-specific determinants modeled from the feature clusters' perspective rather than from the sample clusters' perspective?

**Relation To Broader Scientific Literature:**

The key contributions of the paper are well-positioned in relation to the broader scientific literature. The paper reviews several state-of-the-art methods, and compared to these existing approaches, the main contribution lies in simultaneously considering both perspective-shared and perspective-specific determinants. Additionally, the paper introduces a new algorithm designed to solve complex optimization problems. One of the notable aspects of the proposed algorithm is its linear overload, which distinguishes it from other methods in the literature. This contribution offers a novel approach to tackling the problem, improving efficiency and effectiveness in ways not previously explored.

**Theoretical Claims:**

I have checked the correctness of the proofs for the theoretical claims in the paper, particularly for Theorem 1 and Theorem 2. However, there is an issue with the claim of "global optimal" in these theorems. This paper does not clearly specify whether "global optimal" refers to the global minimum of the original problem or the approximate solution to the proxy problem. Additionally, the use of the term "global" is imprecise, as it does not account for cases where multiple global solutions may exist, especially if the function is not strictly convex. This lack of rigor weakens the theoretical clarity of the claims.

---

> ### Author Rebuttal · Authors · 2025-04-01
>
> Sincerely thank Reviewer 7KrE for the very constructive comments.
>
> **Q1:** Providing iteration level convergence and approximation guarantees would enhance the theoretical rigor.
>
> **A1:** Many thanks! This is a challenging task for authors during the rebuttal period. At this moment, authors have no inspiring ideas about it. So sorry for this. We will strive to explore this topic in future work.
>
> **Q2:** Ambiguities in theoretical details. Like the convergence level, global optimal.
>
> **A2:** Thanks! We will carefully proofread the manuscript to state them more precisely. Specially, the convergence  refers to the objective function value. The global optimal refers to the approximate solution to the proxy problem.
>
> **Q3:** Lack of justification for method choices. Like why NMF.
>
> **A3:** NMF, subspace, kernel and neural network are four common means to tackle IMC problems. Kernel and neural network can perform nonlinear mapping well, while usually suffering from intensive complexity due to the large-sized feature matrix and complex network structure. Moreover, the selection of kernel type and network architecture also heavily relies on empirical knowledge. In virtue of the superior high-dimensional data processing capability, subspace effectively builds affinity via utilizing multiple potential spaces. However, due to the full-sized self-expression characteristics, it generally encounters cubic computing overhead. Unlike them, NMF mines shared low-dimensional structures by decomposing the view data matrix to discover latent clusters. In the decomposed basis vectors, each element can be seen as a contribution to the features, which makes the analysis results more intuitive and helps to understand the reasons for clustering. So, in the paper, we adopt the NMF technique.
>
> **Q4:** Performance gains and statistical validation.
>
> **A4:** In experiments, we run 50 times, and record the average value of clustering results. We will add these statistical information in the next version, and further analyze its features.
>
> **Q5:** Lack thorough review on NMF-based approaches.
>
> **A5:** Thanks! We will thoroughly review the mentioned works in the next version.
>
> **Q6:** Writing and structural Issues. Like section transitions, example clarifications.
>
> **A6:** Thanks! We will carefully polish the manuscript. For the matrix $\mathbf{G}_r$, it consists of 0 and 1, and there is only one 1 in each column.
>
> **Q7:** Complexity analysis. Considering the number of iterations required for convergence.
>
> **A7:** Good suggestion! The number of required iterations indeed affects the time complexity. Deriving the iterations required for convergence is a promising research direction. We will make efforts to explore this topic in future work.
>
> **Q8:** Reproducibility issues.
>
> **A8:** We will release the source code in the final version.
>
> **Q9:** What "global optimal" refers to.
>
> **A9:** It refers to that of the proxy problem.
>
> **Q10:** Why feature clusters' perspective? Why not sample clusters' perspective?
>
> **A10:** Kindly note the feature cluster matrix aims at mapping original data to a potential space, and learns the marginal distribution of original data. Bifurcating it will be conductive to extracting representative features from original data. It can be seen as a feature extractor. The sample cluster matrix mainly plays a role in grouping formed representations to generate clusters. A high-quality cluster partition relies on more discriminative representations. So, we adopt the feature clusters' perspective. Please see the following comparisons. BSC: Bifurcate sample clusters; BFC: Bifurcate feature clusters.
>
> |FLOEVEN||||||||||
> |:---:|:---:|:---:|:---:|:---:|:---:|:---:|:---:|:---:|:---:|
> ||0.2|||0.5|||0.8|||
> ||PUR|ACC|FSC|PUR|ACC|FSC|PUR|ACC|FSC|
> |BSC|10.42|10.11|5.12|9.63|10.12|5.21|10.52|10.13|5.01|
> |BFC|**11.17**|**10.80**|**5.89**|**11.25**|**10.53**|**5.88**|**11.54**|**10.74**|**5.83**|
> |**SYNTHREED**||||||||||
> |BSC|41.21|40.51|35.02|**42.88**|41.01|34.21|40.21|39.94|32.43|
> |BFC|**42.50**|**42.12**|**35.21**|42.83|**42.83**|**35.24**|**41.04**|**41.04**|**34.59**|
> |**DEOLOG**||||||||||
> |BSC|31.21|21.73|18.12|30.76|21.42|17.14|30.87|**24.62**|19.21|
> |BFC|**31.84**|**22.32**|**18.45**|**31.84**|**23.74**|**18.41**|**31.28**|24.25|**19.56**|
> |**YALTHREE**||||||||||
> |BSC|23.21|21.67|7.62|20.52|19.43|6.11|20.62|19.62|6.53|
> |BFC|**24.94**|**23.55**|**7.89**|**21.73**|**20.61**|**6.30**|**21.55**|**20.70**|**6.70**|
> |**BGFEA**||||||||||
> |BSC|21.32|21.42|19.72|21.32|21.22|19.67|21.42|21.22|19.87|
> |BFC|**22.08**|**22.08**|**20.08**|**22.64**|**22.40**|**20.16**|**22.48**|**22.28**|**20.20**|
> |**AWTEN**||||||||||
> |BSC|19.32|**12.26**|10.64|19.86|11.94|10.22|20.01|11.46|10.31|
> |BFC|**20.74**|12.24|**11.01**|**20.17**|**12.01**|**10.97**|**20.18**|**12.06**|**10.94**|
>
> Evidently, BFC is more desirable than BSC in most cases, revealing that feature clusters' perspective is more preferable.

---

> > ### Comment · Reviewer_7KrE · 2025-04-08
> >
> > Thanks for your response. I have read it. Combining the overall contributions of this paper and the response, I choose to keep my score.

---

> > > ### Author Response · Authors · 2025-04-08
> > >
> > > Dear Reviewer 7KrE,
> > >
> > >
> > > Thank you sincerely for acknowledging our research contributions!  We will strive to refine the manuscript following your fairly constructive recommendations.
> > >
> > > Best Wishes,
> > >
> > > The authors

---

### Decision · Program_Chairs · 2025-05-01

**Decision:**

Accept (poster)

**Comment:**

All reviewers recognized the quality of the work.  The work has the following strengths:
* Novel idea
* Effective and efficient algorithm
* Theoretical supports
* Comprehensive numerical evaluation

According to the feedback of the reviewers on the authors' rebuttals, all questions and minor issues have been addressed.